# Exploiting Similarity for Computation and Communication-Efficient Decentralized Optimization

Yuki Takezawa [1 2]  Xiaowen Jiang [3 4]  Anton Rodomanov [3]  Sebastian U. Stich [3]

## Abstract

Reducing communication complexity is critical for efficient decentralized optimization. The proximal decentralized optimization (PDO) framework is particularly appealing, as methods within this framework can exploit functional similarity among nodes to reduce communication rounds. Specifically, when local functions at different nodes are similar, these methods achieve faster convergence with fewer communication steps. However, existing PDO methods often require highly accurate solutions to subproblems associated with the proximal operator, resulting in significant computational overhead.

In this work, we propose the Stabilized Proximal Decentralized Optimization (SPDO) method, which achieves state-of-the-art communication and computational complexities within the PDO framework. Additionally, we refine the analysis of existing PDO methods by relaxing subproblem accuracy requirements and leveraging average functional similarity. Experimental results demonstrate that SPDO significantly outperforms existing methods.

## 1. Introduction

Decentralized optimization is coming to be a pivotal paradigm due to its use for privacy preservation and efficient training on large-scale datasets (Nedic & Ozdaglar, 2009; Lian et al., 2017; Koloskova et al., 2020). In decentralized optimization, nodes have different loss functions and are connected by an underlying network topology. Then, the nodes exchange parameters with their neighbors every round and minimize the average loss function.

One of the most important challenges in decentralized optimization is reducing communication costs since parameter exchanges between nodes are often more expensive than local computation at each node due to high latency and limited bandwidth of the network (Lian et al., 2017; Wang et al., 2019). A key technique to reduce the communication complexity is for each node to perform multiple local updates before the communication. The concept of multiple local updates was initially applied for Decentralized Gradient Descent (Nedic & Ozdaglar, 2009; Lian et al., 2017) and Gradient Tracking (Lorenzo & Scutari, 2016; Pu & Nedic, 2021). However, increasing the number of local steps requires a smaller learning rate, which ultimately requires the same communication cost to achieve the desired accuracy level (Koloskova et al., 2020; Liu et al., 2024a). This is because, in Decentralized Gradient Descent, each node tries to minimize its own local loss function by performing gradient descent multiple times, but fully minimizing its local loss function is not desirable. Gradient Tracking also has the same issue, and this inconsistency hinders the use of multiple local updates to reduce communication complexity. Thus, to take advantage of multiple local updates, we need to resolve this inconsistency.

An effective approach to address this challenge is to develop decentralized optimization methods based on the proximal-point method, where nodes solve the subproblem associated with the proximal operator in their local updates (termed PDO framework in this work). Scutari & Sun (2019) demonstrated that combining proximal-point updates with gradient tracking yields an efficient decentralized optimization scheme, known as SONATA. This method effectively exploits the similarity of local loss functions across nodes to significantly reduce communication complexity (Sun et al., 2022; Tian et al., 2022). However, achieving such low communication complexity requires solving the proximal subproblem with high accuracy at each node, leading to suboptimal computational overhead during local updates.

In this work, we pose the following question: *Can we develop a proximal decentralized optimization (PDO) method that achieves low communication complexity without sacrificing computational efficiency?* We provide an affirmative answer by introducing the **Stabilized Proximal Decentralized Optimization (SPDO)** method. SPDO improves upon

---

[1]Kyoto University [2]OIST [3]CISPA Helmholtz Center for Information Security [4]Saarland University. Correspondence to: Yuki Takezawa <yuki-takezawa@ml.ist.i.kyoto-u.ac.jp>.

*Proceedings of the 42$^{nd}$ International Conference on Machine Learning*, Vancouver, Canada. PMLR 267, 2025. Copyright 2025 by the author(s).

*Table 1.* Summary of communication communication and computational complexities in the **strongly-convex setting** (our additional results for the **convex case** are summarized in Appendix A). "# communication" represents the number of times that each node exchanges parameters with neighboring nodes to reach the target accuracy $\epsilon$, and "# computation" represents the number of gradient oracle queries required by each node. Since Tian et al. (2022) only showed the convergence rate of Accelerated Exact/Inexact-PDO with $\mu \leq \delta_{\max}$, we show the convergence rates under the assumption that $\mu \leq \delta_{\max}$ and $\mu \leq \delta$ for simplicity.

| Algorithm | Reference | # Communication | # Computation | Assumptions |
|---|---|---|---|---|
| Gradient Tracking | Koloskova et al. (2021) | $\mathcal{O}\left(\frac{L}{\mu(1-\rho)^2}\log(\frac{1}{\epsilon})\right)^{(a)}$ | $\mathcal{O}\left(\frac{L}{\mu(1-\rho)^2}\log(\frac{1}{\epsilon})\right)^{(a)}$ | 1, 2, 4 |
| Exact-PDO (SONATA) | Sun et al. (2022),  Th. $1^{(b)}$ | $\mathcal{O}\left(\frac{\delta}{\mu(1-\rho)}\log(\frac{L}{\delta})\log(\frac{1}{\epsilon})\right)^{(b)}$ | n.a.$^{(c)}$ | 1, 2, 4 |
| **Inexact-PDO** | Th. 1, 2 | $\mathcal{O}\left(\frac{\delta}{\mu(1-\rho)}\log(\frac{L}{\delta})\log(\frac{1}{\epsilon})\right)$ | $\mathcal{O}\left(\frac{\sqrt{\delta L}}{\mu}\log\left(\frac{1}{\epsilon}\right)\log\log\left(\frac{1}{\epsilon}\right)\right)$ | 1, 2, 4 |
| **Stabilized-PDO [new]** | Th. 3, 4 | $\mathcal{O}\left(\frac{\delta}{\mu(1-\rho)}\log(\frac{L}{\delta})\log(\frac{1}{\epsilon})\right)$ | $\mathcal{O}\left(\frac{\sqrt{\delta L}}{\mu}\log\left(\frac{1}{\epsilon}\right)\right)$ | 1, 2, 4 |
| Accelerated SONATA | Tian et al. (2022) | $\mathcal{O}\left(\sqrt{\frac{\delta_{\max}}{\mu(1-\rho)}}\log(\frac{L}{\delta})\log(\frac{\delta}{\mu})\log(\frac{1}{\epsilon})\right)^{(d)}$ | n.a.$^{(c)}$ | 1, 2, 3, 4 |
| Inexact Accelerated SONATA | Tian et al. (2022) | $\mathcal{O}\left(\sqrt{\frac{\delta_{\max}}{\mu(1-\rho)}}\log(\frac{L}{\delta})\log(\frac{\delta}{\mu})\log(\frac{1}{\epsilon})\right)^{(d)}$ | $\mathcal{O}\left(\frac{\sqrt{\delta_{\max}L}}{\mu}\log(\frac{\delta}{\mu})\left(\log(\frac{1}{\epsilon})\right)^2\right)^{(d)}$ | 1, 2, 3, 4 |
| **Accelerated Stabilized-PDO [new]** | Th. 5, 6 | $\mathcal{O}\left(\sqrt{\frac{\delta}{\mu(1-\rho)}}\log(\frac{L}{\mu})\log(\frac{1}{\epsilon})\right)$ | $\mathcal{O}\left(\sqrt{\frac{L}{\mu}}\log(\frac{1}{\epsilon})\right)$ | 1, 2, 4 |

(a) It holds that $\delta \leq L$, and $\delta \ll L$ holds when nodes have similar datasets.
(b) Sun et al. (2022) showed that the communication complexity of SONATA is $\tilde{\mathcal{O}}(\frac{\delta_{\max}}{\mu(1-\rho)}\log(\frac{1}{\epsilon}))$, while our new theorem, Theorem 1, shows that the dependence on $\delta_{\max}$ can be reduced to $\delta$.
(c) "n.a." represents that we need to solve the subproblem exactly.
(d) $\delta$ is smaller than $\delta_{\max}$ and can be up to $\sqrt{n}$ times smaller than $\delta_{\max}$.

existing approaches in several key aspects: (i) it requires less accurate solutions to local subproblems, resulting in better computational efficiency, and (ii) it leverages the *average* functional similarity across nodes, leading to more effective optimization. Furthermore, (iii) the accelerated variant of our method, **Accelerated-SPDO**, achieves state-of-the-art communication and computational complexities among decentralized optimization methods, as summarized in Table 1. More specifically, we make the following contributions.

- We propose Accelerated-SPDO, which can attain the best communication and computational complexities among the existing decentralized optimization methods.

- We also propose non-accelerated methods, called SPDO, which can attain the best communication and computational complexities among the existing non-accelerated decentralized optimization methods.

- To clarify the difference between (Accelerated-)SPDO and the existing methods, we provide a refined convergence analysis of various methods in the PDO framework. Our analysis recovers Exact-PDO, also known as SONATA (Sun et al., 2022), as a special instance. While SONATA assumed that the subproblem was exactly solved, our approach allows us to solve the subproblems approximately (Inexact-PDO). We then show that SPDO can solve the subproblem even more coarsely than Inexact-PDO.

- The existing analysis of SONATA showed that the communication complexity depends on the maximum dissimilarity of local functions $\delta_{\max}$ (Sun et al., 2022), while we prove that it depends only on the average dissimilarity $\delta$, which can significantly reduce communication costs.

- Moreover, the prior papers on PDO methods (Sun et al.,

2022; Tian et al., 2022, etc) analyzed the convergence rate only in the strongly-convex case, while we analyze all methods in both convex and strongly-convex cases.

**Notation:**   We write $\|\boldsymbol{x}\|$ for Euclidean norm of $\boldsymbol{x} \in \mathbb{R}^d$ and use $\tilde{\mathcal{O}}(\cdot)$ and $\tilde{\Omega}(\cdot)$ to hide polylogarithmic factors.

## 2. Related Work

**Decentralized Optimization:**   The most basic decentralized optimization method is Decentralized Gradient Descent (Nedic & Ozdaglar, 2009; Lian et al., 2017; Koloskova et al., 2020), while due to the data heterogeneity, Decentralized Gradient Descent cannot achieve the linear convergence rate in the strongly-convex case. Many researchers have tried to achieve the linear convergence rate, proposing EXTRA (Shi et al., 2015), NIDS (Li et al., 2019), Gradient Tracking (Lorenzo & Scutari, 2016; Pu & Nedic, 2021; Yuan et al., 2019; Takezawa et al., 2023; Liu et al., 2024b), and so on (Scaman et al., 2017; Kovalev et al., 2021). These methods can achieve the linear convergence rate, while they still involve a certain amount of communication cost even though nodes have very similar loss functions. The concept of second-order similarity was introduced to capture this phenomenon, in which $\delta$ and $\delta_{\max}$ measure the similarity among the loss functions held by nodes (see Definition 1 and Assumption 3) (Karimireddy et al., 2021; Murata & Suzuki, 2021; Kovalev et al., 2022). The quantities $\delta$ and $\delta_{\max}$ approach zero when the local loss functions come to be similar. To exploit this similarity, many variants of the proximal decentralized optimization method have been proposed (Li et al., 2020; Sun et al., 2022; Tian et al., 2022; Cao et al., 2025; Jiang et al., 2024a;b). Second-order similarity also

plays a crucial role in governing communication complexity in distributed learning (Karimireddy et al., 2021; Khaled & Jin, 2022; Zindari et al., 2023; Patel et al., 2024).

**Proximal-point Methods:** In this section, we briefly introduce Proximal-point Method. Applying Proximal-point Method to minimize $f$ yields the following update rule:

$$\boldsymbol{x}^{(r+1)} = \arg\min_{\boldsymbol{x} \in \mathbb{R}^d} \left\{ F_r(\boldsymbol{x}) \coloneqq f(\boldsymbol{x}) + \frac{\lambda}{2} \left\| \boldsymbol{x} - \boldsymbol{x}^{(r)} \right\|^2 \right\},$$

where $\lambda > 0$ is hyperparameter. The vanilla Proximal-point Method needs to solve the above subproblem exactly, which is infeasible in general. Many researchers have tried to avoid the exact proximal evaluations (Rockafellar, 1976; Solodov & Svaiter, 1999; 2001; Monteiro & Svaiter, 2013). Rockafellar (1976) showed that Proximal-point Method can attain the same convergence rate when $\|\nabla F_r(\boldsymbol{x}^{(r+1)})\| \leq \mathcal{O}(\frac{\lambda}{r}\|\boldsymbol{x}^{(r+1)} - \boldsymbol{x}^{(r)}\|)$ is satisfied. It implies that a more accurate subproblem solution is required as the number of iterations increases. To mitigate this issue, Solodov & Svaiter (1999) proposed Hybrid Projection Proximal-point Method, which can achieve the same convergence rate if $\|\nabla F_r(\boldsymbol{x}^{(r+1)})\| \leq \mathcal{O}(\lambda\|\boldsymbol{x}^{(r+1)} - \boldsymbol{x}^{(r)}\|)$ is satisfied. This condition is significantly weaker than the condition for the vanilla Proximal-point Method.

The accelerated Proximal-point Method (Güler, 1992) also has the same issue, requiring a more accurate subproblem solution as the number of iterations increases. Based on Hybrid Projection Proximal-point Method, Monteiro & Svaiter (2013) proposed its accelerated method and relaxed the condition for an approximate solution of the subproblem.

A recent line of work developed distributed versions of the Proximal-point Method (Shamir & Srebro, 2014; Li et al., 2020; Sun et al., 2022; Jiang et al., 2024a;b). We will further discuss these approaches in Sec. 6 below.

## 3. Problem Setup

In this work, we consider the following problem where the loss functions are distributed among $n$ nodes:

$$\min_{\boldsymbol{x} \in \mathbb{R}^d} f(\boldsymbol{x}), \quad f(\boldsymbol{x}) \coloneqq \frac{1}{n} \sum_{i=1}^{n} f_i(\boldsymbol{x}),$$

where $\boldsymbol{x}$ is the model parameter and $f_i$ is the loss function of $i$-th node. The nodes are connected by an undirected graph, in which $W_{ij} \in [0, 1]$ represents the edge weight between nodes $i$ and $j$, and $W_{ij} > 0$ if and only if nodes $i$ and $j$ are connected. The Metropolis-Hastings weight (Xiao et al., 2005) is a commonly used choice for $\{W_{ij}\}_{ij}$. Following the previous work (Li et al., 2020; Tian et al., 2022; Sun et al., 2022; Lin et al., 2023; Khaled & Jin, 2023; Jiang et al., 2024b), we assume that the loss functions satisfy the following assumptions.

**Assumption 1** (Strong Convexity)**.** There exists $\mu \geq 0$ such that for any $\boldsymbol{x}, \boldsymbol{y} \in \mathbb{R}^d$ and $i$, it holds that

$$f_i(\boldsymbol{x}) \geq f_i(\boldsymbol{y}) + \langle f_i(\boldsymbol{y}), \boldsymbol{x} - \boldsymbol{y} \rangle + \frac{\mu}{2} \|\boldsymbol{x} - \boldsymbol{y}\|^2. \quad (1)$$

**Assumption 2** (Smoothness)**.** There exists $L \geq 0$ such that for any $\boldsymbol{x}, \boldsymbol{y} \in \mathbb{R}^d$ and $i$, it holds that

$$\|\nabla f_i(\boldsymbol{x}) - \nabla f_i(\boldsymbol{y})\| \leq L\|\boldsymbol{x} - \boldsymbol{y}\|. \quad (2)$$

**Definition 1** (Similarity)**.** Under Assumption 2, let $\delta \geq 0$ the the smallest number such that it holds that

$$\frac{1}{n} \sum_{i=1}^{n} \|\nabla h_i(\boldsymbol{x}) - \nabla h_i(\boldsymbol{y})\|^2 \leq \delta^2 \|\boldsymbol{x} - \boldsymbol{y}\|^2, \quad (3)$$

for any $\boldsymbol{x}, \boldsymbol{y} \in \mathbb{R}^d$ where $h_i(\boldsymbol{x}) \coloneqq f(\boldsymbol{x}) - f_i(\boldsymbol{x})$.

**Lemma 1.** *If Assumption 2 holds, there exists $\delta$ such that $\delta \leq L$ and Eq. (3) holds.*

Apart from Definition 1, Assumption 3 is also commonly used in the existing literature (Tian et al., 2022; Sun et al., 2022). However, Assumption 3 needs to assume that $f_i$ is twice differentiable, and $\delta$ can be much smaller than $\delta_{\max}$. We use Definition 1 instead of Assumption 3 in this study.

**Assumption 3.** There exists $\delta_{\max} \geq 0$ such that for any $\boldsymbol{x} \in \mathbb{R}^d$ and $i$, it holds that

$$\left\| \nabla^2 f(\boldsymbol{x}) - \nabla^2 f_i(\boldsymbol{x}) \right\| \leq \delta_{\max}. \quad (4)$$

*Remark* 1. It holds that $\delta \leq \delta_{\max}$. Moreover, $\delta$ can be at most $\sqrt{n}$ times smaller than $\delta_{\max}$.

Then, we assume that edge weights $\{W_{ij}\}_{ij}$ satisfy the following assumption.

**Assumption 4** (Spectral Gap)**.** For any $i$ and $j$, $W_{ij} = W_{ji}$ and $\sum_{i=1}^{n} W_{ij} = 1$. Then, there exists $\rho \in [0, 1)$ such that for any $\boldsymbol{x}_1, \boldsymbol{x}_2, \dots, \boldsymbol{x}_n \in \mathbb{R}^d$, it holds that

$$\sum_{i=1}^{n} \left\| \sum_{j=1}^{n} W_{ij} \boldsymbol{x}_j - \bar{\boldsymbol{x}} \right\|^2 \leq \rho^2 \sum_{i=1}^{n} \left\| \boldsymbol{x}_i - \bar{\boldsymbol{x}} \right\|^2, \quad (5)$$

where $\bar{\boldsymbol{x}} \coloneqq \frac{1}{n} \sum_{i=1}^{n} \boldsymbol{x}_i$.

**Evaluation Metric:** As communication is often more expensive than local computation, the primary goal in developing decentralized optimization methods is to reduce communication complexity. If methods can achieve the same communication complexity, the method with less computational complexity is superior.

## 4. Refined Analysis of Proximal Decentralized Optimization Method

In this section, we briefly explain the framework of Proximal Decentralized Optimization (PDO) methods and provide the

---

**Algorithm 1** Proximal Decentralized Optimization Method (PDO)

---

1: **Input:** The number of rounds $R$, and hyperparameters $M$ and $\lambda$.
2: **for** $r = 0, 1, \ldots, R - 1$ **do**
3:     **for** $i = 1, \ldots, n$ in parallel **do**
4:         $\boldsymbol{x}_i^{(r+\frac{1}{2})} \approx \arg\min_{\boldsymbol{x}} F_{i,r}(\boldsymbol{x})$ where $F_{i,r}(\boldsymbol{x}) := f_i(\boldsymbol{x}) + \langle \boldsymbol{h}_i^{(r)}, \boldsymbol{x} \rangle + \frac{\lambda}{2}\|\boldsymbol{x} - \boldsymbol{x}_i^{(r)}\|^2$.
5:         $\boldsymbol{x}_i^{(r+1)} = \text{MultiGossip}\left(\boldsymbol{x}_i^{(r+\frac{1}{2})}, \boldsymbol{W}, M, i\right)$.
6:         $\boldsymbol{h}_i^{(r+1)} = \text{MultiGossip}\left(\boldsymbol{h}_i^{(r)} + \nabla f_i(\boldsymbol{x}_i^{(r+1)}), \boldsymbol{W}, M, i\right) - \nabla f_i(\boldsymbol{x}_i^{(r+1)})$.
7:     **end for**
8: **end for**

---

---

**Algorithm 2** Multiple Gossip Averaging

---

1: **function** MultiGossip($\{\boldsymbol{a}_i\}_{i=1}^n, \boldsymbol{W}, M, i$)
2:     $\boldsymbol{a}_i^{(0)} = \boldsymbol{a}_i$ for all $i$.
3:     **for** $m = 0, \ldots, M - 1$ **do**
4:         $\boldsymbol{a}_i^{(m+1)} = \sum_{j=1}^n W_{ij}\boldsymbol{a}_j^{(r)}$.
5:     **end for**
6:     **return** $\boldsymbol{a}_i^{(M)}$
7: **end function**

---

convergence analysis. PDO contains SONATA (Sun et al., 2022) as a special instance when the proximal subproblem is solved exactly. The proofs are deferred to Sec. D and G.2.

In decentralized optimization, each node would like to minimize $f$, but node $i$ can access only the local loss function $f_i$. For PDO, each node approximates $f$ as follows:

$$f(\boldsymbol{x}) = f_i(\boldsymbol{x}) + h_i(\boldsymbol{x}) \tag{6}$$
$$\approx f_i(\boldsymbol{x}) + \langle \nabla h_i(\boldsymbol{x}_i^{(r)}), \boldsymbol{x} \rangle + \frac{\lambda}{2}\left\|\boldsymbol{x} - \boldsymbol{x}_i^{(r)}\right\|^2,$$

where $\lambda \geq 0$ is the hyperparameter and we approximate $h_i(\boldsymbol{x})$ by linear approximation. Furthermore, PDO cannot compute $\nabla h_i(\boldsymbol{x}_i^{(r)})$ since node $i$ can exchange parameters only with its neighbors. Thus, PDO estimates $\nabla h_i(\boldsymbol{x}_i^{(r)})$ by gradient tracking. We show the pseudo-code in Alg. 1. In line 4, PDO solves the subproblem associated with Eq. (6), and in lines 5 and 6, PDO estimates $\frac{1}{n}\sum_{i=1}^n \boldsymbol{x}_i^{(r+\frac{1}{2})}$ and $\nabla h_i(\boldsymbol{x}_i^{(r+1)})$ by gossip averaging and gradient tracking.

**Arbitrary Approximate Averaging Operators:** While Alg. 1 is demonstrated using Alg. 2 as a subroutine, it is important to emphasize that the PDO framework is not limited to this specific mechanism. Instead, PDO can be combined with any arbitrary averaging operator that achieves a sufficient level of accuracy.

For instance, it is sufficient to use a single gossip averaging step (i.e., $M = 1$) if the spectral gap parameter $\rho$ satisfies

the condition

$$\rho \leq \frac{\delta}{6L}, \tag{7}$$

as we prove in Lemma 11 in the appendix. This flexibility extends to other averaging mechanisms, provided they meet the same error guarantees.

In many practical scenarios, the choice $M = 1$ may not suffice, as it requires the network topology to be sufficiently dense to ensure adequate communication accuracy. When the network is sparse, a common approach is to perform multiple iterations of gossip averaging (i.e., Alg. 2) to improve the averaging accuracy. Specifically, if gossip averaging is performed $M$ times, it holds that:

$$\sum_{i=1}^n \|\tilde{\boldsymbol{x}}_i - \bar{\boldsymbol{x}}\|^2 \leq \rho^{2M}\sum_{i=1}^n \|\boldsymbol{x}_i - \bar{\boldsymbol{x}}\|^2,$$

where $\{\boldsymbol{x}_i\}_{i=1}^n$ and $\{\tilde{\boldsymbol{x}}_i\}_{i=1}^n$ are the input and output of Alg. 2, and $\bar{\boldsymbol{x}} := \frac{1}{n}\sum_{i=1}^n \boldsymbol{x}_i$. Thus, if we perform gossip averaging more than $M = \frac{1}{1-\rho}\log(\frac{6L}{\delta})$ times, $\rho^M$ is less than or equal to $\frac{\delta}{6L}$, and then PDO works for arbitrary $\rho$.

### 4.1. Convergence Analysis

We now present the convergence analysis of PDO. The proofs are deferred to Sec. D and G.2.

**Theorem 1.** *Consider Alg. 1. Suppose that Assumptions 1, 2, and 4 hold, and $\boldsymbol{h}_i^{(0)} = \nabla h_i(\boldsymbol{x}_i^{(0)})$, $\boldsymbol{x}_i^{(0)} = \bar{\boldsymbol{x}}^{(0)}$, and*

$$\sum_{i=1}^n \left\|\nabla F_{i,r}(\boldsymbol{x}_i^{(r+\frac{1}{2})})\right\|^2$$
$$\leq \frac{\delta(4\delta + \mu)}{4(r+1)(r+2)}\sum_{i=1}^n \left\|\boldsymbol{x}_i^{(r+\frac{1}{2})} - \boldsymbol{x}_i^{(r)}\right\|^2. \tag{8}$$

***Strongly-convex Case:*** *Suppose that $\mu > 0$. When $\lambda = 4\delta$ and $M \geq \frac{1}{1-\rho}\log(\frac{6L}{\delta})$, it holds that $\frac{1}{W_R}\sum_{r=0}^{R-1} w^{(r)}\left(f(\bar{\boldsymbol{x}}^{(r+1)}) - f(\boldsymbol{x}^\star)\right) \leq \epsilon$ after $R = \mathcal{O}(\frac{\mu+\delta}{\mu}\log(\frac{\mu\|\bar{\boldsymbol{x}}^{(0)} - \boldsymbol{x}^\star\|^2}{\epsilon}))$ rounds where $w^{(r)} := (1 + \frac{\mu}{4\delta})^r$*

and $W_R := \sum_{r=0}^{R-1} w^{(r)}$. *Thus, it requires at most*

$$\mathcal{O}\left(\frac{\mu + \delta}{\mu(1-\rho)} \log\left(\frac{L}{\delta}\right) \log\left(\frac{\mu\|\bar{\boldsymbol{x}}^{(0)} - \boldsymbol{x}^\star\|^2}{\epsilon}\right)\right)$$

*communication where* $\bar{\boldsymbol{x}}^{(r)} := \frac{1}{n} \sum_{i=1}^{n} \boldsymbol{x}_i^{(r)}$.

**Convex Case:** *Suppose that* $\mu = 0$. *When* $\lambda = 4\delta$ *and* $M \geq \frac{1}{1-\rho} \log(\frac{6L}{\delta})$, *it holds that* $\frac{1}{R} \sum_{r=0}^{R-1} f(\bar{\boldsymbol{x}}^{(r)}) - f(\boldsymbol{x}^\star) \leq \epsilon$ *after* $R = \mathcal{O}(\frac{\delta\|\bar{\boldsymbol{x}}^{(0)} - \boldsymbol{x}^\star\|^2}{\epsilon})$ *rounds. Thus, it requires at most*

$$\mathcal{O}\left(\frac{\delta\|\bar{\boldsymbol{x}}^{(0)} - \boldsymbol{x}^\star\|^2}{(1-\rho)\epsilon} \log\left(\frac{L}{\delta}\right)\right)$$

*communication.*

Any optimization algorithm can be used to approximately solve the subproblem so that Eq. (8) is satisfied. For instance, if we use Nesterov's Accelerated Gradient Descent (Nesterov, 2018), we can achieve the following result.

**Theorem 2.** *Consider Alg. 1. Suppose that Assumptions 1, 2, and 4 hold, and we use the same initial parameters, $\lambda$, and $M$ as in Theorem 1. Then, if we use Nesterov's Accelerated Gradient Descent with initial parameter* $\boldsymbol{x}_i^{(r)}$ *to approximately solve the subproblem in line 4, each node requires at most*

$$\mathcal{O}\left(\sqrt{\frac{L}{\mu + \delta}} \log\left(\frac{L^2(r+2)^2}{\delta(\delta + \mu)}\right)\right)$$

*iterations to satisfy Eq. (8).*

### 4.2. Discussion

**New Analysis with Inexact Subproblem Solution:** The framework of PDO was initially introduced by Li et al. (2020) and Sun et al. (2022), and Sun et al. (2022) proposed SONATA. However, they assumed that the subproblem was exactly solved, which is infeasible in general. Our new theorem significantly relaxes this condition and shows that PDO can achieve the same communication complexity without solving the subproblem exactly. To clarify this difference from Sun et al. (2022), we refer to PDO with the inexact subproblem solution as **Inexact-PDO**.

**Reducing Dependence on $\delta_{\max}$ to $\delta$:** Sun et al. (2022) showed that SONATA can achieve $\tilde{\mathcal{O}}(\frac{\delta_{\max}}{\mu(1-\rho)} \log(\frac{1}{\epsilon}))$ communication complexity in the strongly-convex case. However, Theorem 1 improves it to $\tilde{\mathcal{O}}(\frac{\delta}{\mu(1-\rho)} \log(\frac{1}{\epsilon}))$. This improvement indicates a more favorable dependence on the functional dissimilarity, which can lead to significantly lower communication costs, as shown in Remark 1.

**New Analysis in Convex Case:** SONATA was analyzed only in the strongly-convex case (Sun et al., 2022). Our theorem is the first to provide the analysis in both convex and strongly-convex cases.

**Initial Values:** Computing the initial value of $\boldsymbol{h}_i$ requires $d_{\mathcal{G}}$ communications where $d_{\mathcal{G}}$ is the diameter of the underlying network. However, this overhead is negligible compared to the total communication cost. If the initial values are instead set to zero, additional terms may appear in the communication costs (see Koloskova et al. (2021)).

## 5. Stabilized Proximal Decentralized Optimization Method

Theorem 1 shows that PDO can achieve the best communication complexity among the existing non-accelerated decentralized methods. However, Eq. (8) implies that PDO needs to solve the subproblem more accurately as the number of rounds increases, which involves the suboptimal computational complexity. In this section, we propose **Stabilized Proximal Decentralized Optimization (SPDO)**, which can

---

**Algorithm 3** Stabilized Proximal Decentralized Optimization Method (SPDO)

1: **Input:** The number of rounds $R$, and hyperparameters $M$ and $\lambda$.
2: **for** $r = 0, 1, \ldots, R-1$ **do**
3:      **for** $i = 1, \ldots, n$ in parallel **do**
4:          $\boldsymbol{x}_i^{(r+1)} \approx \arg\min_{\boldsymbol{x} \in \mathbb{R}^d} F_{i,r}(\boldsymbol{x})$ where $F_{i,r}(\boldsymbol{x}) := f_i(\boldsymbol{x}) + \langle \boldsymbol{h}_i^{(r)}, \boldsymbol{x}\rangle + \frac{\lambda}{2}\|\boldsymbol{x} - \boldsymbol{v}_i^{(r)}\|^2$.
5:          $\boldsymbol{v}_i^{(r+\frac{1}{2})} = \arg\min_{\boldsymbol{v} \in \mathbb{R}^d} \left\{ \langle \nabla f_i(\boldsymbol{x}_i^{(r+1)}) + \boldsymbol{h}_i^{(r)}, \boldsymbol{v}\rangle + \frac{\mu}{2}\|\boldsymbol{v} - \boldsymbol{x}_i^{(r+1)}\|^2 + \frac{\lambda}{2}\|\boldsymbol{v} - \boldsymbol{v}_i^{(r)}\|^2 \right\}$.[a]
6:          $\boldsymbol{v}_i^{(r+1)} = \text{MULTIGOSSIP}\left(\boldsymbol{v}_i^{(r+\frac{1}{2})}, \boldsymbol{W}, M, i\right)$.
7:          $\boldsymbol{h}_i^{(r+1)} = \text{MULTIGOSSIP}\left(\boldsymbol{h}_i^{(r)} + \nabla f_i(\boldsymbol{v}_i^{(r+1)}), \boldsymbol{W}, M, i\right) - \nabla f_i(\boldsymbol{v}_i^{(r+1)})$.
8:      **end for**
9: **end for**

---

[a]This update rule can be rewritten as $\boldsymbol{v}_i^{(r+\frac{1}{2})} = \frac{1}{\mu+\lambda}\left(\mu\boldsymbol{x}_i^{(r+1)} + \lambda\boldsymbol{v}_i^{(r)} - \nabla f_i(\boldsymbol{x}_i^{(r+1)}) - \boldsymbol{h}_i^{(r)}\right)$.

achieve the same communication complexity while enjoying a better computational complexity than PDO.

For simplicity, we consider that case when $\mu = 0$ and $M = 1$. To relax the condition for an approximate subproblem solution, we use the idea of Hybrid Projection Proximal-point Method (Solodov & Svaiter, 1999). Straightforwardly adapting Hybrid Projection-proximal Point Method to PDO yields the following update rules:[1]

$$x_i^{(r+1)} \approx \arg\min_{x \in \mathbb{R}^d} \left\{ f_i(x) + \langle h_i^{(r)}, x \rangle + \frac{\lambda}{2} \|x - v_i^{(r)}\|^2 \right\},$$

$$v_i^{(r+\frac{1}{2})} = \arg\min_{v \in \mathbb{R}^d} \left\{ \langle \nabla f_i(x_i^{(r+1)}), v \rangle + \frac{\lambda}{2} \|v - v_i^{(r)}\|^2 \right\},$$

$$v_i^{(r+1)} = \sum_{j=1}^n W_{ij} v_j^{(r+\frac{1}{2})},$$

$$h_i^{(r+1)} = \sum_{j=1}^n W_{ij} \left( h_j^{(r)} + \nabla f_j(v_j^{(r+1)}) \right) - \nabla f_i(v_i^{(r+1)}).$$

However, we found that the above update rules do not work. In the above update rule, $v_i^{(r+\frac{1}{2})} \neq x_i^{(r+1)}$ even though the subproblem is solved exactly. That is, the above update rules do not recover PDO when the subproblem is exactly solved. This discrepancy can hinder the convergence of the parameters to the optimal solution. Intuitively, when we solve the subproblem exactly, $v_i^{(r+\frac{1}{2})}$ should equal to $x_i^{(r+1)}$ since the above technique is not necessary. Motivated by this intuition, we modify the update rule of $v_i^{(r+\frac{1}{2})}$ as in Alg. 3, in which the modification is highlighted in blue. We refer to this method as **Stabilized Proximal Decentralized Optimization (SPDO)**. As a special instance of SPDO, we recover PDO. When the subproblem is exactly solved, i.e., $\nabla F_{i,r}(x_i^{(r+1)}) = 0$, it holds

$$\nabla f_i(x_i^{(r+1)}) + h_i^{(r)} + \lambda \left( x_i^{(r+1)} - v_i^{(r)} \right) = 0,$$

and with this choice of $h_i^{(r)} = \lambda(v_i^{(r)} - x_i^{(r+1)}) - \nabla f_i(x_i^{(r+1)})$, Alg. 3 recovers PDO.

### 5.1. Convergence Analysis

In this section, we show the convergence analysis of SPDO. The proofs are deferred to Sec. E and G.3.

**Theorem 3.** *Consider Alg. 3. Suppose that Assumptions 1, 2, and 4 hold. Then, suppose that $h_i^{(0)} = \nabla h_i(v_i^{(0)})$, $v_i^{(0)} = \bar{v}^{(0)}$, and*

$$\sum_{i=1}^n \left\| \nabla F_{i,r}(x_i^{(r+1)}) \right\|^2 \leq \frac{\lambda^2}{10} \sum_{i=1}^n \left\| v_i^{(r)} - x_i^{(r+1)} \right\|^2. \quad (9)$$

[1]Note that $\arg\min_{v \in \mathbb{R}^d}(\cdot)$ has a closed-form solution.

**Strongly-convex Case:** *Suppose that $\mu > 0$. When $\lambda = 20\delta$ and $M \geq \frac{1}{1-\rho} \log(\frac{5L}{\delta})$, it holds that $\frac{1}{W_R} \sum_{r=0}^{R-1} w^{(r)} \left( f(\bar{x}^{(r+1)}) - f(x^\star) \right) \leq \epsilon$ after $R = \mathcal{O}(\frac{\mu+\delta}{\mu} \log(\frac{\mu\|\bar{v}^{(0)} - x^\star\|^2}{\epsilon}))$ rounds where $w^{(r)} := (1 + \frac{\mu}{20\delta})^r$ and $W_R := \sum_{r=0}^{R-1} w^{(r)}$. Thus, it requires at most*

$$\mathcal{O} \left( \frac{\mu + \delta}{\mu(1 - \rho)} \log \left( \frac{L}{\delta} \right) \log \left( \frac{\mu\|\bar{v}^{(0)} - x^\star\|^2}{\epsilon} \right) \right)$$

*communication where $\bar{v}^{(r)} := \frac{1}{n} \sum_{i=1}^n v_i^{(r)}$.*

**Convex Case:** *Suppose that $\mu = 0$. When $\lambda = 20\delta$ and $M \geq \frac{1}{1-\rho} \log(\frac{5L}{\delta})$, it holds that $\frac{1}{R} \sum_{r=1}^R f(\bar{x}^{(r)}) - f(x^\star) \leq \epsilon$ after $R = \mathcal{O}(\frac{\delta\|\bar{v}^{(0)} - x^\star\|^2}{\epsilon})$ rounds. Thus, it requires at most*

$$\mathcal{O} \left( \frac{\delta\|\bar{v}^{(0)} - x^\star\|^2}{(1 - \rho)\epsilon} \log \left( \frac{L}{\delta} \right) \right)$$

*communication.*

**Theorem 4.** *Consider Alg. 3. Suppose that Assumptions 1, 2, and 4 hold, and we use the same initial parameters, $\lambda$, and $M$ as in Theorem 3. If we use the algorithm proposed in Remark 1 in Nesterov et al. (2018) with initial parameter $v_i^{(r)}$ to approximately solve the subproblem in line 4, each node requires at most $\mathcal{O}(\sqrt{\frac{L}{\delta}})$ iterations to satisfy Eq. (9).*

*Remark* 2. Suppose that the same assumptions hold as in Theorem 4. Then, if we use Nesterov's Accelerated Gradient Descent with initial parameter $v_i^{(r)}$ to approximately solve the subproblem in line 4, each node requires at most $\mathcal{O}(\sqrt{\frac{L}{\mu+\delta}} \log(\frac{L}{\delta}))$ iterations to satisfy Eq. (9).

**Discussion:** Eq. (8) shows that Inexact-PDO needs to solve the subproblem more accurately as the number of rounds increases, whereas Eq. (9) indicates that SPDO does not. Thus, as shown in Theorem 1, 2, 3, and 4, SPDO can attain the same communication complexity as PDO while enjoying a better computational complexity.

## 6. Accelerated Stabilized Proximal Decentralized Optimization Method

In this section, we propose **Accelerated-SPDO**, which can attain the best communication and computational complexities among the existing methods. We present the pseudo-code in Alg. 4. We use Monteiro and Svaiter acceleration (Monteiro & Svaiter, 2013) and accelerated gossip averaging (Liu & Morse, 2011) in Accelerated-SPDO.

### 6.1. Convergence Analysis

We now provide the convergence analysis of Accelerated-SPDO. The proofs are deferred to Sec. F and G.5.

---

**Algorithm 4** Accelerated Stabilized Proximal Decentralized Optimization Method (Accelerated-SPDO)

1: **Input:** The number of rounds $R$, and hyperparameters $\gamma$, $M$, and $\lambda$.
2: $A_0 = 0$ and $B_0 = 1$.
3: **for** $r = 0, 1, \ldots, R - 1$ **do**
4:   Find $a_{r+1} > 0$ that satisfies $\lambda = \frac{(A_r + a_{r+1})B_r}{a_{r+1}^2}$.
5:   $A_{r+1} = A_r + a_{r+1}$.
6:   $B_{r+1} = B_r + \mu a_{r+1}$.
7:   **for** $i = 1, \ldots, n$ in parallel **do**
8:    $\boldsymbol{y}_i^{(r)} = \frac{A_r \boldsymbol{x}_i^{(r)} + a_{r+1} \boldsymbol{v}_i^{(r)}}{A_{r+1}}$.
9:    **if** $r \geq 1$ **then**
10:     $\boldsymbol{h}_i^{(r)} = \text{FASTGOSSIP}\left(\boldsymbol{h}_i^{(r-1)} + \nabla f_i(\boldsymbol{y}_i^{(r)}), \boldsymbol{W}, M, \gamma, i\right) - \nabla f_i(\boldsymbol{y}_i^{(r)})$.
11:    **end if**
12:    $\boldsymbol{x}_i^{(r+\frac{1}{2})} \approx \arg\min_{\boldsymbol{x} \in \mathbb{R}^d} F_{i,r}(\boldsymbol{x})$ where $F_{i,r}(\boldsymbol{x}) := f_i(\boldsymbol{x}) + \langle \boldsymbol{h}_i^{(r)}, \boldsymbol{x}\rangle + \frac{\lambda}{2}\|\boldsymbol{x} - \boldsymbol{y}_i^{(r)}\|^2$.
13:    $\boldsymbol{v}_i^{(r+\frac{1}{2})} = \arg\min_{\boldsymbol{v} \in \mathbb{R}^d}\left\{ a_{r+1}\left(\langle \nabla f_i(\boldsymbol{x}_i^{(r+\frac{1}{2})}) + \boldsymbol{h}_i^{(r)}, \boldsymbol{v}\rangle + \frac{\mu}{2}\|\boldsymbol{v} - \boldsymbol{x}_i^{(r+\frac{1}{2})}\|^2\right) + \frac{B_r}{2}\|\boldsymbol{v} - \boldsymbol{v}_i^{(r)}\|^2\right\}^a$.
14:    $\boldsymbol{x}_i^{(r+1)} = \text{FASTGOSSIP}\left(\boldsymbol{x}_i^{(r+\frac{1}{2})}, \boldsymbol{W}, M, \gamma, i\right)$.
15:    $\boldsymbol{v}_i^{(r+1)} = \text{FASTGOSSIP}\left(\boldsymbol{v}_i^{(r+\frac{1}{2})}, \boldsymbol{W}, M, \gamma, i\right)$.
16:   **end for**
17: **end for**

---

[a]This update rule can be rewritten as $\boldsymbol{v}_i^{(r+\frac{1}{2})} = \frac{1}{B_{r+1}}\left(B_r \boldsymbol{v}_i^{(r)} + \mu a_{r+1}\boldsymbol{x}_i^{(r+\frac{1}{2})} - a_{r+1}\left(\nabla f_i(\boldsymbol{x}_i^{(r+\frac{1}{2})}) + \boldsymbol{h}_i^{(r)}\right)\right)$.

---

**Algorithm 5** Fast Gossip Averaging

1: **function** FASTGOSSIP($\{\boldsymbol{a}_i\}_{i=1}^n$, $\boldsymbol{W}$, $M$, $\gamma$, $i$)
2:   $\boldsymbol{a}_i^{(0)} = \boldsymbol{a}_i$ and $\boldsymbol{a}_i^{(-1)} = \boldsymbol{a}_i$ for all $i$.
3:   **for** $m = 0, \ldots, M - 1$ **do**
4:    $\boldsymbol{a}_i^{(m+1)} = (1 + \gamma)\sum_{j=1}^n W_{ij}\boldsymbol{a}_j^{(m)} - \gamma \boldsymbol{a}_i^{(m-1)}$.
5:   **end for**
6:   **return** $\boldsymbol{a}_i^{(M)}$.
7: **end function**

**Theorem 5.** *Consider Alg. 4. Suppose that Assumptions 1, 2, and 4 hold. Then, suppose that $\boldsymbol{h}_i^{(0)} = \nabla h_i(\boldsymbol{v}_i^{(0)})$, $\boldsymbol{x}_i^{(0)} = \boldsymbol{v}_i^{(0)} = \bar{\boldsymbol{v}}^{(0)}$, and*

$$\sum_{i=1}^n \left\|\nabla F_{i,r}(\boldsymbol{x}_i^{(r+\frac{1}{2})})\right\|^2 \leq \frac{\lambda^2}{352}\sum_{i=1}^n \left\|\boldsymbol{y}_i^{(r)} - \boldsymbol{x}_i^{(r+\frac{1}{2})}\right\|^2. \quad (10)$$

**Strongly-convex Case:** *Suppose that $\mu > 0$. When $\lambda = 96\delta$, $M \geq \frac{4}{\sqrt{1-\rho}}\log(\frac{18L^2(192\delta + \mu)}{\mu\delta^2})$, and $\gamma = \frac{1 - \sqrt{1-\rho^2}}{1 + \sqrt{1+\rho^2}}$, it holds that $f(\bar{\boldsymbol{x}}^{(R)}) - f(\boldsymbol{x}^\star) \leq \epsilon$ holds after $R = \mathcal{O}(\sqrt{\frac{\mu + \delta}{\mu}}\log(1 + \sqrt{\frac{\min\{\mu, \delta\}\|\bar{\boldsymbol{x}}^{(0)} - \boldsymbol{x}^\star\|^2}{\epsilon}}))$ rounds. Thus, it requires at most*

$$\mathcal{O}\left(\sqrt{\frac{\mu + \delta}{\mu(1 - \rho)}}\log\left(\frac{L^2(\mu + \delta)}{\mu\delta^2}\right)\right.$$
$$\left.\log\left(1 + \sqrt{\frac{\min\{\mu, \delta\}\|\bar{\boldsymbol{x}}^{(0)} - \boldsymbol{x}^\star\|^2}{\epsilon}}\right)\right)$$

*communication where $\bar{\boldsymbol{x}}^{(r)} := \frac{1}{n}\sum_{i=1}^n \boldsymbol{x}_i^{(r)}$.*

**Convex Case:** *Suppose that $\mu = 0$. When $\lambda = 208\delta$, $M \geq \frac{3}{2\sqrt{1-\rho}}\log(\max\{\frac{12L}{\delta}, \frac{20384\delta\|\bar{\boldsymbol{x}}^{(0)} - \boldsymbol{x}^\star\|^2}{\epsilon}\})$, and $\gamma = \frac{1 - \sqrt{1-\rho^2}}{1 + \sqrt{1+\rho^2}}$, it holds that $f(\bar{\boldsymbol{x}}^{(R)}) - f(\boldsymbol{x}^\star) \leq \epsilon$ holds after $R = \mathcal{O}(\sqrt{\frac{\delta\|\bar{\boldsymbol{x}}^{(0)} - \boldsymbol{x}^\star\|^2}{\epsilon}})$ rounds. Thus, it requires at most*

$$\mathcal{O}\left(\sqrt{\frac{\delta\left\|\bar{\boldsymbol{x}}^{(0)} - \boldsymbol{x}^\star\right\|^2}{(1 - \rho)\epsilon}}\log\left(\max\left\{\frac{L}{\delta}, \frac{\delta\|\bar{\boldsymbol{x}}^{(0)} - \boldsymbol{x}^\star\|^2}{\epsilon}\right\}\right)\right)$$

*communication.*

**Theorem 6.** *Consider Alg. 4. Suppose that Assumptions 1, 2, and 4 hold, and we use the same initial parameters, $\gamma$, $\lambda$, and $M$ as in Theorem 5. Then, if we use the algorithm proposed in Remark 1 in Nesterov et al. (2018) with initial parameter $\boldsymbol{y}_i^{(r)}$ to approximately solve the subproblem in line 12, it requires at most $\mathcal{O}(\sqrt{\frac{L}{\delta}})$ iterations to satisfy Eq. (10).*

*Remark 3.* Suppose that the same assumptions hold as in Theorem 6. Then, if we use Nesterov's Accelerated Gradient Descent with initial parameter $\boldsymbol{y}_i^{(r)}$ to approximately solve the subproblem in line 12, it requires at most $\mathcal{O}(\sqrt{\frac{L}{\mu + \delta}}\log(\frac{L}{\delta}))$ iterations to satisfy Eq. (10).

### 6.2. Discussion

In this section, we compare Accelerated-SPDO with the existing state-of-the-art method, (Inexact) Accelerated

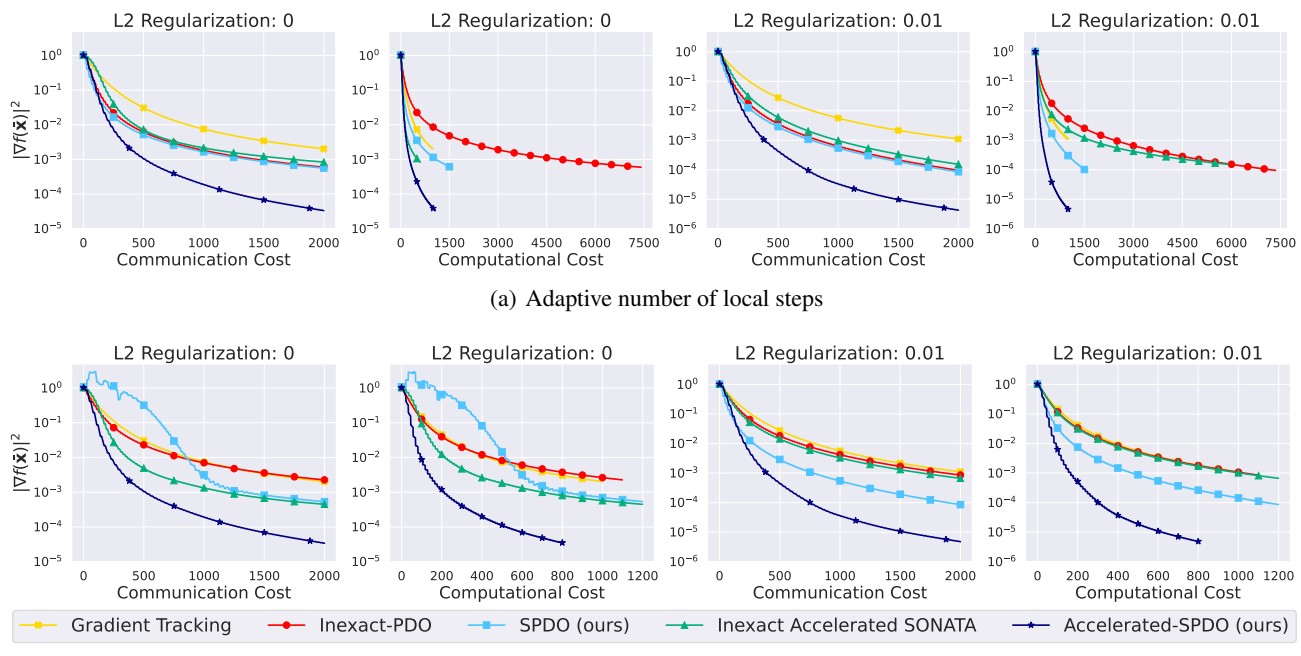

(a) Adaptive number of local steps

(b) Fixed number of local steps

*Figure 1.* Convergence of the gradient norm with $\alpha = 0.1$. In (a), we ran gradient descent until the condition for an approximate subproblem solution was satisfied for all methods except for Gradient Tracking. For Gradient Tracking, we ran gradient descent for 10 times. In (b), we ran gradient descent for 10 times to approximately solve the subproblem. See Sec. H for a more detailed setting.

SONATA (Tian et al., 2022). Tian et al. (2022) assumed that $f_i$ is strongly convex and $\mu \leq \delta_{\max}$. Thus, in the following, we discuss the case when $f_i$ is strongly-convex, $\mu \leq \delta$, and $\mu \leq \delta_{\max}$. See Table 1 for the results.

**Computational Complexity:** Accelerated-SPDO can achieve the best computational complexity among the existing decentralized optimization methods. Accelerated SONATA needs to solve the subproblem exactly, which is infeasible in general. Inexact Accelerated SONATA solves the subproblem approximately and can achieve $\tilde{\mathcal{O}}((\log(\frac{1}{\epsilon}))^2)$ computational complexity. Compared with these results, Accelerated-SPDO can achieve $\tilde{\mathcal{O}}(\log(\frac{1}{\epsilon}))$ computational complexity. This is because Inexact Accelerated SONATA requires a more precise subproblem solution as the number of rounds increases, while Accelerated-SPDO does not.

**Communication Complexity:** Accelerated-SPDO can achieve the best communication complexity among the existing methods. In the worst case, $\delta$ can be equal to $\delta_{\max}$, and Accelerated-SPDO attains the same communication complexity as (Inexact) Accelerated SONATA up to logarithmic factors. However, $\delta$ can be significantly smaller than $\delta_{\max}$. In this case, Accelerated-SPDO can achieve better communication complexity.

**Acceleration in Convex Case:** (Inexact) Accelerated SONATA has been shown to accelerate the rate only in

the strongly-convex case, whereas Accelerated-SPDO can accelerate it in both convex and strongly-convex cases.

### 6.3. Discussion on Federated Learning

In this section, we discuss the relationship between our PDO-based methods and the existing methods developed for federated learning. Decentralized optimization generalizes federated learning, and when the underlying network is a complete graph, decentralized learning becomes equivalent to federated learning. PDO recovers DANE (Shamir et al., 2014) when the graph is fully-connected (i.e., $W_{ij} = \frac{1}{n}$). SPDO and Accelerated-SPDO contain S-DANE and Accelerated-S-DANE (Jiang et al., 2024b) as a special instance, respectively. Note that when the graph is fully-connected, $\rho = 0$ and multiple gossip averaging is not necessary. Furthermore, since $\sum_{i=1}^{n} \boldsymbol{h}_i^{(r)} = \boldsymbol{0}$, the technique we proposed in Sec. 5 is also not required (see the discussion in Sec. 5). However, in decentralized optimization, the network is generally not fully connected; hence, these techniques become essential to ensure convergence.

## 7. Numerical Evaluation

**Experimental Setup:** We used MNIST (Lecun et al., 1998) and logistic loss with L2 regularization. We set the coefficient of L2 regularization to 0 and 0.01 and the number of nodes $n$ to 25. We used the ring as the underlying

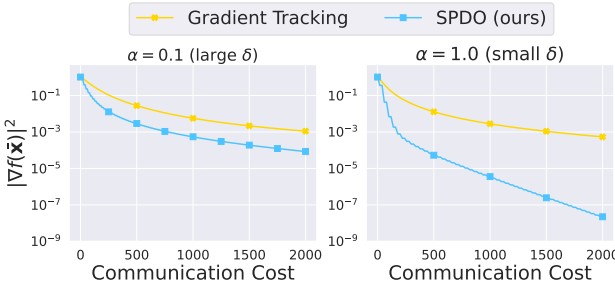

*Figure 2.* Convergence of the gradient norm with different $\alpha$. We set the coefficient of L2 regularization to $0.01$. For SPDO, we approximately solve the subproblem as in Fig. 1(a).

network topology and distributed the data to nodes by using Dirichlet distribution with parameter $\alpha > 0$ (Hsu et al., 2019). For Inexact-PDO, Inexact Accelerated SONATA, SPDO, and Accelerated-SPDO, we used gradient descent with stepsize $\eta$ to solve the subproblem approximately. We set the number of communications to 2000 for all methods and tuned other hyperparameters, e.g., $\lambda$, $M$, and $\eta$, to minimize the norm of the last gradient for each method. See Sec. H for a more detailed setting.

**Results:** Fig. 1(a) indicates that Accelerated-SPDO can achieve the best communication and computational complexities. SPDO achieved the same communication complexity as Inexact-PDO while enjoying a better computation complexity. These results are consistent with Theorem 1 and 3. In Fig. 1(b), we ran gradient descent for 10 times to solve the subproblem approximately. Accelerated-SPDO achieved the best communication and computational complexities as in Fig. 1(a). The results indicate that SPDO outperforms Inexact-PDO in both communication and computational complexities. This is because the condition for an approximate subproblem solution Eq. (8) is not satisfied when the number of rounds is sufficiently large, which makes Inexact-PDO need to communicate more.

**Effect of Similarity $\delta$:** Figure 2 compares the convergence behavior with different $\alpha$. As $\alpha$ increases, nodes come to have similar datasets, i.e., $\delta$ comes to be small. The convergence behaviors of Gradient Tracking were almost the same in both cases, whereas SPDO can converge faster as $\alpha$ increases. This observation was consistent with the comparison in Table 1. Gradient Tracking is not affected by data heterogeneity, while it still requires a certain amount of communication even though nodes have very similar functions. In contrast, SPDO can exploit the similarity of local functions and reduce the communication complexity when nods have similar functions.

## 8. Conclusion

In this work, we develop a communication and computation-efficient decentralized optimization method. To achieve this,

we first introduce the proximal decentralized optimization (PDO) methods and analyze the communication and computational complexities. We then propose SPDO, which can achieve the same communication complexity as PDO while enjoying a better computational complexity. Finally, we propose Accelerated-SPDO, which can achieve the best communication and computational complexities among the existing methods. Throughout the numerical experiments, we confirmed that our proposed method can outperform the existing methods as shown by our analysis.

## Impact Statement

This paper studies communication and computation-efficient decentralized optimization methods. Our work mainly focuses on theoretical aspects of decentralized optimization, and there are no specific societal consequences that must be highlighted here.

## Acknowledgments

This work was done while YT visited the CISPA Helmholtz Center for Information Security. YT was supported by JSPS KAKENHI Grant Number 23KJ1336.

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

# A. Comparison of Communication and Computational Complexities in Convex Case

*Table 2.* Summary of communication communication and computational complexities in the convex setting.

| Algorithm | Reference | # Communication | # Computation | Assumptions |
|---|---|---|---|---|
| Gradient Tracking | Koloskova et al. (2021) | $\mathcal{O}\left(\frac{Ld_0^2}{(1-\rho)^2\epsilon}\right)^{(a)}$ | $\mathcal{O}\left(\frac{Ld_0^2}{(1-\rho)^2\epsilon}\right)^{(a)}$ | 1, 2, 4 |
| Exact-PDO (SONATA) | Theorems 1, 2$^{(b)}$ | $\mathcal{O}\left(\frac{\delta d_0^2}{(1-\rho)\epsilon}\log\left(\frac{L}{\delta}\right)\right)$ | n.a.$^{(c)}$ | 1, 2, 4 |
| **Inexact-PDO** | Theorems 1, 2 | $\mathcal{O}\left(\frac{\delta d_0^2}{(1-\rho)\epsilon}\log\left(\frac{L}{\delta}\right)\right)$ | $\mathcal{O}\left(\frac{\sqrt{\delta L}d_0^2}{\epsilon}\log(\frac{1}{\epsilon})\right)^{(b)}$ | 1, 2, 4 |
| **Stabilized-PDO [new]** | Theorems 3, 4 | $\mathcal{O}\left(\frac{\delta d_0^2}{(1-\rho)\epsilon}\log\left(\frac{L}{\delta}\right)\right)$ | $\mathcal{O}\left(\frac{\sqrt{\delta L}d_0^2}{\epsilon}\right)^{(b)}$ | 1, 2, 4 |
| Accelerated SONATA | Tian et al. (2022) | - | n.a.$^{(c)}$ | - |
| Inexact Accelerated SONATA | Tian et al. (2022) | - | - | - |
| **Accelerated Stabilized-PDO [new]** | Theorems 5, 6 | $\mathcal{O}\left(\sqrt{\frac{\delta d_0^2}{(1-\rho)\epsilon}}\log\left(\max\left\{\frac{L}{\delta},\frac{\delta d_0^2}{\epsilon}\right\}\right)\right)$ | $\mathcal{O}\left(\sqrt{\frac{Ld_0^2}{\epsilon}}\right)$ | 1, 2, 4 |

(a) It holds that $\delta \leq L$, and $\delta \ll L$ holds when nodes have similar datasets.

(b) Sun et al. (2022) did not analyze the communication complexity of SONATA in the convex case, while our new theorem, Theorem 1, is the first to analyze it.

(c) "n.a." represents that we need to solve the subproblem exactly.

(d) $\delta$ is smaller than $\delta_{\max}$ and can be at most $\sqrt{n}$ times smaller than $\delta_{\max}$.

## B. Comparison with Lower Bound

In this section, we compare the communication and computational complexities of Accelerated SPDO and the lower bound.

**Communication Complexity:** Tian et al. (2022) showed that lower bound requires at least

$$\Omega\left(\sqrt{\frac{\delta}{\mu(1-\rho)}}\log\left(\frac{\mu|\boldsymbol{x}^\star|^2}{\epsilon}\right)\right)$$

communication to achieve $f(x) - f(x^\star) \le \epsilon$. Our Accelerated SPDO can achieve the following communication complexity:

$$\mathcal{O}\left(\sqrt{\frac{1+\frac{\delta}{\mu}}{1-\rho}}\log\left(\frac{L^2(\mu+\delta)}{\mu\delta^2}\right)\log\left(1+\sqrt{\frac{\min\{\mu,\delta\}\|\boldsymbol{x}^{(0)}-\boldsymbol{x}^\star\|^2}{\epsilon}}\right)\right)$$

Thus, when $\delta \ge \mu$, Accelerated SPDO is optimal up to the logarithmic factor.

**Computational Complexity:** For non-distributed cases, any first-order algorithm requires at least

$$\Omega\left(\sqrt{\frac{L}{\mu}}\log\left(\frac{\mu\|\boldsymbol{x}^{(0)}-\boldsymbol{x}^\star\|^2}{\epsilon}\right)\right)$$

gradient-oracles to satisfy $f(x) - f^\star \le \epsilon$ (See Theorem 2.1.13 in Nesterov (2018)). When $\delta \ge \mu$, Accelerated SPDO can achieve the following computational complexity:

$$\mathcal{O}\left(\sqrt{\frac{L}{\mu}}\log\left(1+\sqrt{\frac{\mu\|\boldsymbol{x}^{(0)}-\boldsymbol{x}^\star\|^2}{\epsilon}}\right)\right),$$

when we use the algorithm shown in Remark 1 in Nesterov et al. (2018) (see Theorem 6). Thus, the computational complexity of Accelerated SPDO is optimal up to constant factors.

## C. Technical Lemmas

### C.1. Basic Lemmas

**Lemma 2.** *For any $\boldsymbol{a}, \boldsymbol{b} \in \mathbb{R}^d$, it holds that for any $\alpha > 0$*

$$|\langle \boldsymbol{a}, \boldsymbol{b} \rangle| \leq \frac{1}{2\alpha} \|\boldsymbol{a}\|^2 + \frac{\alpha}{2} \|\boldsymbol{b}\|^2. \tag{11}$$

**Lemma 3.** *For any $\boldsymbol{a}, \boldsymbol{b} \in \mathbb{R}^d$ and $\gamma > 0$, it holds that*

$$\|\boldsymbol{a} + \boldsymbol{b}\|^2 \leq (1 + \gamma) \|\boldsymbol{a}\|^2 + \left(1 + \gamma^{-1}\right) \|\boldsymbol{b}\|^2. \tag{12}$$

**Lemma 4.** *For any $\boldsymbol{a}_1, \boldsymbol{a}_2, \ldots, \boldsymbol{a}_n \in \mathbb{R}^d$, it holds that*

$$\|\bar{\boldsymbol{a}}\|^2 \leq \frac{1}{n} \sum_{i=1}^{n} \|\boldsymbol{a}_i\|^2, \tag{13}$$

*where $\bar{\boldsymbol{a}} := \frac{1}{n} \sum_{i=1}^{n} \boldsymbol{a}_i$.*

**Lemma 5.** *For any $\boldsymbol{a}_1, \ldots, \boldsymbol{a}_n \in \mathbb{R}^2$ and $\boldsymbol{b} \in \mathbb{R}$, it holds that*

$$\frac{1}{n} \sum_{i=1}^{n} \|\boldsymbol{a}_i - \boldsymbol{b}\|^2 = \|\bar{\boldsymbol{a}} - \boldsymbol{b}\|^2 + \frac{1}{n} \sum_{i=1}^{n} \|\boldsymbol{a}_i - \bar{\boldsymbol{a}}\|^2. \tag{14}$$

*Proof.* See Lemma 14 in Jiang et al. (2024a). $\qquad \square$

### C.2. Useful Lemmas

**Lemma 6.** *Let $\boldsymbol{x}_1, \boldsymbol{x}_2, \ldots, \boldsymbol{x}_n$ be vectors in $\mathbb{R}^d$ and $\boldsymbol{W} \in \mathbb{R}^{n \times n}$ be a matrix that satisfies $\sum_{i=1}^{n} W_{ij} = 1$. Let $\tilde{\boldsymbol{x}}_i := \sum_{j=1}^{n} W_{ij} \boldsymbol{x}_j$. Then, it holds that*

$$\sum_{i=1}^{n} \boldsymbol{x}_i = \sum_{i=1}^{n} \tilde{\boldsymbol{x}}_i.$$

*Proof.* We have

$$\sum_{i=1}^{n} \tilde{\boldsymbol{x}}_i = \sum_{i=1}^{n} \sum_{j=1}^{n} W_{ij} \boldsymbol{x}_j = \sum_{j=1}^{n} \left( \sum_{i=1}^{n} W_{ij} \right) \boldsymbol{x}_j = \sum_{j=1}^{n} \boldsymbol{x}_j.$$

This concludes the statement. $\qquad \square$

**Lemma 7.** *For any matrix $\boldsymbol{X} \in \mathbb{R}^{n \times d}$ and any matrix $\boldsymbol{W} \in \mathbb{R}^{n \times n}$ that satisfies Assumption 4, it holds that*

$$\|(\boldsymbol{W} - \boldsymbol{I})\boldsymbol{X}\|_F \leq 2 \|\boldsymbol{X}\|_F. \tag{15}$$

*Proof.* We have

$$
\begin{aligned}
\|(\boldsymbol{W} - \boldsymbol{I})\boldsymbol{X}\|_F &\leq \left\| \left( \boldsymbol{W} - \frac{1}{n} \mathbf{1}_n \mathbf{1}_n^\top \right) \boldsymbol{X} \right\|_F + \left\| \left( \frac{1}{n} \mathbf{1}_n \mathbf{1}_n^\top - \boldsymbol{I} \right) \boldsymbol{X} \right\|_F \\
&\overset{(5)}{\leq} (\rho + 1) \left\| \left( \frac{1}{n} \mathbf{1}_n \mathbf{1}_n^\top - \boldsymbol{I} \right) \boldsymbol{X} \right\|_F \\
&\leq 2 \left\| \boldsymbol{I} - \frac{1}{n} \mathbf{1}_n \mathbf{1}_n^\top \right\|_{\text{op}} \|\boldsymbol{X}\|_F,
\end{aligned}
$$

where $\mathbf{1}_n$ is $n$-demential vector with all ones and we use $\rho \leq 1$ in the last inequality. Using $\|\boldsymbol{I} - \frac{1}{n} \mathbf{1}_n \mathbf{1}_n^\top\|_{\text{op}} \leq 1$, we obtain the desired result. $\qquad \square$

**Lemma 8.** *Let $f_i : \mathbb{R}^d \to \mathbb{R}$ be a function that satisfies Assumptions 2 and $\boldsymbol{W} \in \mathbb{R}^{n \times n}$ be a matrix that satisfies Assumption 4. Let $\{\{\boldsymbol{x}_i^{(r)}\}_{i=}^n\}_{r=0}^R$ be a set of vectors in $\mathbb{R}^d$ and define the update rule of $\{\boldsymbol{h}_i^{(r)}\}_{i=1}^n$ as follows:*

$$\boldsymbol{h}_i^{(r+1)} = \sum_{j=1}^n W_{ij} \left( \boldsymbol{h}_j^{(r)} + \nabla f_j(\boldsymbol{x}_j^{(r)}) \right) - \nabla f_i(\boldsymbol{x}_i^{(r)}).$$

*Then, when $\sum_{i=1}^n \boldsymbol{h}_i^{(0)} = \boldsymbol{0}$, it holds that for all $r$*

$$\sum_{i=1}^n \boldsymbol{h}_i^{(r)} = \boldsymbol{0}, \tag{16}$$

*and*

$$\mathcal{E}^{(r+1)} \le 4\rho^2 \mathcal{E}^{(r)} + 4\rho^2 \delta^2 \left\| \bar{\boldsymbol{x}}^{(r+1)} - \bar{\boldsymbol{x}}^{(r)} \right\|^2 + 32 L^2 \Xi_{\boldsymbol{x}}^{(r+1)}, \tag{17}$$

*where $\mathcal{E}^{(r)} := \frac{1}{n} \sum_{i=1}^n \|\boldsymbol{h}_i^{(r)} - \nabla h_i(\bar{\boldsymbol{x}}^{(r)})\|^2$ and $\Xi_{\boldsymbol{x}}^{(r+1)} := \frac{1}{n} \sum_{i=1}^n \|\boldsymbol{x}_i^{(r+1)} - \bar{\boldsymbol{x}}^{(r+1)}\|^2$.*

*Proof.* We have

$$\sum_{i=1}^n \boldsymbol{h}_i^{(r+1)} = \sum_{i=1}^n \left( \sum_{j=1}^n W_{ij} \left( \boldsymbol{h}_j^{(r)} + \nabla f_j(\boldsymbol{x}_j^{(r+1)}) \right) - \nabla f_i(\boldsymbol{x}_i^{(r+1)}) \right)$$

$$= \sum_{j=1}^n \left( \sum_{i=1}^n W_{ij} \right) \left( \boldsymbol{h}_j^{(r)} + \nabla f_j(\boldsymbol{x}_j^{(r+1)}) \right) - \sum_{i=1}^n \nabla f_i(\boldsymbol{x}_i^{(r+1)})$$

$$= \sum_{i=1}^n \boldsymbol{h}_i^{(r)}.$$

Since $\sum_{i=1}^n \boldsymbol{h}_i^{(0)} = \boldsymbol{0}$, it holds that $\sum_{i=1}^n \boldsymbol{h}_i^{(r)} = \boldsymbol{0}$ for any $r$. For the second statement, we have

$$\frac{1}{n} \sum_{i=1}^n \left\| \boldsymbol{h}_i^{(r+1)} - \nabla h_i(\bar{\boldsymbol{x}}^{(r+1)}) \right\|^2$$

$$\overset{(11)}{\le} \frac{2}{n} \sum_{i=1}^n \left\| \sum_{j=1}^n W_{ij} \left( \boldsymbol{h}_j^{(r)} + \nabla f_j(\bar{\boldsymbol{x}}^{(r+1)}) \right) - \nabla f(\bar{\boldsymbol{x}}^{(r+1)}) \right\|^2$$

$$+ \frac{2}{n} \sum_{i=1}^n \left\| \sum_{j=1}^n (W_{ij} - \mathbb{1}_{i=j}) \left( \nabla f_j(\boldsymbol{x}_j^{(r+1)}) - \nabla f_j(\bar{\boldsymbol{x}}^{(r+1)}) \right) \right\|^2$$

$$\le \frac{2}{n} \sum_{i=1}^n \left\| \sum_{j=1}^n W_{ij} \left( \boldsymbol{h}_j^{(r)} + \nabla f_j(\bar{\boldsymbol{x}}^{(r+1)}) \right) - \nabla f(\bar{\boldsymbol{x}}^{(r+1)}) \right\|^2 + \frac{32}{n} \sum_{i=1}^n \left\| \nabla f_i(\boldsymbol{x}_i^{(r+1)}) - \nabla f_i(\bar{\boldsymbol{x}}^{(r+1)}) \right\|^2$$

$$\le \frac{2}{n} \sum_{i=1}^n \left\| \sum_{j=1}^n W_{ij} \left( \boldsymbol{h}_j^{(r)} + \nabla f_j(\bar{\boldsymbol{x}}^{(r+1)}) \right) - \nabla f(\bar{\boldsymbol{x}}^{(r+1)}) \right\|^2 + \frac{32 L^2}{n} \sum_{i=1}^n \left\| \boldsymbol{x}_i^{(r+1)} - \bar{\boldsymbol{x}}^{(r+1)} \right\|^2,$$

where $\mathbb{1}_{i=j}$ is an indicator function. Using Eq. (16), we obtain

$$
\frac{1}{n} \sum_{i=1}^{n} \left\| \boldsymbol{h}_i^{(r+1)} - \nabla h_i(\bar{\boldsymbol{x}}^{(r+1)}) \right\|^2
$$

$$
\overset{(5)}{\leq} \frac{2\rho^2}{n} \sum_{i=1}^{n} \left\| \boldsymbol{h}_i^{(r)} - \nabla h_i(\bar{\boldsymbol{x}}^{(r+1)}) \right\|^2 + \frac{32L^2}{n} \sum_{i=1}^{n} \left\| \boldsymbol{x}_i^{(r+1)} - \bar{\boldsymbol{x}}^{(r+1)} \right\|^2
$$

$$
\overset{(12)}{\leq} \frac{4\rho^2}{n} \sum_{i=1}^{n} \left\| \boldsymbol{h}_i^{(r)} - \nabla h_i(\bar{\boldsymbol{x}}^{(r)}) \right\|^2 + \frac{4\rho^2}{n} \sum_{i=1}^{n} \left\| \nabla h_i(\bar{\boldsymbol{x}}^{(r+1)}) - \nabla h_i(\bar{\boldsymbol{x}}^{(r)}) \right\|^2 + \frac{32L^2}{n} \sum_{i=1}^{n} \left\| \boldsymbol{x}_i^{(r+1)} - \bar{\boldsymbol{x}}^{(r+1)} \right\|^2
$$

$$
\overset{(3)}{\leq} \frac{4\rho^2}{n} \sum_{i=1}^{n} \left\| \boldsymbol{h}_i^{(r)} - \nabla h_i(\bar{\boldsymbol{x}}^{(r)}) \right\|^2 + 4\delta^2 \rho^2 \left\| \bar{\boldsymbol{x}}^{(r+1)} - \bar{\boldsymbol{x}}^{(r)} \right\|^2 + \frac{32L^2}{n} \sum_{i=1}^{n} \left\| \boldsymbol{x}_i^{(r+1)} - \bar{\boldsymbol{x}}^{(r+1)} \right\|^2 .
$$

This concludes the statement. □

**Lemma 1.** *If Assumption 2 holds, there exists $\delta$ such that $\delta \leq L$ and Eq. (3) holds.*

*Proof.* We have

$$
\frac{1}{n} \sum_{i=1}^{n} \| \nabla f_i(\boldsymbol{x}) - \nabla f(\boldsymbol{x}) - \nabla f_i(\boldsymbol{y}) + \nabla f(\boldsymbol{y}) \|^2
$$

$$
= \frac{1}{n} \sum_{i=1}^{n} \| \nabla f_i(\boldsymbol{x}) - \nabla f_i(\boldsymbol{y}) \|^2 - \frac{2}{n} \sum_{i=1}^{n} \langle \nabla f_i(\boldsymbol{x}) - \nabla f_i(\boldsymbol{y}), \nabla f(\boldsymbol{x}) - \nabla f(\boldsymbol{y}) \rangle + \| \nabla f(\boldsymbol{x}) - \nabla f(\boldsymbol{y}) \|^2
$$

$$
\leq \frac{1}{n} \sum_{i=1}^{n} \| \nabla f_i(\boldsymbol{x}) - \nabla f_i(\boldsymbol{y}) \|^2
$$

$$
\leq L \| \boldsymbol{x} - \boldsymbol{y} \|^2 ,
$$

where we use Assumption 2 in the last inequality. □

# D. Proof of Theorem 1

**Lemma 9.** *Suppose that Assumptions 1 and 4 hold, and $\sum_{i=1}^n \boldsymbol{h}_i^{(0)} = \boldsymbol{0}$ and $M = 1$. Then, it holds that*

$$f(\boldsymbol{x}^\star) + \frac{\lambda}{2n} \sum_{i=1}^n \left\| \boldsymbol{x}_i^{(r)} - \boldsymbol{x}^\star \right\|^2 + \frac{1}{2\delta} \mathcal{E}^{(r)} \tag{18}$$

$$\geq f(\bar{\boldsymbol{x}}^{(r+1)}) + \frac{\mu+\lambda}{2n} \sum_{i=1}^n \left\| \boldsymbol{x}_i^{(r+1)} - \boldsymbol{x}^\star \right\|^2 + \left[ \frac{1}{\rho^2} \left( \frac{\mu+\lambda}{2} - \delta \right) - \frac{\mu+\lambda}{2} \right] \Xi^{(r+1)}$$

$$+ \left( \frac{\lambda}{4} - \frac{\delta}{2} \right) \left\| \bar{\boldsymbol{x}}^{(r+1)} - \bar{\boldsymbol{x}}^{(r)} \right\|^2 + \frac{\lambda}{4n} \sum_{i=1}^n \left\| \boldsymbol{x}_i^{(r+\frac{1}{2})} - \boldsymbol{x}_i^{(r)} \right\|^2$$

$$+ \frac{1}{n} \sum_{i=1}^n \left\langle \nabla F_{i,r}(\boldsymbol{x}_i^{(r+\frac{1}{2})}), \boldsymbol{x}^\star - \boldsymbol{x}_i^{(r+\frac{1}{2})} \right\rangle,$$

*where $\Xi^{(r)} := \frac{1}{n} \sum_{i=1}^n \|\boldsymbol{x}_i^{(r)} - \bar{\boldsymbol{x}}^{(r)}\|^2$ and $\mathcal{E}^{(r)} := \frac{1}{n} \sum_{i=1}^n \|\boldsymbol{h}_i^{(r)} - \nabla h_i(\bar{\boldsymbol{x}}^{(r)})\|^2$.*

*Proof.* We have

$$f_i(\boldsymbol{x}^\star) + \frac{\lambda}{2} \left\| \boldsymbol{x}_i^{(r)} - \boldsymbol{x}^\star \right\|^2$$

$$\overset{(1)}{\geq} f_i(\boldsymbol{x}_i^{(r+\frac{1}{2})}) + \left\langle \nabla f_i(\boldsymbol{x}_i^{(r+\frac{1}{2})}) + \lambda \left( \boldsymbol{x}_i^{(r+\frac{1}{2})} - \boldsymbol{x}_i^{(r)} \right), \boldsymbol{x}^\star - \boldsymbol{x}_i^{(r+\frac{1}{2})} \right\rangle + \frac{\lambda}{2} \left\| \boldsymbol{x}_i^{(r+\frac{1}{2})} - \boldsymbol{x}_i^{(r)} \right\|^2 + \frac{\mu+\lambda}{2} \left\| \boldsymbol{x}_i^{(r+\frac{1}{2})} - \boldsymbol{x}^\star \right\|^2$$

$$\overset{(1)}{\geq} f_i(\bar{\boldsymbol{x}}^{(r+1)}) + \left\langle \nabla f_i(\bar{\boldsymbol{x}}^{(r+1)}), \boldsymbol{x}_i^{(r+\frac{1}{2})} - \bar{\boldsymbol{x}}^{(r+1)} \right\rangle$$

$$+ \left\langle \nabla f_i(\boldsymbol{x}_i^{(r+\frac{1}{2})}) + \lambda \left( \boldsymbol{x}_i^{(r+\frac{1}{2})} - \boldsymbol{x}_i^{(r)} \right), \boldsymbol{x}^\star - \boldsymbol{x}_i^{(r+\frac{1}{2})} \right\rangle + \frac{\lambda}{2} \left\| \boldsymbol{x}_i^{(r+\frac{1}{2})} - \boldsymbol{x}_i^{(r)} \right\|^2 + \frac{\mu+\lambda}{2} \left\| \boldsymbol{x}_i^{(r+\frac{1}{2})} - \boldsymbol{x}^\star \right\|^2,$$

where we use the fact that $f_i(\boldsymbol{x}) + \frac{\lambda}{2}\|\boldsymbol{x} - \boldsymbol{x}_i^{(r)}\|^2$ is $(\mu + \lambda)$-strongly convex in the first inequality. From the definition of $F_{i,r}(\cdot)$, we have

$$\nabla F_{i,r}(\boldsymbol{x}_i^{(r+\frac{1}{2})}) = \nabla f_i(\boldsymbol{x}_i^{(r+\frac{1}{2})}) + \boldsymbol{h}_i^{(r)} + \lambda \left( \boldsymbol{x}_i^{(r+\frac{1}{2})} - \boldsymbol{x}_i^{(r)} \right).$$

Using the above equality, we obtain

$$f_i(\boldsymbol{x}^\star) + \frac{\lambda}{2} \left\| \boldsymbol{x}_i^{(r)} - \boldsymbol{x}^\star \right\|^2$$

$$\geq f_i(\bar{\boldsymbol{x}}^{(r+1)}) + \left\langle \nabla f_i(\bar{\boldsymbol{x}}^{(r+1)}), \boldsymbol{x}_i^{(r+\frac{1}{2})} - \bar{\boldsymbol{x}}^{(r+1)} \right\rangle + \left\langle \boldsymbol{h}_i^{(r)}, \boldsymbol{x}_i^{(r+\frac{1}{2})} - \boldsymbol{x}^\star \right\rangle$$

$$+ \left\langle \nabla F_{i,r}(\boldsymbol{x}_i^{(r+\frac{1}{2})}), \boldsymbol{x}^\star - \boldsymbol{x}_i^{(r+\frac{1}{2})} \right\rangle + \frac{\lambda}{2} \left\| \boldsymbol{x}_i^{(r+\frac{1}{2})} - \boldsymbol{x}_i^{(r)} \right\|^2 + \frac{\mu+\lambda}{2} \left\| \boldsymbol{x}_i^{(r+\frac{1}{2})} - \boldsymbol{x}^\star \right\|^2.$$

By summing up the above inequality from $i = 1$ to $i = n$, we obtain

$$f(\boldsymbol{x}^\star) + \frac{\lambda}{2n} \sum_{i=1}^n \left\| \boldsymbol{x}_i^{(r)} - \boldsymbol{x}^\star \right\|^2$$

$$\geq f(\bar{\boldsymbol{x}}^{(r+1)}) + \frac{1}{n} \sum_{i=1}^n \left[ \left\langle \nabla f_i(\bar{\boldsymbol{x}}^{(r+1)}), \boldsymbol{x}_i^{(r+\frac{1}{2})} - \bar{\boldsymbol{x}}^{(r+1)} \right\rangle + \left\langle \boldsymbol{h}_i^{(r)}, \boldsymbol{x}_i^{(r+\frac{1}{2})} - \boldsymbol{x}^\star \right\rangle \right.$$

$$+ \left. \left\langle \nabla F_{i,r}(\boldsymbol{x}_i^{(r+\frac{1}{2})}), \boldsymbol{x}^\star - \boldsymbol{x}_i^{(r+\frac{1}{2})} \right\rangle + \frac{\lambda}{2} \left\| \boldsymbol{x}_i^{(r+\frac{1}{2})} - \boldsymbol{x}_i^{(r)} \right\|^2 + \frac{\mu+\lambda}{2} \left\| \boldsymbol{x}_i^{(r+\frac{1}{2})} - \boldsymbol{x}^\star \right\|^2 \right]$$

$$= f(\bar{\boldsymbol{x}}^{(r+1)}) + \frac{1}{n} \sum_{i=1}^n \left[ \left\langle \boldsymbol{h}_i - \nabla h_i(\bar{\boldsymbol{x}}^{(r)}), \boldsymbol{x}_i^{(r+\frac{1}{2})} - \bar{\boldsymbol{x}}^{(r+1)} \right\rangle + \left\langle \nabla h_i(\bar{\boldsymbol{x}}^{(r)}) - \nabla h_i(\bar{\boldsymbol{x}}^{(r+1)}), \boldsymbol{x}_i^{(r+\frac{1}{2})} - \bar{\boldsymbol{x}}^{(r+1)} \right\rangle \right.$$

$$+ \left. \left\langle \nabla F_{i,r}(\boldsymbol{x}_i^{(r+\frac{1}{2})}), \boldsymbol{x}^\star - \boldsymbol{x}_i^{(r+\frac{1}{2})} \right\rangle + \frac{\lambda}{2} \left\| \boldsymbol{x}_i^{(r+\frac{1}{2})} - \boldsymbol{x}_i^{(r)} \right\|^2 + \frac{\mu+\lambda}{2} \left\| \boldsymbol{x}_i^{(r+\frac{1}{2})} - \boldsymbol{x}^\star \right\|^2 \right],$$

where we use Eq. (16) in the last equality. Then, we have

$$\frac{1}{n}\sum_{i=1}^{n}\left\|\boldsymbol{x}_i^{(r+\frac{1}{2})}-\boldsymbol{x}^\star\right\|^2 \overset{(14)}{=} \left\|\bar{\boldsymbol{x}}^{(r+1)}-\boldsymbol{x}^\star\right\|^2 + \frac{1}{n}\sum_{i=1}^{n}\left\|\boldsymbol{x}_i^{(r+\frac{1}{2})}-\bar{\boldsymbol{x}}^{(r+1)}\right\|^2$$

$$\overset{(14)}{=} \frac{1}{n}\sum_{i=1}^{n}\left(\left\|\boldsymbol{x}_i^{(r+1)}-\boldsymbol{x}^\star\right\|^2 - \left\|\boldsymbol{x}_i^{(r+1)}-\bar{\boldsymbol{x}}^{(r+1)}\right\|^2 + \left\|\boldsymbol{x}_i^{(r+\frac{1}{2})}-\bar{\boldsymbol{x}}^{(r+1)}\right\|^2\right)$$

$$\overset{(5)}{\geq} \frac{1}{n}\sum_{i=1}^{n}\left(\left\|\boldsymbol{x}_i^{(r+1)}-\boldsymbol{x}^\star\right\|^2 + \left(\frac{1}{\rho^2}-1\right)\left\|\boldsymbol{x}_i^{(r+1)}-\bar{\boldsymbol{x}}^{(r+1)}\right\|^2\right),$$

where we use the fact that $\sum_{i=1}^{n}\boldsymbol{x}_i^{(r+\frac{1}{2})} = \sum_{i=1}^{n}\boldsymbol{x}_i^{(r+1)}$ in the first equality (see Lemma 6). Using the above inequality, we have

$$f(\boldsymbol{x}^\star) + \frac{\lambda}{2n}\sum_{i=1}^{n}\left\|\boldsymbol{x}_i^{(r)}-\boldsymbol{x}^\star\right\|^2$$

$$\geq f(\bar{\boldsymbol{x}}^{(r+1)}) + \frac{\mu+\lambda}{2n}\sum_{i=1}^{n}\left\|\boldsymbol{x}_i^{(r+1)}-\boldsymbol{x}^\star\right\|^2$$

$$+ \frac{1}{n}\sum_{i=1}^{n}\left[\left\langle\boldsymbol{h}_i-\nabla h_i(\bar{\boldsymbol{x}}^{(r)}), \boldsymbol{x}_i^{(r+\frac{1}{2})}-\bar{\boldsymbol{x}}^{(r+1)}\right\rangle + \left\langle\nabla h_i(\bar{\boldsymbol{x}}^{(r)})-\nabla h_i(\bar{\boldsymbol{x}}^{(r+1)}), \boldsymbol{x}_i^{(r+\frac{1}{2})}-\bar{\boldsymbol{x}}^{(r+1)}\right\rangle\right.$$

$$\left. + \left\langle\nabla F_{i,r}(\boldsymbol{x}_i^{(r+\frac{1}{2})}), \boldsymbol{x}^\star-\boldsymbol{x}_i^{(r+\frac{1}{2})}\right\rangle + \frac{\lambda}{2}\left\|\boldsymbol{x}_i^{(r+\frac{1}{2})}-\boldsymbol{x}_i^{(r)}\right\|^2 + \frac{\mu+\lambda}{2}\left(\frac{1}{\rho^2}-1\right)\left\|\boldsymbol{x}_i^{(r+1)}-\bar{\boldsymbol{x}}^{(r+1)}\right\|^2\right].$$

Then, using Eq. (11) with $\alpha = \delta$, we obtain

$$f(\boldsymbol{x}^\star) + \frac{\lambda}{2n}\sum_{i=1}^{n}\left\|\boldsymbol{x}_i^{(r)}-\boldsymbol{x}^\star\right\|^2$$

$$\overset{(3)}{\geq} f(\bar{\boldsymbol{x}}^{(r+1)}) + \frac{\mu+\lambda}{2n}\sum_{i=1}^{n}\left\|\boldsymbol{x}_i^{(r+1)}-\boldsymbol{x}^\star\right\|^2$$

$$+ \frac{1}{n}\sum_{i=1}^{n}\left[-\frac{1}{2\delta}\left\|\boldsymbol{h}_i-\nabla h_i(\bar{\boldsymbol{x}}^{(r)})\right\|^2 - \frac{\delta}{2}\left\|\bar{\boldsymbol{x}}^{(r)}-\bar{\boldsymbol{x}}^{(r+1)}\right\|^2 - \delta\left\|\boldsymbol{x}_i^{(r+\frac{1}{2})}-\bar{\boldsymbol{x}}^{(r+1)}\right\|^2\right.$$

$$\left. + \left\langle\nabla F_{i,r}(\boldsymbol{x}_i^{(r+\frac{1}{2})}), \boldsymbol{x}^\star-\boldsymbol{x}_i^{(r+\frac{1}{2})}\right\rangle + \frac{\lambda}{2}\left\|\boldsymbol{x}_i^{(r+\frac{1}{2})}-\boldsymbol{x}_i^{(r)}\right\|^2 + \frac{\mu+\lambda}{2}\left(\frac{1}{\rho^2}-1\right)\left\|\boldsymbol{x}_i^{(r+1)}-\bar{\boldsymbol{x}}^{(r+1)}\right\|^2\right]$$

$$\overset{(5)}{\geq} f(\bar{\boldsymbol{x}}^{(r+1)}) + \frac{\mu+\lambda}{2n}\sum_{i=1}^{n}\left\|\boldsymbol{x}_i^{(r+1)}-\boldsymbol{x}^\star\right\|^2$$

$$+ \frac{1}{n}\sum_{i=1}^{n}\left[-\frac{1}{2\delta}\left\|\boldsymbol{h}_i-\nabla h_i(\bar{\boldsymbol{x}}^{(r)})\right\|^2 - \frac{\delta}{2}\left\|\bar{\boldsymbol{x}}^{(r)}-\bar{\boldsymbol{x}}^{(r+1)}\right\|^2\right.$$

$$\left. + \left\langle\nabla F_{i,r}(\boldsymbol{x}_i^{(r+\frac{1}{2})}), \boldsymbol{x}^\star-\boldsymbol{x}_i^{(r+\frac{1}{2})}\right\rangle + \frac{\lambda}{2}\left\|\boldsymbol{x}_i^{(r+\frac{1}{2})}-\boldsymbol{x}_i^{(r)}\right\|^2 + \left[\frac{\mu+\lambda}{2}\left(\frac{1}{\rho^2}-1\right)-\frac{\delta}{\rho^2}\right]\left\|\boldsymbol{x}_i^{(r+1)}-\bar{\boldsymbol{x}}^{(r+1)}\right\|^2\right]$$

$$\overset{(13)}{\geq} f(\bar{\boldsymbol{x}}^{(r+1)}) + \frac{\mu+\lambda}{2n}\sum_{i=1}^{n}\left\|\boldsymbol{x}_i^{(r+1)}-\boldsymbol{x}^\star\right\|^2$$

$$+ \frac{1}{n}\sum_{i=1}^{n}\left[-\frac{1}{2\delta}\left\|\boldsymbol{h}_i-\nabla h_i(\bar{\boldsymbol{x}}^{(r)})\right\|^2 + \left(\frac{\lambda}{4}-\frac{\delta}{2}\right)\left\|\bar{\boldsymbol{x}}^{(r)}-\bar{\boldsymbol{x}}^{(r+1)}\right\|^2\right.$$

$$\left. + \left\langle\nabla F_{i,r}(\boldsymbol{x}_i^{(r+\frac{1}{2})}), \boldsymbol{x}^\star-\boldsymbol{x}_i^{(r+\frac{1}{2})}\right\rangle + \frac{\lambda}{4}\left\|\boldsymbol{x}_i^{(r+\frac{1}{2})}-\boldsymbol{x}_i^{(r)}\right\|^2 + \left[\frac{\mu+\lambda}{2}\left(\frac{1}{\rho^2}-1\right)-\frac{\delta}{\rho^2}\right]\left\|\boldsymbol{x}_i^{(r+1)}-\bar{\boldsymbol{x}}^{(r+1)}\right\|^2\right].$$

This concludes the statement. $\qquad\square$

**Lemma 10.** *Suppose that Assumptions 1, 2 and 4 hold, $\sum_{i=1}^{n} h_i^{(0)} = 0$, $M = 1$, and*

$$\rho \leq \frac{\delta}{6L}.$$

*Then, when $\lambda = 4\delta$, it holds that*

$$f(\boldsymbol{x}^{\star}) + \frac{2\delta}{n} \sum_{i=1}^{n} \left\| \boldsymbol{x}_i^{(r)} - \boldsymbol{x}^{\star} \right\|^2 + \frac{1}{\delta} \mathcal{E}^{(r)} \tag{19}$$

$$\geq f(\bar{\boldsymbol{x}}^{(r+1)}) + \left(1 + \frac{\mu}{4\delta}\right) \frac{2\delta}{n} \sum_{i=1}^{n} \left\| \boldsymbol{x}_i^{(r+1)} - \boldsymbol{x}^{\star} \right\|^2 + \frac{1}{\delta} \left(1 + \frac{\mu}{4\delta}\right) \mathcal{E}^{(r+1)}$$

$$+ \frac{\delta}{n} \sum_{i=1}^{n} \left\| \boldsymbol{x}_i^{(r+\frac{1}{2})} - \boldsymbol{x}_i^{(r)} \right\|^2 + \frac{1}{n} \sum_{i=1}^{n} \left\langle \nabla F_{i,r}(\boldsymbol{x}_i^{(r+\frac{1}{2})}), \boldsymbol{x}^{\star} - \boldsymbol{x}_i^{(r+\frac{1}{2})} \right\rangle.$$

*Proof.* Since $\delta \leq L$, $\rho \leq \frac{\delta}{6L}$ implies that $\rho \leq \frac{1}{4}$. Using $\rho \leq \frac{1}{4}$ and Lemma 8, we obtain

$$\mathcal{E}^{(r+1)} \leq \rho \mathcal{E}^{(r)} + \delta^2 \rho \left\| \bar{\boldsymbol{x}}^{(r+1)} - \bar{\boldsymbol{x}}^{(r)} \right\|^2 + 32L^2 \Xi^{(r+1)}, \tag{20}$$

Using $\lambda = 4\delta$ and Eq. (18) $+ \frac{1}{\delta}(1 + \frac{\mu}{4\delta}) \times$ Eq. (20), we obtain

$$f(\boldsymbol{x}^{\star}) + \frac{4\delta}{2n} \sum_{i=1}^{n} \left\| \boldsymbol{x}_i^{(r)} - \boldsymbol{x}^{\star} \right\|^2 + \left( \frac{1}{2\delta} + \frac{\rho}{\delta} \left(1 + \frac{\mu}{4\delta}\right) \right) \mathcal{E}^{(r)}$$

$$\geq f(\bar{\boldsymbol{x}}^{(r+1)}) + \frac{\mu + 4\delta}{2n} \sum_{i=1}^{n} \left\| \boldsymbol{x}_i^{(r+1)} - \boldsymbol{x}^{\star} \right\|^2 + \frac{1}{\delta} \left(1 + \frac{\mu}{4\delta}\right) \mathcal{E}^{(r+1)}$$

$$+ \left[ \frac{1}{\rho^2} \left( \frac{\mu + 4\delta}{2} - \delta \right) - \frac{\mu + 4\delta}{2} - \frac{32L^2}{\delta} \left(1 + \frac{\mu}{4\delta}\right) \right] \Xi^{(r+1)}$$

$$+ \left( \frac{\delta}{2} - \delta\rho \left(1 + \frac{\mu}{4\delta}\right) \right) \left\| \bar{\boldsymbol{x}}^{(r+1)} - \bar{\boldsymbol{x}}^{(r)} \right\|^2$$

$$+ \frac{\delta}{n} \sum_{i=1}^{n} \left\| \boldsymbol{x}_i^{(r+\frac{1}{2})} - \boldsymbol{x}_i^{(r)} \right\|^2 + \frac{1}{n} \sum_{i=1}^{n} \left\langle \nabla F_{i,r}(\boldsymbol{x}_i^{(r+\frac{1}{2})}), \boldsymbol{x}^{\star} - \boldsymbol{x}_i^{(r+\frac{1}{2})} \right\rangle.$$

Using the assumption on $\rho$, we have

$$\frac{1}{2\delta} + \frac{\rho}{\delta} \left(1 + \frac{\mu}{4\delta}\right) \leq \frac{1}{\delta},$$

$$\frac{1}{\rho^2} \left( \frac{\mu + 4\delta}{2} - \delta \right) - \frac{\mu + 4\delta}{2} - \frac{32L^2}{\delta} \left(1 + \frac{\mu}{4\delta}\right) \geq 0,$$

$$\frac{\delta}{2} - \delta\rho \left(1 + \frac{\mu}{4\delta}\right) \geq 0,$$

where we use $\mu \leq L$ and $\delta \leq L$. This concludes the statement. $\qquad\square$

**Lemma 11.** *Suppose that Assumptions 1, 2 and 4 hold, $M = 1$, $\boldsymbol{x}_i^{(0)} = \bar{\boldsymbol{x}}^{(0)}$, $h_i^{(0)} := \nabla f(\bar{\boldsymbol{x}}^{(0)}) - \nabla f_i(\bar{\boldsymbol{x}}^{(0)})$, and*

$$\rho \leq \frac{\delta}{6L},$$

$$\sum_{i=1}^{n} \left\| \nabla F_{i,r}(\boldsymbol{x}_i^{(r+\frac{1}{2})}) \right\|^2 \leq e_{r+1} \sum_{i=1}^{n} \left\| \boldsymbol{x}_i^{(r+\frac{1}{2})} - \boldsymbol{x}_i^{(r)} \right\|^2.$$

*Then, when $\lambda = 4\delta$ and $\sum_{r=1}^{R} e_r \leq \frac{\delta(4\delta+\mu)}{4}$, we obtain*

$$\frac{1}{\sum_{r=1}^{R} \frac{1}{q^r}} \sum_{r=1}^{R} \frac{f(\bar{\boldsymbol{x}}^{(r)}) - f(\boldsymbol{x}^\star)}{q^r} \leq \frac{\mu}{(1+\frac{\mu}{4\delta})^R - 1} \left\| \bar{\boldsymbol{x}}^{(0)} - \boldsymbol{x}^\star \right\|^2 \leq \frac{4\delta}{R} \left\| \bar{\boldsymbol{x}}^{(0)} - \boldsymbol{x}^\star \right\|^2.$$

*Proof.* From Eq. (19), we have

$$f(\bar{\boldsymbol{x}}^{(r+1)}) - f(\boldsymbol{x}^\star) + \left(1 + \frac{\mu}{4\delta}\right) \left[\frac{2\delta}{n} \sum_{i=1}^{n} \left\| \boldsymbol{x}_i^{(r+1)} - \boldsymbol{x}^\star \right\|^2 + \frac{1}{\delta} \mathcal{E}^{(r+1)}\right] + \frac{\delta}{n} \sum_{i=1}^{n} \left\| \boldsymbol{x}_i^{(r+\frac{1}{2})} - \boldsymbol{x}_i^{(r)} \right\|^2$$

$$\leq \left[\frac{2\delta}{n} \sum_{i=1}^{n} \left\| \boldsymbol{x}_i^{(r)} - \boldsymbol{x}^\star \right\|^2 + \frac{1}{\delta} \mathcal{E}^{(r)}\right] + \frac{1}{n} \sum_{i=1}^{n} \left\langle \nabla F_{i,r}(\boldsymbol{x}_i^{(r+\frac{1}{2})}), \boldsymbol{x}_i^{(r+\frac{1}{2})} - \boldsymbol{x}^\star \right\rangle$$

$$\leq \left[\frac{2\delta}{n} \sum_{i=1}^{n} \left\| \boldsymbol{x}_i^{(r)} - \boldsymbol{x}^\star \right\|^2 + \frac{1}{\delta} \mathcal{E}^{(r)}\right] + \sqrt{\left(\frac{1}{n} \sum_{i=1}^{n} \left\| \nabla F_{i,r}(\boldsymbol{x}_i^{(r+\frac{1}{2})}) \right\|^2\right) \left(\frac{1}{n} \sum_{i=1}^{n} \left\| \boldsymbol{x}_i^{(r+\frac{1}{2})} - \boldsymbol{x}^\star \right\|^2\right)},$$

where we use Cauchy–Schwarz inequality in the last inequality. Using $\mathcal{E}^{(0)} = 0$ and dividing the above inequality by $\frac{4\delta+\mu}{2}$, we obtain

$$\frac{2}{4\delta+\mu} \left(f(\bar{\boldsymbol{x}}^{(r+1)}) - f(\boldsymbol{x}^\star)\right) + \frac{1}{n} \sum_{i=1}^{n} \left\| \boldsymbol{x}_i^{(r+1)} - \boldsymbol{x}^\star \right\|^2 + \frac{2\delta}{4\delta+\mu} \frac{1}{n} \sum_{i=1}^{n} \left\| \boldsymbol{x}_i^{(r+\frac{1}{2})} - \boldsymbol{x}_i^{(r)} \right\|^2$$

$$\leq \frac{4\delta}{4\delta+\mu} \frac{1}{n} \sum_{i=1}^{n} \left\| \boldsymbol{x}_i^{(r)} - \boldsymbol{x}^\star \right\|^2 + \frac{2}{4\delta+\mu} \sqrt{\left(\frac{1}{n} \sum_{i=1}^{n} \left\| \nabla F_{i,r}(\boldsymbol{x}_i^{(r+\frac{1}{2})}) \right\|^2\right) \left(\frac{1}{n} \sum_{i=1}^{n} \left\| \boldsymbol{x}_i^{(r+\frac{1}{2})} - \boldsymbol{x}^\star \right\|^2\right)},$$

Using the following notation,

$$A_{r+1} := \frac{1}{n} \sum_{i=1}^{n} \left\| \boldsymbol{x}_i^{(r+1)} - \boldsymbol{x}^\star \right\|^2,$$

$$B_{r+1} := \frac{1}{n} \sum_{i=1}^{n} \left\| \boldsymbol{x}_i^{(r+\frac{1}{2})} - \boldsymbol{x}_i^{(r)} \right\|^2,$$

$$C_{r+1} := \frac{1}{n} \sum_{i=1}^{n} \left\| \nabla F_{i,r}(\boldsymbol{x}_i^{(r+\frac{1}{2})}) \right\|^2,$$

$$q := \frac{4\delta}{4\delta+\mu},$$

we can simplify the above inequality as follows:

$$\frac{2}{4\delta+\mu} \left(f(\bar{\boldsymbol{x}}^{(r+1)}) - f(\boldsymbol{x}^\star)\right) + A_{r+1} + \frac{2\delta}{4\delta+\mu} B_{r+1} \leq q A_r + \frac{2}{4\delta+\mu} \sqrt{C_{r+1} A_{r+1}}.$$

Dividing the above inequality by $q^{r+1}$ and summing up the above inequality from $r = 0$ to $r = R - 1$, we get

$$\frac{2}{4\delta+\mu} \sum_{r=1}^{R} \frac{f(\bar{\boldsymbol{x}}^{(r)}) - f(\boldsymbol{x}^\star)}{q^r} + \frac{A_R}{q^R} + \frac{2\delta}{4\delta+\mu} \sum_{r=1}^{R} \frac{B_r}{q^r} \leq A_0 + \frac{2}{4\delta+\mu} \sum_{r=1}^{R} \frac{\sqrt{C_r A_r}}{q^r}.$$

Define $Q_R$ as follows:

$$Q_R := A_0 + \frac{2}{4\delta+\mu} \sum_{r=1}^{R} \frac{\sqrt{C_r A_r}}{q^r}.$$

The above inequality implies that $A_R \leq q^R Q_R$ for any $R$. Using this inequality, we get

$$Q_{R+1} - Q_R = \frac{2}{4\delta + \mu} \frac{\sqrt{C_{R+1} A_{R+1}}}{q^{R+1}}$$

$$\leq \frac{2}{4\delta + \mu} \sqrt{\frac{C_{R+1} Q_{R+1}}{q^{R+1}}}.$$

Since we now get the recursive inequality about $Q_R$, we obtain the following by using Lemma 12 in Jiang et al. (2024a).

$$Q_R \leq 2A_0 + \frac{8}{(4\delta + \mu)^2} \left( \sum_{r=1}^{R} \sqrt{\frac{C_r}{q^r}} \right)^2.$$

Then, by using $\sum_{i=1}^{n} \|\nabla F_{i,r}(\boldsymbol{x}_i^{(r+\frac{1}{2})})\|^2 \leq e_{r+1} \sum_{i=1}^{n} \|\boldsymbol{x}_i^{(r+\frac{1}{2})} - \boldsymbol{x}_i^{(r)}\|^2$, we have

$$\frac{2}{4\delta + \mu} \sum_{r=1}^{R} \frac{f(\bar{\boldsymbol{x}}^{(r)}) - f(\boldsymbol{x}^\star)}{q^r} + \frac{A_R}{q^R} + \frac{2\delta}{4\delta + \mu} \sum_{r=1}^{R} \frac{B_r}{q^r} \leq 2A_0 + \frac{8}{(4\delta + \mu)^2} \left( \sum_{r=1}^{R} \sqrt{\frac{e_r B_r}{q^r}} \right)^2$$

$$\leq 2A_0 + \frac{8}{(4\delta + \mu)^2} \left( \sum_{r=1}^{R} e_r \right) \left( \sum_{r=1}^{R} \frac{B_r}{q^r} \right),$$

where we use Cauchy–Schwarz inequality in the last inequality. Using the assumption that $\sum_{r=1}^{R} e_r \leq \frac{\delta(4\delta + \mu)}{4}$, we get

$$\frac{1}{4\delta + \mu} \sum_{r=1}^{R} \frac{f(\bar{\boldsymbol{x}}^{(r)}) - f(\boldsymbol{x}^\star)}{q^r} \leq \frac{1}{n} \sum_{i=1}^{n} \left\| \boldsymbol{x}_i^{(0)} - \boldsymbol{x}^\star \right\|^2.$$

Thus, we have

$$\frac{1}{\sum_{r=1}^{R} \frac{1}{q^r}} \sum_{r=1}^{R} \frac{f(\bar{\boldsymbol{x}}^{(r)}) - f(\boldsymbol{x}^\star)}{q^r} \leq \frac{\mu}{(1 + \frac{\mu}{4\delta})^R - 1} \frac{1}{n} \sum_{i=1}^{n} \left\| \boldsymbol{x}_i^{(0)} - \boldsymbol{x}^\star \right\|^2 \leq \frac{4\delta}{R} \frac{1}{n} \sum_{i=1}^{n} \left\| \boldsymbol{x}_i^{(0)} - \boldsymbol{x}^\star \right\|^2.$$

This concludes the statement. $\qquad\square$

**Lemma 12.** *Suppose that Assumptions 1, 2 and 4 hold, and $\boldsymbol{x}_i^{(0)} = \bar{\boldsymbol{x}}^{(0)}$, $\boldsymbol{h}_i^{(0)} := \nabla f(\bar{\boldsymbol{x}}^{(0)}) - \nabla f_i(\bar{\boldsymbol{x}}^{(0)})$, and*

$$\sum_{i=1}^{n} \left\| \nabla F_{i,r}(\boldsymbol{x}_i^{(r+\frac{1}{2})}) \right\|^2 \leq \frac{\delta(4\delta + \mu)}{4(r+1)(r+2)} \sum_{i=1}^{n} \left\| \boldsymbol{x}_i^{(r+\frac{1}{2})} - \boldsymbol{x}_i^{(r)} \right\|^2.$$

*Then, when $\lambda = 4\delta$ and $M \geq \frac{\log(\frac{6L}{\delta})}{1-\rho}$, we obtain*

$$\frac{1}{\sum_{r=1}^{R} \frac{1}{q^r}} \sum_{r=1}^{R} \frac{f(\bar{\boldsymbol{x}}^{(r)}) - f(\boldsymbol{x}^\star)}{q^r} \leq \frac{\mu}{(1 + \frac{\mu}{4\delta})^R - 1} \left\| \bar{\boldsymbol{x}}^{(0)} - \boldsymbol{x}^\star \right\|^2 \leq \frac{4\delta}{R} \left\| \bar{\boldsymbol{x}}^{(0)} - \boldsymbol{x}^\star \right\|^2.$$

*Proof.* When $M \geq \frac{\log(\frac{6L}{\delta})}{1-\rho}$, it holds that

$$\rho^M \leq \frac{\delta}{6L},$$

where we use $\log(x) \geq 1 - \frac{1}{x}$. Furthermore, it holds that

$$\sum_{r=1}^{R} \frac{\delta(4\delta + \mu)}{4r(r+1)} = \frac{\delta(4\delta + \mu)}{4} \sum_{r=1}^{R} \left( \frac{1}{r} - \frac{1}{r+1} \right) \leq \frac{\delta(4\delta + \mu)}{4}.$$

Thus, from Lemma 11, we obtain the desired result. $\qquad\square$

# E. Proof of Theorem 3

**Lemma 13.** *Suppose that Assumptions 1 and 4 hold and the following inequality is satisfied:*

$$\sum_{i=1}^{n} \left\| \nabla F_{i,r}(\boldsymbol{x}_i^{(r+1)}) \right\|^2 \leq \frac{\lambda^2}{10} \sum_{i=1}^{n} \left\| \boldsymbol{v}_i^{(r)} - \boldsymbol{x}_i^{(r+1)} \right\|^2, \tag{21}$$

*Then, when $\sum_{i=1}^{n} \boldsymbol{h}_i^{(0)} = \boldsymbol{0}$ and $M = 1$, it holds that*

$$f(\boldsymbol{x}^\star) + \frac{\lambda}{2n} \sum_{i=1}^{n} \left\| \boldsymbol{v}_i^{(r)} - \boldsymbol{x}^\star \right\|^2 + \frac{1}{2\delta} \mathcal{E}^{(r)} + 2\delta \Xi^{(r)} \tag{22}$$

$$\geq f(\bar{\boldsymbol{x}}^{(r+1)}) + \frac{\mu+\lambda}{2n} \sum_{i=1}^{n} \left\| \boldsymbol{v}_i^{(r+1)} - \boldsymbol{x}^\star \right\|^2 + \frac{\lambda+\mu}{2} \left( \frac{1}{\rho^2} - 1 \right) \Xi^{(r+1)} + \frac{\lambda}{8} \left\| \bar{\boldsymbol{v}}^{(r)} - \bar{\boldsymbol{v}}^{(r+1)} \right\|^2$$

$$+ \left( \frac{\lambda}{12} - \frac{3\delta}{2} \right) \left\| \bar{\boldsymbol{v}}^{(r)} - \bar{\boldsymbol{x}}^{(r+1)} \right\|^2,$$

*where $\Xi^{(r)} := \frac{1}{n} \sum_{i=1}^{n} \left\| \boldsymbol{v}_i^{(r)} - \bar{\boldsymbol{v}}^{(r)} \right\|^2$ and $\mathcal{E}^{(r)} := \frac{1}{n} \sum_{i=1}^{n} \left\| \boldsymbol{h}_i^{(r)} - \nabla h_i(\bar{\boldsymbol{v}}^{(r)}) \right\|^2$.*

*Proof.* We have

$$f(\boldsymbol{x}^\star) + \frac{\lambda}{2n} \sum_{i=1}^{n} \left\| \boldsymbol{v}_i^{(r)} - \boldsymbol{x}^\star \right\|^2$$

$$= \frac{1}{n} \sum_{i=1}^{n} \left( f_i(\boldsymbol{x}^\star) + \frac{\lambda}{2} \left\| \boldsymbol{v}_i^{(r)} - \boldsymbol{x}^\star \right\|^2 \right)$$

$$\overset{(1)}{\geq} \frac{1}{n} \sum_{i=1}^{n} \left( f_i(\boldsymbol{x}_i^{(r+1)}) + \left\langle \nabla f_i(\boldsymbol{x}_i^{(r+1)}), \boldsymbol{x}^\star - \boldsymbol{x}_i^{(r+1)} \right\rangle + \frac{\mu}{2} \left\| \boldsymbol{x}_i^{(r+1)} - \boldsymbol{x}^\star \right\|^2 + \frac{\lambda}{2} \left\| \boldsymbol{v}_i^{(r)} - \boldsymbol{x}^\star \right\|^2 \right)$$

$$\overset{(1)}{\geq} \frac{1}{n} \sum_{i=1}^{n} \left( f_i(\bar{\boldsymbol{x}}^{(r+1)}) + \left\langle \nabla f_i(\bar{\boldsymbol{x}}^{(r+1)}), \boldsymbol{x}_i^{(r+1)} - \bar{\boldsymbol{x}}^{(r+1)} \right\rangle + \left\langle \nabla f_i(\boldsymbol{x}_i^{(r+1)}), \boldsymbol{x}^\star - \boldsymbol{x}_i^{(r+1)} \right\rangle \right.$$

$$\left. + \frac{\mu}{2} \left\| \boldsymbol{x}_i^{(r+1)} - \bar{\boldsymbol{x}}^{(r+1)} \right\|^2 + \frac{\mu}{2} \left\| \boldsymbol{x}_i^{(r+1)} - \boldsymbol{x}^\star \right\|^2 + \frac{\lambda}{2} \left\| \boldsymbol{v}_i^{(r)} - \boldsymbol{x}^\star \right\|^2 \right).$$

Using the following equation,

$$\sum_{i=1}^{n} \left\langle \nabla f_i(\bar{\boldsymbol{x}}^{(r+1)}), \boldsymbol{x}_i^{(r+1)} - \bar{\boldsymbol{x}}^{(r+1)} \right\rangle = \sum_{i=1}^{n} \left\langle -\nabla h_i(\bar{\boldsymbol{x}}^{(r+1)}), \boldsymbol{x}_i^{(r+1)} - \bar{\boldsymbol{x}}^{(r+1)} \right\rangle,$$

we get

$$f(\boldsymbol{x}^\star) + \frac{\lambda}{2n} \sum_{i=1}^{n} \left\| \boldsymbol{v}_i^{(r)} - \boldsymbol{x}^\star \right\|^2$$

$$\geq f(\bar{\boldsymbol{x}}^{(r+1)}) + \frac{1}{n} \sum_{i=1}^{n} \left[ \left\langle -\nabla h_i(\bar{\boldsymbol{x}}^{(r+1)}), \boldsymbol{x}_i^{(r+1)} - \bar{\boldsymbol{x}}^{(r+1)} \right\rangle \right.$$

$$+ \underbrace{\left\langle \nabla f_i(\boldsymbol{x}_i^{(r+1)}), \boldsymbol{x}^\star - \boldsymbol{x}_i^{(r+1)} \right\rangle + \frac{\mu}{2} \left\| \boldsymbol{x}_i^{(r+1)} - \boldsymbol{x}^\star \right\|^2 + \frac{\lambda}{2} \left\| \boldsymbol{v}_i^{(r)} - \boldsymbol{x}^\star \right\|^2 + \frac{\mu}{2} \left\| \boldsymbol{x}_i^{(r+1)} - \bar{\boldsymbol{x}}^{(r+1)} \right\|^2}_{\mathcal{T}_i^\star} \right].$$

Note that $h_i(\boldsymbol{x}) := f(\boldsymbol{x}) - f_i(\boldsymbol{x})$. Using Eq. (16), we obtain

$$\frac{1}{n} \sum_{i=1}^{n} \mathcal{T}_i^\star \overset{(16)}{=} \frac{1}{n} \sum_{i=1}^{n} \left[ \left\langle \nabla f_i(\boldsymbol{x}_i^{(r+1)}) + \boldsymbol{h}_i^{(r)}, \boldsymbol{x}^\star \right\rangle + \left\langle \nabla f_i(\boldsymbol{x}_i^{(r+1)}), -\boldsymbol{x}_i^{(r+1)} \right\rangle + \frac{\mu}{2} \left\| \boldsymbol{x}_i^{(r+1)} - \boldsymbol{x}^\star \right\|^2 + \frac{\lambda}{2} \left\| \boldsymbol{v}_i^{(r)} - \boldsymbol{x}^\star \right\|^2 \right].$$

We define $G_{i,r}(\boldsymbol{v}) := \langle \nabla f_i(\boldsymbol{x}_i^{(r+1)}) + \boldsymbol{h}_i^{(r)}, \boldsymbol{v} \rangle + \frac{\mu}{2}\|\boldsymbol{v} - \boldsymbol{x}_i^{(r+1)}\|^2 + \frac{\lambda}{2}\|\boldsymbol{v} - \boldsymbol{v}_i^{(r)}\|^2$. By using that fact that $G_{i,r}$ is $(\mu + \lambda)$-strongly convex, we get

$$\frac{1}{n}\sum_{i=1}^{n}\mathcal{T}_i^{\star} \overset{(1)}{\geq} \frac{1}{n}\sum_{i=1}^{n}\left[\left\langle \nabla f_i(\boldsymbol{x}_i^{(r+1)}) + \boldsymbol{h}_i^{(r)}, \boldsymbol{v}_i^{(r+\frac{1}{2})}\right\rangle + \left\langle \nabla f_i(\boldsymbol{x}_i^{(r+1)}), -\boldsymbol{x}_i^{(r+1)}\right\rangle + \frac{\mu}{2}\left\|\boldsymbol{x}_i^{(r+1)} - \boldsymbol{v}_i^{(r+\frac{1}{2})}\right\|^2\right.$$
$$\left. + \frac{\lambda}{2}\left\|\boldsymbol{v}_i^{(r)} - \boldsymbol{v}_i^{(r+\frac{1}{2})}\right\|^2 + \frac{\lambda + \mu}{2}\left\|\boldsymbol{v}_i^{(r+\frac{1}{2})} - \boldsymbol{x}^{\star}\right\|^2\right]$$

$$\overset{(14)}{=} \frac{1}{n}\sum_{i=1}^{n}\left[\left\langle \nabla f_i(\boldsymbol{x}_i^{(r+1)}) + \boldsymbol{h}_i^{(r)}, \boldsymbol{v}_i^{(r+\frac{1}{2})}\right\rangle + \left\langle \nabla f_i(\boldsymbol{x}_i^{(r+1)}), -\boldsymbol{x}_i^{(r+1)}\right\rangle + \frac{\mu}{2}\left\|\boldsymbol{x}_i^{(r+1)} - \boldsymbol{v}_i^{(r+\frac{1}{2})}\right\|^2\right.$$
$$\left. + \frac{\lambda}{2}\left\|\boldsymbol{v}_i^{(r)} - \boldsymbol{v}_i^{(r+\frac{1}{2})}\right\|^2 + \frac{\lambda + \mu}{2}\left\|\boldsymbol{v}_i^{(r+\frac{1}{2})} - \bar{\boldsymbol{v}}^{(r+1)}\right\|^2 + \frac{\lambda + \mu}{2}\left\|\bar{\boldsymbol{v}}^{(r+1)} - \boldsymbol{x}^{\star}\right\|^2\right]$$

$$\overset{(14)}{=} \frac{1}{n}\sum_{i=1}^{n}\left[\left\langle \nabla f_i(\boldsymbol{x}_i^{(r+1)}) + \boldsymbol{h}_i^{(r)}, \boldsymbol{v}_i^{(r+\frac{1}{2})}\right\rangle + \left\langle \nabla f_i(\boldsymbol{x}_i^{(r+1)}), -\boldsymbol{x}_i^{(r+1)}\right\rangle + \frac{\mu}{2}\left\|\boldsymbol{x}_i^{(r+1)} - \boldsymbol{v}_i^{(r+\frac{1}{2})}\right\|^2\right.$$
$$+ \frac{\lambda}{2}\left\|\boldsymbol{v}_i^{(r)} - \boldsymbol{v}_i^{(r+\frac{1}{2})}\right\|^2 + \frac{\lambda + \mu}{2}\left\|\boldsymbol{v}_i^{(r+\frac{1}{2})} - \bar{\boldsymbol{v}}^{(r+1)}\right\|^2$$
$$\left. + \frac{\lambda + \mu}{2}\left\|\boldsymbol{v}_i^{(r+1)} - \boldsymbol{x}^{\star}\right\|^2 - \frac{\lambda + \mu}{2}\left\|\boldsymbol{v}_i^{(r+1)} - \bar{\boldsymbol{v}}^{(r+1)}\right\|^2\right].$$

Using the above inequality, we obtain

$$f(\boldsymbol{x}^{\star}) + \frac{\lambda}{2n}\sum_{i=1}^{n}\left\|\boldsymbol{v}_i^{(r)} - \boldsymbol{x}^{\star}\right\|^2$$

$$\geq f(\bar{\boldsymbol{x}}^{(r+1)}) + \frac{\mu + \lambda}{2n}\sum_{i=1}^{n}\left\|\boldsymbol{v}_i^{(r+1)} - \boldsymbol{x}^{\star}\right\|^2$$
$$+ \frac{1}{n}\sum_{i=1}^{n}\left[\left\langle -\nabla h_i(\bar{\boldsymbol{x}}^{(r+1)}), \boldsymbol{x}_i^{(r+1)} - \bar{\boldsymbol{x}}^{(r+1)}\right\rangle + \frac{\mu}{2}\left\|\boldsymbol{x}_i^{(r+1)} - \boldsymbol{v}_i^{(r+\frac{1}{2})}\right\|^2 + \frac{\lambda}{2}\left\|\boldsymbol{v}_i^{(r)} - \boldsymbol{v}_i^{(r+\frac{1}{2})}\right\|^2\right.$$
$$+ \frac{\lambda + \mu}{2}\left\|\boldsymbol{v}_i^{(r+\frac{1}{2})} - \bar{\boldsymbol{v}}^{(r+1)}\right\|^2 - \frac{\lambda + \mu}{2}\left\|\boldsymbol{v}_i^{(r+1)} - \bar{\boldsymbol{v}}^{(r+1)}\right\|^2$$
$$+ \underbrace{\left\langle \nabla f_i(\boldsymbol{x}_i^{(r+1)}) + \boldsymbol{h}_i^{(r)}, \boldsymbol{v}_i^{(r+\frac{1}{2})}\right\rangle + \left\langle \nabla f_i(\boldsymbol{x}_i^{(r+1)}), -\boldsymbol{x}_i^{(r+1)}\right\rangle}_{\mathcal{T}_i^{\star\star}}\bigg].$$

From the definition of $F_{i,r}$, we have

$$\nabla F_{i,r}(\boldsymbol{x}) = \nabla f_i(\boldsymbol{x}) + \boldsymbol{h}_i^{(r)} + \lambda(\boldsymbol{x} - \boldsymbol{v}_i^{(r)}).$$

By using the above equality, $\mathcal{T}_i^{\star\star}$ can be rewritten as follows:

$$\mathcal{T}_i^{\star\star} = \left\langle \nabla f_i(\boldsymbol{x}_i^{(r+1)}), \boldsymbol{v}_i^{(r+\frac{1}{2})} - \boldsymbol{x}_i^{(r+1)}\right\rangle + \left\langle \boldsymbol{h}_i^{(r)}, \boldsymbol{v}_i^{(r+\frac{1}{2})}\right\rangle$$
$$= \left\langle \nabla F_{i,r}(\boldsymbol{x}_i^{(r+1)}) + \lambda(\boldsymbol{v}_i^{(r)} - \boldsymbol{x}_i^{(r+1)}), \boldsymbol{v}_i^{(r+\frac{1}{2})} - \boldsymbol{x}_i^{(r+1)}\right\rangle + \left\langle \boldsymbol{h}_i^{(r)}, \boldsymbol{x}_i^{(r+1)}\right\rangle$$
$$= \left\langle \nabla F_{i,r}(\boldsymbol{x}_i^{(r+1)}), \boldsymbol{v}_i^{(r)} - \boldsymbol{x}_i^{(r+1)}\right\rangle + \lambda\left\|\boldsymbol{v}_i^{(r)} - \boldsymbol{x}_i^{(r+1)}\right\|^2 + \left\langle \boldsymbol{h}_i^{(r)}, \boldsymbol{x}_i^{(r+1)}\right\rangle$$
$$+ \underbrace{\left\langle \nabla F_{i,r}(\boldsymbol{x}_i^{(r+1)}) + \lambda(\boldsymbol{v}_i^{(r)} - \boldsymbol{x}_i^{(r+1)}), \boldsymbol{v}_i^{(r+\frac{1}{2})} - \boldsymbol{v}_i^{(r)}\right\rangle}_{\mathcal{T}_i^{\star\star\star}}.$$

Then, $\mathcal{T}_i^{\star\star\star}$ can be bounded as follows:

$$\mathcal{T}_i^{\star\star\star} \overset{(11)}{\geq} -\frac{2}{3\lambda}\left\|\nabla F_{i,r}(\boldsymbol{x}_i^{(r+1)}) + \lambda(\boldsymbol{v}_i^{(r)} - \boldsymbol{x}_i^{(r+1)})\right\|^2 - \frac{3\lambda}{8}\left\|\boldsymbol{v}_i^{(r+\frac{1}{2})} - \boldsymbol{v}_i^{(r)}\right\|^2$$
$$= -\frac{2}{3\lambda}\left\|\nabla F_{i,r}(\boldsymbol{x}_i^{(r+1)})\right\|^2 - \frac{4}{3}\left\langle \nabla F_{i,r}(\boldsymbol{x}_i^{(r+1)}), \boldsymbol{v}_i^{(r)} - \boldsymbol{x}_i^{(r+1)}\right\rangle - \frac{2\lambda}{3}\left\|\boldsymbol{v}_i^{(r)} - \boldsymbol{x}_i^{(r+1)}\right\|^2 - \frac{3\lambda}{8}\left\|\boldsymbol{v}_i^{(r+\frac{1}{2})} - \boldsymbol{v}_i^{(r)}\right\|^2.$$

Using the above inequality, we get

$$
\begin{aligned}
\mathcal{T}_i^{\star\star} &\geq \frac{\lambda}{3} \left\| v_i^{(r)} - x_i^{(r+1)} \right\|^2 + \left\langle h_i^{(r)}, x_i^{(r+1)} \right\rangle - \frac{2}{3\lambda} \left\| \nabla F_{i,r}(x_i^{(r+1)}) \right\|^2 - \frac{1}{3} \left\langle \nabla F_{i,r}(x_i^{(r+1)}), v_i^{(r)} - x_i^{(r+1)} \right\rangle \\
&\quad - \frac{3\lambda}{8} \left\| v_i^{(r+\frac{1}{2})} - v_i^{(r)} \right\|^2 \\
&\overset{(11)}{\geq} \frac{\lambda}{6} \left\| v_i^{(r)} - x_i^{(r+1)} \right\|^2 + \left\langle h_i^{(r)}, x_i^{(r+1)} \right\rangle - \frac{5}{6\lambda} \left\| \nabla F_{i,r}(x_i^{(r+1)}) \right\|^2 - \frac{3\lambda}{8} \left\| v_i^{(r+\frac{1}{2})} - v_i^{(r)} \right\|^2 .
\end{aligned}
$$

Using the assumption that $\sum_{i=1}^n \left\| \nabla F_{i,r}(x_i^{(r+1)}) \right\|^2 \leq \frac{\lambda^2}{10} \sum_{i=1}^n \left\| v_i^{(r)} - x_i^{(r+1)} \right\|^2$, we get

$$
\sum_{i=1}^n \mathcal{T}_i^{\star\star} \geq \sum_{i=1}^n \left[ \frac{\lambda}{12} \left\| v_i^{(r)} - x_i^{(r+1)} \right\|^2 + \left\langle h_i^{(r)}, x_i^{(r+1)} \right\rangle - \frac{3\lambda}{8} \left\| v_i^{(r+\frac{1}{2})} - v_i^{(r)} \right\|^2 \right] .
$$

Using the above inequalities, we get

$$
\begin{aligned}
f(x^\star) &+ \frac{\lambda}{2n} \sum_{i=1}^n \left\| v_i^{(r)} - x^\star \right\|^2 \\
&\geq f(\bar{x}^{(r+1)}) + \frac{\mu + \lambda}{2n} \sum_{i=1}^n \left\| v_i^{(r+1)} - x^\star \right\|^2 \\
&\quad + \frac{1}{n} \sum_{i=1}^n \left[ \frac{\mu}{2} \left\| x_i^{(r+1)} - v_i^{(r+\frac{1}{2})} \right\|^2 + \frac{\lambda}{8} \left\| v_i^{(r)} - v_i^{(r+\frac{1}{2})} \right\|^2 + \frac{\lambda + \mu}{2} \left\| v_i^{(r+\frac{1}{2})} - \bar{v}^{(r+1)} \right\|^2 - \frac{\lambda + \mu}{2} \left\| v_i^{(r+1)} - \bar{v}^{(r+1)} \right\|^2 \right. \\
&\quad \left. + \frac{\lambda}{12} \left\| v_i^{(r)} - x_i^{(r+1)} \right\|^2 + \underbrace{\left\langle h_i^{(r)}, x_i^{(r+1)} \right\rangle + \left\langle -\nabla h_i(\bar{x}^{(r+1)}), x_i^{(r+1)} - \bar{x}^{(r+1)} \right\rangle}_{\mathcal{T}_i^{\star\star\star\star}} \right]
\end{aligned}
$$

$\mathcal{T}_i^{\star\star\star\star}$ can be bounded by blow as follows:

$$
\begin{aligned}
\frac{1}{n} \sum_{i=1}^n \mathcal{T}_i^{\star\star\star\star} &\overset{(16)}{=} \frac{1}{n} \sum_{i=1}^n \left\langle h_i^{(r)} - \nabla h_i(\bar{x}^{(r+1)}), x_i^{(r+1)} - \bar{v}^{(r)} \right\rangle \\
&= \frac{1}{n} \sum_{i=1}^n \left[ \left\langle h_i^{(r)} - \nabla h_i(\bar{v}^{(r)}), x_i^{(r+1)} - \bar{v}^{(r)} \right\rangle + \left\langle \nabla h_i(\bar{v}^{(r)}) - \nabla h_i(\bar{x}^{(r+1)}), x_i^{(r+1)} - \bar{v}^{(r)} \right\rangle \right] \\
&\overset{(11)}{\geq} \frac{1}{n} \sum_{i=1}^n \left[ -\frac{1}{2\delta} \left\| h_i^{(r)} - \nabla h_i(\bar{v}^{(r)}) \right\|^2 - \delta \left\| x_i^{(r+1)} - \bar{v}^{(r)} \right\|^2 - \frac{1}{2\delta} \left\| \nabla h_i(\bar{v}^{(r)}) - \nabla h_i(\bar{x}^{(r+1)}) \right\|^2 \right] \\
&\overset{(3)}{\geq} \frac{1}{n} \sum_{i=1}^n \left[ -\frac{1}{2\delta} \left\| h_i^{(r)} - \nabla h_i(\bar{v}^{(r)}) \right\|^2 - \delta \left\| x_i^{(r+1)} - \bar{v}^{(r)} \right\|^2 - \frac{\delta}{2} \left\| \bar{v}^{(r)} - \bar{x}^{(r+1)} \right\|^2 \right] \\
&\overset{(12)}{\geq} \frac{1}{n} \sum_{i=1}^n \left[ -\frac{1}{2\delta} \left\| h_i^{(r)} - \nabla h_i(\bar{v}^{(r)}) \right\|^2 - 2\delta \left\| x_i^{(r+1)} - v_i^{(r)} \right\|^2 - 2\delta \left\| v_i^{(r)} - \bar{v}^{(r)} \right\|^2 - \frac{\delta}{2} \left\| \bar{v}^{(r)} - \bar{x}^{(r+1)} \right\|^2 \right] .
\end{aligned}
$$

Using the above inequality, we obtain

$$
f(\boldsymbol{x}^\star) + \frac{\lambda}{2n} \sum_{i=1}^{n} \left\| \boldsymbol{v}_i^{(r)} - \boldsymbol{x}^\star \right\|^2 + \frac{1}{2\delta n} \sum_{i=1}^{n} \left\| \boldsymbol{h}_i^{(r)} - \nabla h_i(\bar{\boldsymbol{v}}^{(r)}) \right\|^2 + \frac{2\delta}{n} \sum_{i=1}^{n} \left\| \boldsymbol{v}_i^{(r)} - \bar{\boldsymbol{v}}^{(r)} \right\|^2
$$

$$
\geq f(\bar{\boldsymbol{x}}^{(r+1)}) + \frac{\mu + \lambda}{2n} \sum_{i=1}^{n} \left\| \boldsymbol{v}_i^{(r+1)} - \boldsymbol{x}^\star \right\|^2
$$

$$
+ \frac{1}{n} \sum_{i=1}^{n} \left[ \frac{\mu}{2} \left\| \boldsymbol{x}_i^{(r+1)} - \boldsymbol{v}_i^{(r+\frac{1}{2})} \right\|^2 + \frac{\lambda}{8} \left\| \boldsymbol{v}_i^{(r)} - \boldsymbol{v}_i^{(r+\frac{1}{2})} \right\|^2 + \frac{\lambda + \mu}{2} \left\| \boldsymbol{v}_i^{(r+\frac{1}{2})} - \bar{\boldsymbol{v}}^{(r+1)} \right\|^2 - \frac{\lambda + \mu}{2} \left\| \boldsymbol{v}_i^{(r+1)} - \bar{\boldsymbol{v}}^{(r+1)} \right\|^2 \right.
$$

$$
\left. + \left( \frac{\lambda}{12} - 2\delta \right) \left\| \boldsymbol{v}_i^{(r)} - \boldsymbol{x}_i^{(r+1)} \right\|^2 - \frac{\delta}{2} \left\| \bar{\boldsymbol{v}}^{(r)} - \bar{\boldsymbol{x}}^{(r+1)} \right\|^2 \right]
$$

$$
\overset{(5)}{\geq} f(\bar{\boldsymbol{x}}^{(r+1)}) + \frac{\mu + \lambda}{2n} \sum_{i=1}^{n} \left\| \boldsymbol{v}_i^{(r+1)} - \boldsymbol{x}^\star \right\|^2 + \frac{\lambda + \mu}{2n} \left( \frac{1}{\rho^2} - 1 \right) \sum_{i=1}^{n} \left\| \boldsymbol{v}_i^{(r+1)} - \bar{\boldsymbol{v}}^{(r+1)} \right\|^2
$$

$$
+ \frac{1}{n} \sum_{i=1}^{n} \left[ \frac{\mu}{2} \left\| \boldsymbol{x}_i^{(r+1)} - \boldsymbol{v}_i^{(r+\frac{1}{2})} \right\|^2 + \frac{\lambda}{8} \left\| \boldsymbol{v}_i^{(r)} - \boldsymbol{v}_i^{(r+\frac{1}{2})} \right\|^2 + \left( \frac{\lambda}{12} - 2\delta \right) \left\| \boldsymbol{v}_i^{(r)} - \boldsymbol{x}_i^{(r+1)} \right\|^2 - \frac{\delta}{2} \left\| \bar{\boldsymbol{v}}^{(r)} - \bar{\boldsymbol{x}}^{(r+1)} \right\|^2 \right].
$$

Using Eq. (16), we get the desired result. $\qquad\square$

**Lemma 14.** *Suppose that Assumptions 1 and 4 hold and the following inequality is satisfied:*

$$
\sum_{i=1}^{n} \left\| \nabla F_{i,r}(\boldsymbol{x}_i^{(r+1)}) \right\|^2 \leq \frac{\lambda^2}{10} \sum_{i=1}^{n} \left\| \boldsymbol{v}_i^{(r)} - \boldsymbol{x}_i^{(r+1)} \right\|^2. \tag{23}
$$

*Then, when $\lambda = 20\delta$, $\sum_{i=1}^{n} \boldsymbol{h}_i^{(0)} = \boldsymbol{0}$, $M = 1$, and $\rho \leq \frac{\delta}{5L}$, it holds that*

$$
f(\boldsymbol{x}^\star) + \frac{10\delta}{n} \sum_{i=1}^{n} \left\| \boldsymbol{v}_i^{(r)} - \boldsymbol{x}^\star \right\|^2 + \frac{1}{\delta} \mathcal{E}^{(r)} + 2\delta \Xi^{(r)}
$$

$$
\geq f(\bar{\boldsymbol{x}}^{(r+1)}) + \left( 1 + \frac{\mu}{20\delta} \right) \left[ \frac{10\delta}{n} \sum_{i=1}^{n} \left\| \boldsymbol{v}_i^{(r+1)} - \boldsymbol{x}^\star \right\|^2 + \frac{1}{\delta} \mathcal{E}^{(r+1)} + 2\delta \Xi^{(r+1)} \right], \tag{24}
$$

*where $\Xi^{(r)} := \frac{1}{n} \sum_{i=1}^{n} \left\| \boldsymbol{v}_i^{(r)} - \bar{\boldsymbol{v}}^{(r)} \right\|^2$ and $\mathcal{E}^{(r)} := \frac{1}{n} \sum_{i=1}^{n} \left\| \boldsymbol{h}_i^{(r)} - \nabla h_i(\bar{\boldsymbol{v}}^{(r)}) \right\|^2$.*

*Proof.* Since $\delta \leq L$, $\rho \leq \frac{\delta}{5L}$ implies $\rho \leq \frac{1}{4}$. Thus, using Lemma 8, we obtain

$$
\rho \mathcal{E}^{(r)} \geq \mathcal{E}^{(r+1)} - \rho \delta^2 \left\| \bar{\boldsymbol{v}}^{(r+1)} - \bar{\boldsymbol{v}}^{(r)} \right\|^2 - 32 L^2 \Xi^{(r+1)}. \tag{25}
$$

By calculating Eq. (22) + Eq. (25) $\times \frac{1}{\delta}(1 + \frac{\mu}{\lambda})$, we obtain

$$
f(\boldsymbol{x}^\star) + \frac{\lambda}{2n} \sum_{i=1}^{n} \left\| \boldsymbol{v}_i^{(r)} - \boldsymbol{x}^\star \right\|^2 + \frac{1}{\delta} \left( \frac{1}{2} + \rho \left( 1 + \frac{\mu}{\lambda} \right) \right) \mathcal{E}^{(r)} + 2\delta \Xi^{(r)}
$$

$$
\geq f(\bar{\boldsymbol{x}}^{(r+1)}) + \frac{\mu + \lambda}{2n} \sum_{i=1}^{n} \left\| \boldsymbol{v}_i^{(r+1)} - \boldsymbol{x}^\star \right\|^2 + \frac{1}{\delta} \left( 1 + \frac{\mu}{\lambda} \right) \mathcal{E}^{(r+1)} + \left( 1 + \frac{\mu}{\lambda} \right) \left( \frac{\lambda}{2} \left( \frac{1}{\rho^2} - 1 \right) - \frac{32 L^2}{\delta} \right) \Xi^{(r+1)}
$$

$$
+ \left( \frac{\lambda}{8} - \left( 1 + \frac{\mu}{\lambda} \right) \rho \delta \right) \left\| \bar{\boldsymbol{v}}^{(r)} - \bar{\boldsymbol{v}}^{(r+1)} \right\|^2 + \left( \frac{\lambda}{12} - \frac{3\delta}{2} \right) \left\| \bar{\boldsymbol{v}}^{(r)} - \bar{\boldsymbol{x}}^{(r+1)} \right\|^2,
$$

By inserting $\lambda = 20\delta$, we obtain

$$f(\boldsymbol{x}^\star) + \frac{10\delta}{n} \sum_{i=1}^n \left\| \boldsymbol{v}_i^{(r)} - \boldsymbol{x}^\star \right\|^2 + \frac{1}{\delta} \left( \frac{1}{2} + \rho \left( 1 + \frac{\mu}{20\delta} \right) \right) \mathcal{E}^{(r)} + 2\delta \Xi^{(r)}$$

$$\geq f(\bar{\boldsymbol{x}}^{(r+1)}) + \frac{10\delta}{n} \left( 1 + \frac{\mu}{20\delta} \right) \sum_{i=1}^n \left\| \boldsymbol{v}_i^{(r+1)} - \boldsymbol{x}^\star \right\|^2 + \frac{1}{\delta} \left( 1 + \frac{\mu}{20\delta} \right) \mathcal{E}^{(r+1)}$$

$$+ \left( 1 + \frac{\mu}{20\delta} \right) \left( 10\delta \left( \frac{1}{\rho^2} - 1 \right) - \frac{32L^2}{\delta} \right) \Xi^{(r+1)} + \left( \frac{5\delta}{2} - \left( 1 + \frac{\mu}{10\delta} \right) \rho\delta \right) \left\| \bar{\boldsymbol{v}}^{(r)} - \bar{\boldsymbol{v}}^{(r+1)} \right\|^2.$$

$\rho \leq \frac{\delta}{5L}$ implies that

$$\frac{1}{2} + \rho \left( 1 + \frac{\mu}{20\delta} \right) \leq 1,$$

$$10\delta \left( \frac{1}{\rho^2} - 1 \right) - \frac{32L^2}{\delta} \geq 2\delta,$$

$$\frac{5\delta}{2} - \left( 1 + \frac{\mu}{20\delta} \right) \rho\delta \geq 0,$$

where we use $\mu \leq L$ and $\mu \leq L$. Using the above inequalities, we get

$$f(\boldsymbol{x}^\star) + \frac{10\delta}{n} \sum_{i=1}^n \left\| \boldsymbol{v}_i^{(r)} - \boldsymbol{x}^\star \right\|^2 + \frac{1}{\delta} \mathcal{E}^{(r)} + 2\delta \Xi^{(r)}$$

$$\geq f(\bar{\boldsymbol{x}}^{(r+1)}) + \frac{10\delta}{n} \left( 1 + \frac{\mu}{20\delta} \right) \sum_{i=1}^n \left\| \boldsymbol{v}_i^{(r+1)} - \boldsymbol{x}^\star \right\|^2 + \frac{1}{\delta} \left( 1 + \frac{\mu}{20\delta} \right) \mathcal{E}^{(r+1)} + 2\delta \left( 1 + \frac{\mu}{20\delta} \right) \Xi^{(r+1)}.$$

Thus, we can get the desired result. $\qquad \square$

**Lemma 15.** *Suppose that Assumptions 1 and 4 hold and the following inequality is satisfied:*

$$\sum_{i=1}^n \left\| \nabla F_{i,r}(\boldsymbol{x}_i^{(r+1)}) \right\|^2 \leq \frac{\lambda^2}{10} \sum_{i=1}^n \left\| \boldsymbol{v}_i^{(r)} - \boldsymbol{x}_i^{(r+1)} \right\|^2. \tag{26}$$

*Then, when $\lambda = 20\delta$, $M = 1$, $\boldsymbol{v}_i^{(0)} = \bar{\boldsymbol{v}}^{(0)}$, $\boldsymbol{h}_i^{(0)} = \nabla f(\bar{\boldsymbol{v}}^{(0)}) - \nabla f_i(\bar{\boldsymbol{v}}^{(0)})$, and $\rho \leq \frac{\delta}{5L}$, we obtain*

$$\frac{1}{W_R} \sum_{r=0}^{R-1} w^{(r)} \left( f(\bar{\boldsymbol{x}}^{(r+1)}) - f(\boldsymbol{x}^\star) \right) \leq \frac{\mu}{2((1 + \frac{\mu}{20\delta})^R - 1)} \left\| \bar{\boldsymbol{v}}^{(0)} - \boldsymbol{x}^\star \right\|^2 \leq \frac{10\delta}{R} \left\| \bar{\boldsymbol{v}}^{(0)} - \boldsymbol{x}^\star \right\|^2,$$

*where $w^{(r)} := (1 + \frac{\mu}{20\delta})^r$ and $W_R := \sum_{r=0}^{R-1} w^{(r)}$.*

*Proof.* From Eq. (24), we have

$$f(\bar{\boldsymbol{x}}^{(r+1)}) - f(\boldsymbol{x}^\star)$$

$$\leq 10\delta \left[ \frac{1}{n} \sum_{i=1}^n \left\| \boldsymbol{v}_i^{(r)} - \boldsymbol{x}^\star \right\|^2 + \frac{1}{10\delta^2} \mathcal{E}^{(r)} + \frac{1}{5} \Xi^{(r)} \right] - 10\delta \left( 1 + \frac{\mu}{20\delta} \right) \left[ \frac{1}{n} \sum_{i=1}^n \left\| \boldsymbol{v}_i^{(r+1)} - \boldsymbol{x}^\star \right\|^2 + \frac{1}{10\delta^2} \mathcal{E}^{(r+1)} + \frac{1}{5} \Xi^{(r+1)} \right].$$

We define $w^{(r)} := (1 + \frac{\mu}{20\delta})^r$. We obtain

$$\frac{1}{W_R} \sum_{r=0}^{R-1} w^{(r)} \left( f(\bar{\boldsymbol{x}}^{(r+1)}) - f(\boldsymbol{x}^\star) \right) \leq \frac{10\delta}{W_R} \left[ \frac{1}{n} \sum_{i=1}^n \left\| \boldsymbol{v}_i^{(0)} - \boldsymbol{x}^\star \right\|^2 + \frac{1}{10\delta^2} \mathcal{E}^{(0)} + \frac{1}{5} \Xi^{(0)} \right],$$

where $W_R := \sum_{r=0}^{R-1} w^{(r)}$. Using $\mathcal{E}^{(0)} = 0$ and $\Xi^{(0)} = 0$, we obtain

$$\frac{1}{W_R} \sum_{r=0}^{R-1} w^{(r)} \left( f(\bar{\boldsymbol{x}}^{(r+1)}) - f(\boldsymbol{x}^\star) \right) \leq \frac{10\delta}{W_R} \left\| \bar{\boldsymbol{v}}^{(0)} - \boldsymbol{x}^\star \right\|^2.$$

Using $W_R = \frac{20\delta}{\mu}\left(\left(1 + \frac{\mu}{20\delta}\right)^R - 1\right)$, we obtain

$$\frac{1}{W_R} \sum_{r=0}^{R-1} w^{(r)} \left( f(\bar{\boldsymbol{x}}^{(r+1)}) - f(\boldsymbol{x}^\star) \right) \leq \frac{\mu}{2((1 + \frac{\mu}{20\delta})^R - 1)} \left\| \bar{\boldsymbol{v}}^{(0)} - \boldsymbol{x}^\star \right\|^2 \leq \frac{10\delta}{R} \left\| \bar{\boldsymbol{v}}^{(0)} - \boldsymbol{x}^\star \right\|^2.$$

$\square$

**Lemma 16.** *Suppose that Assumptions 1, 2, and 4 hold and the following inequality is satisfied:*

$$\sum_{i=1}^{n} \left\| \nabla F_{i,r}(\boldsymbol{x}_i^{(r+1)}) \right\|^2 \leq \frac{\lambda^2}{10} \sum_{i=1}^{n} \left\| \boldsymbol{v}_i^{(r)} - \boldsymbol{x}_i^{(r+1)} \right\|^2. \tag{27}$$

*When $\boldsymbol{v}_i^{(0)} = \bar{\boldsymbol{v}}^{(0)}$, $\boldsymbol{h}_i^{(0)} = \nabla f(\bar{\boldsymbol{v}}^{(0)}) - \nabla f_i(\bar{\boldsymbol{v}}^{(0)})$, $\lambda = 20\delta$, and $M \geq \frac{\log(\frac{5L}{\delta})}{1-\rho}$ it holds that*

$$\frac{1}{W_R} \sum_{r=0}^{R-1} w^{(r)} \left( f(\bar{\boldsymbol{x}}^{(r+1)}) - f(\boldsymbol{x}^\star) \right) \leq \frac{\mu}{2(1 + \frac{\mu}{20\delta})^R - 1)} \left\| \bar{\boldsymbol{v}}^{(0)} - \boldsymbol{x}^\star \right\|^2 \leq \frac{10\delta}{R} \left\| \bar{\boldsymbol{v}}^{(0)} - \boldsymbol{x}^\star \right\|^2,$$

*where $w^{(r)} := (1 + \frac{\mu}{20\delta})^r$ and $W_R := \sum_{r=0}^{R-1} w^{(r)}$.*

*Proof.* When $M \geq \frac{1}{1-\rho} \log(\frac{5L}{\delta})$, it holds that

$$\rho^M \leq \frac{\delta}{5L},$$

where we use $\log(x) \geq 1 - \frac{1}{x}$. From Lemma 15, we obtain the desired result. $\square$

# F. Proof of Theorem 5

## F.1. Useful Lemmas

**Lemma 17.** *When $\mu \leq 4\lambda$, it holds that for any $r \geq 0$*

$$A_r \geq \frac{1}{4\mu}\left[\left(1+\sqrt{\frac{\mu}{4\lambda}}\right)^r - \left(1-\sqrt{\frac{\mu}{4\lambda}}\right)^r\right]^2 \geq \frac{r^2}{4\lambda}.$$

*Otherwise, for any $r \geq 1$, it holds that*

$$A_r \geq \frac{1}{4\lambda}\left(1+\sqrt{\frac{\mu}{4\lambda}}\right)^{2(r-1)}.$$

*Proof.* See Lemma 12 in Jiang et al. (2024b). $\qquad\square$

**Lemma 18.** *When $\mu = 0$, it holds that for any $0 \leq r \leq R$,*

$$A_r \leq \frac{r^2}{\lambda}. \tag{28}$$

*Proof.* Denote $C_r := \sqrt{A_r}$. From the definition of $A_r$ and $a_r$, it holds that

$$C_{r+1}^2 = \lambda(C_{r+1}^2 - C_r^2)^2 = \lambda(C_{r+1} - C_r)^2(C_{r+1} + C_r)^2 \geq \lambda(C_{r+1} - C_r)^2 C_{r+1}^2.$$

Thus, we have

$$C_{r+1} - C_r \geq \sqrt{\frac{1}{\lambda}}.$$

This concludes the statement. $\qquad\square$

**Lemma 19.** *It holds that for any $r \geq 1$,*

$$2\left(1+\frac{\mu}{\lambda}\right)^{-1}\left(1+\sqrt{1+\frac{4\lambda}{\mu}}\right)^{-1} \leq \frac{A_r}{a_{r+1}} \leq \frac{2\lambda}{\mu}. \tag{29}$$

*Furthermore, when $\mu = 0$, it holds that for any $r \geq 1$,*

$$\frac{r}{5} \leq \frac{A_r}{a_{r+1}} \leq r. \tag{30}$$

*Note that the inequalities on the right-hand side also hold when $r = 0$ since $\frac{A_0}{a_1} = 0$.*

*Proof.* From the definition of $a_{r+1}$, we have

$$a_{r+1} = \frac{1 + \mu A_r + \sqrt{(1+\mu A_r)^2 + 4\lambda A_r(1+\mu A_r)}}{2\lambda} \geq \frac{\mu A_r}{2\lambda},$$

$$a_{r+1} = \frac{1 + \mu A_r + \sqrt{(1+\mu A_r)^2 + 4\lambda A_r(1+\mu A_r)}}{2\lambda} \leq \frac{1+\mu A_r}{2\lambda}\left(1+\sqrt{1+\frac{4\lambda}{\mu}}\right).$$

When $\mu = 0$, we have

$$a_{r+1} = \frac{1 + \sqrt{1 + 4\lambda A_r}}{2\lambda} \geq \frac{A_r}{\sqrt{\lambda A_r}} \geq \frac{A_r}{r},$$

$$a_{r+1} = \frac{1 + \sqrt{1 + 4\lambda A_r}}{2\lambda} \leq A_r\sqrt{\frac{5}{\lambda A_r}} \leq \frac{5A_r}{r},$$

where we use Lemma 17 and 18. This concludes the statement. $\qquad\square$

**Lemma 20.** *For all $r \geq 0$, it holds that*

$$\frac{a_{r+2}}{a_{r+1}} \leq \frac{1}{2}\left(1 + \frac{\mu}{\lambda}\right)\left(1 + \frac{2\lambda}{\mu}\right)\left(1 + \sqrt{1 + \frac{4\lambda}{\mu}}\right). \tag{31}$$

*Furthermore, when $\mu = 0$, it holds that for all $r \geq 0$,*

$$1 \leq \frac{a_{r+2}}{a_{r+1}} \leq 4. \tag{32}$$

*Proof.* From Lemma 19, we get

$$a_{r+2} \overset{(29)}{\leq} \frac{A_{r+1}}{2}\left(1 + \frac{\mu}{\lambda}\right)\left(1 + \sqrt{1 + \frac{4\lambda}{\mu}}\right)$$

$$= \frac{A_r + a_{r+1}}{2}\left(1 + \frac{\mu}{\lambda}\right)\left(1 + \sqrt{1 + \frac{4\lambda}{\mu}}\right) \overset{(29)}{\leq} \frac{a_{r+1}}{2}\left(1 + \frac{2\lambda}{\mu}\right)\left(1 + \frac{\mu}{\lambda}\right)\left(1 + \sqrt{1 + \frac{4\lambda}{\mu}}\right).$$

When $\mu = 0$, we get

$$\frac{a_{r+1}}{a_r} = \sqrt{\frac{A_{r+1}}{A_r}} \leq \sqrt{\frac{(r+1)^2}{\frac{r^2}{4}}} = \frac{2(r+1)}{r} \leq 4,$$

$$\frac{a_{r+1}}{a_r} = \sqrt{\frac{A_{r+1}}{A_r}} \geq 1,$$

where we use $A_{r+1} = A_r + a_{r+1} \geq A_r$. This concludes the statement. $\qquad\square$

### F.2. Main Proof

**Lemma 21.** *Suppose that Assumptions 1 and 4 hold and the following inequality is satisfied:*

$$\sum_{i=1}^{n}\left\|\nabla F_{i,r}(\boldsymbol{x}_i^{(r+\frac{1}{2})})\right\|^2 \leq \frac{\lambda^2}{352}\sum_{i=1}^{n}\left\|\boldsymbol{x}_i^{(r+\frac{1}{2})} - \boldsymbol{y}_i^{(r)}\right\|^2.$$

*Then, when $M = 1$, it holds that*

$$A_r f(\bar{\boldsymbol{x}}^{(r)}) + a_{r+1}f(\boldsymbol{x}^\star) + \frac{B_r}{2n}\sum_{i=1}^{n}\left\|\boldsymbol{v}_i^{(r)} - \boldsymbol{x}^\star\right\|^2 + \frac{A_{r+1}}{\delta}\mathcal{E}^{(r)} + A_r\left(\frac{33\lambda}{16} + \delta\right)\Xi_{\boldsymbol{x}}^{(r)} + \delta a_{r+1}\Xi_{\boldsymbol{v}}^{(r)} \tag{33}$$

$$\geq A_{r+1}f(\bar{\boldsymbol{x}}^{(r+1)}) + \frac{B_{r+1}}{2n}\sum_{i=1}^{n}\left\|\boldsymbol{v}_i^{(r+1)} - \boldsymbol{x}^\star\right\|^2 + \frac{\mu A_{r+1}}{2\rho^2}\Xi_{\boldsymbol{x}}^{(r+1)} + \frac{B_{r+1}}{2}\left(\frac{1}{\rho^2} - 1\right)\Xi_{\boldsymbol{v}}^{(r+1)}$$

$$+ \frac{A_{r+1}}{n}\left(\frac{\lambda}{32} - 2\delta\right)\sum_{i=1}^{n}\left\|\boldsymbol{x}_i^{(r+\frac{1}{2})} - \boldsymbol{y}_i^{(r)}\right\|^2 + \frac{B_r}{12n}\sum_{i=1}^{n}\left\|\boldsymbol{v}_i^{(r+\frac{1}{2})} - \boldsymbol{v}_i^{(r)}\right\|^2 + \frac{\mu a_{r+1}}{2n}\sum_{i=1}^{n}\left\|\boldsymbol{x}_i^{(r+\frac{1}{2})} - \boldsymbol{v}_i^{(r+\frac{1}{2})}\right\|^2,$$

*where $\Xi_{\boldsymbol{x}}^{(r)} := \frac{1}{n}\sum_{i=1}^{n}\left\|\boldsymbol{x}_i^{(r)} - \bar{\boldsymbol{x}}^{(r)}\right\|^2$, $\Xi_{\boldsymbol{v}}^{(r)} := \frac{1}{n}\sum_{i=1}^{n}\left\|\boldsymbol{v}_i^{(r)} - \bar{\boldsymbol{v}}^{(r)}\right\|^2$, and $\mathcal{E}^{(r)} := \frac{1}{n}\sum_{i=1}^{n}\left\|\boldsymbol{h}_i^{(r)} - \nabla h_i(\bar{\boldsymbol{y}}^{(r)})\right\|^2$.*

*Proof.* We have

$$A_r f(\bar{\boldsymbol{x}}^{(r)}) + a_{r+1} f(\boldsymbol{x}^\star) + \frac{B_r}{2n} \sum_{i=1}^n \left\| \boldsymbol{v}_i^{(r)} - \boldsymbol{x}^\star \right\|^2$$

$$\overset{(1)}{\geq} \frac{A_{r+1}}{n} \sum_{i=1}^n f_i(\boldsymbol{x}_i^{(r+\frac{1}{2})}) + \frac{1}{n} \sum_{i=1}^n \left\langle \nabla f_i(\boldsymbol{x}_i^{(r)}), A_r \bar{\boldsymbol{x}}^{(r)} - A_{r+1} \boldsymbol{x}_i^{(r+\frac{1}{2})} + a_{r+1} \boldsymbol{x}^\star \right\rangle$$

$$+ \frac{B_r}{2n} \sum_{i=1}^n \left\| \boldsymbol{v}_i^{(r)} - \boldsymbol{x}^\star \right\|^2 + \frac{\mu A_r}{2n} \sum_{i=1}^n \left\| \boldsymbol{x}_i^{(r+\frac{1}{2})} - \bar{\boldsymbol{x}}^{(r)} \right\|^2 + \frac{\mu a_{r+1}}{2n} \sum_{i=1}^n \left\| \boldsymbol{x}_i^{(r+\frac{1}{2})} - \boldsymbol{x}^\star \right\|^2$$

$$= \frac{A_{r+1}}{n} \sum_{i=1}^n f_i(\boldsymbol{x}_i^{(r+\frac{1}{2})}) + \frac{1}{n} \sum_{i=1}^n \left\langle \nabla f_i(\boldsymbol{x}_i^{(r)}), A_r \bar{\boldsymbol{x}}^{(r)} - A_{r+1} \boldsymbol{x}_i^{(r+\frac{1}{2})} \right\rangle + \frac{\mu A_r}{2n} \sum_{i=1}^n \left\| \boldsymbol{x}_i^{(r+\frac{1}{2})} - \bar{\boldsymbol{x}}^{(r)} \right\|^2$$

$$+ \frac{a_{r+1}}{n} \sum_{i=1}^n \left( \left\langle \nabla f_i(\boldsymbol{x}_i^{(r)}) + \boldsymbol{h}_i^{(r)}, \boldsymbol{x}^\star \right\rangle + \frac{\mu}{2} \left\| \boldsymbol{x}_i^{(r+\frac{1}{2})} - \boldsymbol{x}^\star \right\|^2 \right) + \frac{B_r}{2n} \sum_{i=1}^n \left\| \boldsymbol{v}_i^{(r)} - \boldsymbol{x}^\star \right\|^2$$

$$\overset{(a)}{\geq} \frac{A_{r+1}}{n} \sum_{i=1}^n f_i(\boldsymbol{x}_i^{(r+\frac{1}{2})}) + \frac{1}{n} \sum_{i=1}^n \left\langle \nabla f_i(\boldsymbol{x}_i^{(r)}), A_r \bar{\boldsymbol{x}}^{(r)} - A_{r+1} \boldsymbol{x}_i^{(r+\frac{1}{2})} \right\rangle + \frac{\mu A_r}{2n} \sum_{i=1}^n \left\| \boldsymbol{x}_i^{(r+\frac{1}{2})} - \bar{\boldsymbol{x}}^{(r)} \right\|^2$$

$$+ \frac{a_{r+1}}{n} \sum_{i=1}^n \left( \left\langle \nabla f_i(\boldsymbol{x}_i^{(r)}) + \boldsymbol{h}_i^{(r)}, \boldsymbol{v}_i^{(r+\frac{1}{2})} \right\rangle + \frac{\mu}{2} \left\| \boldsymbol{x}_i^{(r+\frac{1}{2})} - \boldsymbol{v}_i^{(r+\frac{1}{2})} \right\|^2 \right) + \frac{B_r}{2n} \sum_{i=1}^n \left\| \boldsymbol{v}_i^{(r)} - \boldsymbol{v}_i^{(r+\frac{1}{2})} \right\|^2$$

$$+ \frac{B_{r+1}}{2n} \sum_{i=1}^n \left\| \boldsymbol{v}_i^{(r+\frac{1}{2})} - \boldsymbol{x}^\star \right\|^2,$$

where we use the fact that $a_{r+1}(\langle \nabla f_i(\boldsymbol{x}_i^{(r+\frac{1}{2})}) + \boldsymbol{h}_i^{(r)}, \boldsymbol{v} \rangle + \frac{\mu}{2} \|\boldsymbol{v} - \boldsymbol{x}_i^{(r+\frac{1}{2})}\|^2) + \frac{B_r}{2} \|\boldsymbol{v} - \boldsymbol{v}_i^{(r)}\|^2$ is $B_{r+1}$-strongly convex in (a). By using the above inequalities,

$$\frac{B_{r+1}}{2n} \sum_{i=1}^n \left\| \boldsymbol{v}_i^{(r+\frac{1}{2})} - \boldsymbol{x}^\star \right\|^2 \overset{(14)}{=} \frac{B_{r+1}}{2} \left\| \bar{\boldsymbol{v}}^{(r+1)} - \boldsymbol{x}^\star \right\|^2 + \frac{B_{r+1}}{2n} \sum_{i=1}^n \left\| \boldsymbol{v}_i^{(r+\frac{1}{2})} - \bar{\boldsymbol{v}}^{(r+1)} \right\|^2$$

$$\overset{(14)}{=} \frac{B_{r+1}}{2n} \sum_{i=1}^n \left\| \boldsymbol{v}_i^{(r+1)} - \boldsymbol{x}^\star \right\|^2 - \frac{B_{r+1}}{2n} \sum_{i=1}^n \left\| \boldsymbol{v}_i^{(r+1)} - \bar{\boldsymbol{v}}^{(r+1)} \right\|^2 + \frac{B_{r+1}}{2n} \sum_{i=1}^n \left\| \boldsymbol{v}_i^{(r+\frac{1}{2})} - \bar{\boldsymbol{v}}^{(r+1)} \right\|^2$$

$$\overset{(5)}{\geq} \frac{B_{r+1}}{2n} \sum_{i=1}^n \left\| \boldsymbol{v}_i^{(r+1)} - \boldsymbol{x}^\star \right\|^2 + \frac{B_{r+1}}{2n} \left( \frac{1}{\rho^2} - 1 \right) \sum_{i=1}^n \left\| \boldsymbol{v}_i^{(r+1)} - \bar{\boldsymbol{v}}^{(r+1)} \right\|^2,$$

we get

$$A_r f(\bar{\boldsymbol{x}}^{(r)}) + a_{r+1} f(\boldsymbol{x}^\star) + \frac{B_r}{2n} \sum_{i=1}^n \left\| \boldsymbol{v}_i^{(r)} - \boldsymbol{x}^\star \right\|^2$$

$$\geq \frac{A_{r+1}}{n} \sum_{i=1}^n f_i(\boldsymbol{x}_i^{(r+\frac{1}{2})}) + \frac{A_{r+1}}{n} \sum_{i=1}^n \left\langle \boldsymbol{h}_i^{(r)}, \boldsymbol{x}_i^{(r+\frac{1}{2})} \right\rangle + \frac{B_{r+1}}{2n} \sum_{i=1}^n \left\| \boldsymbol{v}_i^{(r+1)} - \boldsymbol{x}^\star \right\|^2$$

$$+ \underbrace{\frac{1}{n} \sum_{i=1}^n \left\langle \nabla f_i(\boldsymbol{x}_i^{(r)}) + \boldsymbol{h}_i^{(r)}, A_r \bar{\boldsymbol{x}}^{(r)} - A_{r+1} \boldsymbol{x}_i^{(r+\frac{1}{2})} + a_{r+1} \boldsymbol{v}_i^{(r+\frac{1}{2})} \right\rangle}_{\mathcal{T}^\star} + \frac{\mu A_r}{2n} \sum_{i=1}^n \left\| \boldsymbol{x}_i^{(r+\frac{1}{2})} - \bar{\boldsymbol{x}}^{(r)} \right\|^2$$

$$+ \frac{\mu a_{r+1}}{2n} \sum_{i=1}^n \left\| \boldsymbol{x}_i^{(r+\frac{1}{2})} - \boldsymbol{v}_i^{(r+\frac{1}{2})} \right\|^2 + \frac{B_r}{2n} \sum_{i=1}^n \left\| \boldsymbol{v}_i^{(r)} - \boldsymbol{v}_i^{(r+\frac{1}{2})} \right\|^2$$

$$+ \frac{B_{r+1}}{2n} \left( \frac{1}{\rho^2} - 1 \right) \sum_{i=1}^n \left\| \boldsymbol{v}_i^{(r+1)} - \bar{\boldsymbol{v}}^{(r+1)} \right\|^2.$$

From the definition of $F_{i,r}$, we have

$$\nabla F_{i,r}(\boldsymbol{x}) = \nabla f_i(\boldsymbol{x}) + \boldsymbol{h}_i^{(r)} + \lambda(\boldsymbol{x} - \boldsymbol{y}_i^{(r)}).$$

Using the above equality, we get

$$\mathcal{T}^{\star} = \frac{1}{n} \sum_{i=1}^{n} \left\langle \nabla F_{i,r}(\boldsymbol{x}_i^{(r+\frac{1}{2})}) + \lambda \left( \boldsymbol{y}_i^{(r)} - \boldsymbol{x}_i^{(r+\frac{1}{2})} \right), A_r \bar{\boldsymbol{x}}^{(r)} - A_{r+1} \boldsymbol{x}_i^{(r+\frac{1}{2})} + a_{r+1} \boldsymbol{v}_i^{(r+\frac{1}{2})} \right\rangle$$

$$= \frac{1}{n} \sum_{i=1}^{n} \left\langle \nabla F_{i,r}(\boldsymbol{x}_i^{(r+\frac{1}{2})}) + \lambda \left( \boldsymbol{y}_i^{(r)} - \boldsymbol{x}_i^{(r+\frac{1}{2})} \right), A_r \left( \bar{\boldsymbol{x}}^{(r)} - \boldsymbol{x}_i^{(r)} \right) + A_{r+1} \left( \boldsymbol{y}_i^{(r)} - \boldsymbol{x}_i^{(r+\frac{1}{2})} \right) + a_{r+1} \left( \boldsymbol{v}_i^{(r+\frac{1}{2})} - \boldsymbol{v}_i^{(r)} \right) \right\rangle$$

$$= \frac{1}{n} \sum_{i=1}^{n} \left\langle \nabla F_{i,r}(\boldsymbol{x}_i^{(r+\frac{1}{2})}), A_r \left( \bar{\boldsymbol{x}}^{(r)} - \boldsymbol{x}_i^{(r)} \right) + A_{r+1} \left( \boldsymbol{y}_i^{(r)} - \boldsymbol{x}_i^{(r+\frac{1}{2})} \right) + a_{r+1} \left( \boldsymbol{v}_i^{(r+\frac{1}{2})} - \boldsymbol{v}_i^{(r)} \right) \right\rangle$$

$$+ \frac{\lambda}{n} \sum_{i=1}^{n} \left\langle \boldsymbol{y}_i^{(r)} - \boldsymbol{x}_i^{(r+\frac{1}{2})}, A_r \left( \bar{\boldsymbol{x}}^{(r)} - \boldsymbol{x}_i^{(r)} \right) + a_{r+1} \left( \boldsymbol{v}_i^{(r+\frac{1}{2})} - \boldsymbol{v}_i^{(r)} \right) \right\rangle + \frac{\lambda A_{r+1}}{n} \sum_{i=1}^{n} \left\| \boldsymbol{x}_i^{(r+\frac{1}{2})} - \boldsymbol{y}_i^{(r)} \right\|^2$$

$$\overset{(11)}{\geq} \frac{1}{n} \sum_{i=1}^{n} \left\langle \nabla F_{i,r}(\boldsymbol{x}_i^{(r+\frac{1}{2})}), A_r \left( \bar{\boldsymbol{x}}^{(r)} - \boldsymbol{x}_i^{(r)} \right) + A_{r+1} \left( \boldsymbol{y}_i^{(r)} - \boldsymbol{x}_i^{(r+\frac{1}{2})} \right) + a_{r+1} \left( \boldsymbol{v}_i^{(r+\frac{1}{2})} - \boldsymbol{v}_i^{(r)} \right) \right\rangle$$

$$- \frac{2\lambda A_r}{n} \sum_{i=1}^{n} \left\| \bar{\boldsymbol{x}}^{(r)} - \boldsymbol{x}_i^{(r)} \right\|^2 - \frac{B_r}{3n} \sum_{i=1}^{n} \left\| \boldsymbol{v}_i^{(r+\frac{1}{2})} - \boldsymbol{v}_i^{(r)} \right\|^2 + \frac{\lambda A_{r+1}}{8n} \sum_{i=1}^{n} \left\| \boldsymbol{x}_i^{(r+\frac{1}{2})} - \boldsymbol{y}_i^{(r)} \right\|^2 .$$

We have

$$A_r \left\langle \nabla F_{i,r}(\boldsymbol{x}_i^{(r+\frac{1}{2})}), \bar{\boldsymbol{x}}^{(r)} - \boldsymbol{x}_i^{(r)} \right\rangle \overset{(11)}{\geq} -\frac{4A_r}{\lambda} \left\| \nabla F_{i,r}(\boldsymbol{x}_i^{(r+\frac{1}{2})}) \right\|^2 - \frac{\lambda A_r}{16} \left\| \bar{\boldsymbol{x}}^{(r)} - \boldsymbol{x}_i^{(r)} \right\|^2 ,$$

$$A_{r+1} \left\langle \nabla F_{i,r}(\boldsymbol{x}_i^{(r+\frac{1}{2})}), \boldsymbol{y}_i^{(r)} - \boldsymbol{x}_i^{(r+\frac{1}{2})} \right\rangle \overset{(11)}{\geq} -\frac{4A_{r+1}}{\lambda} \left\| \nabla F_{i,r}(\boldsymbol{x}_i^{(r+\frac{1}{2})}) \right\|^2 - \frac{\lambda A_{r+1}}{16} \left\| \boldsymbol{x}_i^{(r+\frac{1}{2})} - \boldsymbol{y}_i^{(r)} \right\|^2 ,$$

$$a_{r+1} \left\langle \nabla F_{i,r}(\boldsymbol{x}_i^{(r+\frac{1}{2})}), \boldsymbol{v}_i^{(r+\frac{1}{2})} - \boldsymbol{v}_i^{(r)} \right\rangle \overset{(11)}{\geq} -\frac{3A_{r+1}}{\lambda} \left\| \nabla F_{i,r}(\boldsymbol{x}_i^{(r+\frac{1}{2})}) \right\|^2 - \frac{B_r}{12} \left\| \boldsymbol{v}_i^{(r+\frac{1}{2})} - \boldsymbol{v}_i^{(r)} \right\|^2 ,$$

where we use $\lambda = A_{r+1} B_r a_{r+1}^{-2}$ in the last inequality. By using the above inequalities, we get

$$\mathcal{T}^{\star} \geq -\frac{11 A_{r+1}}{\lambda} \left\| \nabla F_{i,r}(\boldsymbol{x}_i^{(r+\frac{1}{2})}) \right\|^2$$

$$- \frac{33\lambda A_r}{16n} \sum_{i=1}^{n} \left\| \bar{\boldsymbol{x}}^{(r)} - \boldsymbol{x}_i^{(r)} \right\|^2 - \frac{5B_r}{12n} \sum_{i=1}^{n} \left\| \boldsymbol{v}_i^{(r+\frac{1}{2})} - \boldsymbol{v}_i^{(r)} \right\|^2 + \frac{\lambda A_{r+1}}{16n} \sum_{i=1}^{n} \left\| \boldsymbol{x}_i^{(r+\frac{1}{2})} - \boldsymbol{y}_i^{(r)} \right\|^2 .$$

Then, using $\frac{1}{n} \sum_{i=1}^{n} \left\| \nabla F_{i,r}(\boldsymbol{x}_i^{(r+\frac{1}{2})}) \right\|^2 \leq \frac{\lambda^2}{352n} \sum_{i=1}^{n} \left\| \boldsymbol{x}_i^{(r+\frac{1}{2})} - \boldsymbol{y}_i^{(r)} \right\|^2$, we obtain

$$\mathcal{T}^{\star} \geq \frac{\lambda A_{r+1}}{32n} \sum_{i=1}^{n} \left\| \boldsymbol{x}_i^{(r+\frac{1}{2})} - \boldsymbol{y}_i^{(r)} \right\|^2 - \frac{33\lambda A_r}{16n} \sum_{i=1}^{n} \left\| \bar{\boldsymbol{x}}^{(r)} - \boldsymbol{x}_i^{(r)} \right\|^2 - \frac{5B_r}{12n} \sum_{i=1}^{n} \left\| \boldsymbol{v}_i^{(r+\frac{1}{2})} - \boldsymbol{v}_i^{(r)} \right\|^2 .$$

We have

$$A_r f(\bar{\boldsymbol{x}}^{(r)}) + a_{r+1} f(\boldsymbol{x}^{\star}) + \frac{B_r}{2n} \sum_{i=1}^{n} \left\| \boldsymbol{v}_i^{(r)} - \boldsymbol{x}^{\star} \right\|^2$$

$$\geq \frac{A_{r+1}}{n} \sum_{i=1}^{n} f_i(\boldsymbol{x}_i^{(r+\frac{1}{2})}) + \frac{A_{r+1}}{n} \sum_{i=1}^{n} \left\langle \boldsymbol{h}_i^{(r)}, \boldsymbol{x}_i^{(r+\frac{1}{2})} \right\rangle + \frac{B_{r+1}}{2n} \sum_{i=1}^{n} \left\| \boldsymbol{v}_i^{(r+1)} - \boldsymbol{x}^{\star} \right\|^2$$

$$+ \frac{\lambda A_{r+1}}{32n} \sum_{i=1}^{n} \left\| \boldsymbol{x}_i^{(r+\frac{1}{2})} - \boldsymbol{y}_i^{(r)} \right\|^2 - \frac{33\lambda A_r}{16n} \sum_{i=1}^{n} \left\| \bar{\boldsymbol{x}}^{(r)} - \boldsymbol{x}_i^{(r)} \right\|^2 + \frac{\mu A_r}{2n} \sum_{i=1}^{n} \left\| \boldsymbol{x}_i^{(r+\frac{1}{2})} - \bar{\boldsymbol{x}}^{(r)} \right\|^2$$

$$+ \frac{\mu a_{r+1}}{2n} \sum_{i=1}^{n} \left\| \boldsymbol{x}_i^{(r+\frac{1}{2})} - \boldsymbol{v}_i^{(r+\frac{1}{2})} \right\|^2 + \frac{B_r}{12n} \sum_{i=1}^{n} \left\| \boldsymbol{v}_i^{(r)} - \boldsymbol{v}_i^{(r+\frac{1}{2})} \right\|^2$$

$$+ \frac{B_{r+1}}{2n} \left( \frac{1}{\rho^2} - 1 \right) \sum_{i=1}^{n} \left\| \boldsymbol{v}_i^{(r+1)} - \bar{\boldsymbol{v}}^{(r+1)} \right\|^2 .$$

Using the following inequality,

$$\frac{A_{r+1}}{n} \sum_{i=1}^{n} f_i(\boldsymbol{x}_i^{(r+\frac{1}{2})})$$

$$\geq A_{r+1} f(\bar{\boldsymbol{x}}^{(r+1)}) + \frac{A_{r+1}}{n} \sum_{i=1}^{n} \left\langle \nabla f_i(\bar{\boldsymbol{x}}^{(r+1)}), \boldsymbol{x}_i^{(r+\frac{1}{2})} - \bar{\boldsymbol{x}}^{(r+1)} \right\rangle + \frac{\mu A_{r+1}}{2n} \sum_{i=1}^{n} \left\| \boldsymbol{x}_i^{(r+\frac{1}{2})} - \bar{\boldsymbol{x}}^{(r+1)} \right\|^2.$$

we get

$$A_r f(\bar{\boldsymbol{x}}^{(r)}) + a_{r+1} f(\boldsymbol{x}^\star) + \frac{B_r}{2n} \sum_{i=1}^{n} \left\| \boldsymbol{v}_i^{(r)} - \boldsymbol{x}^\star \right\|^2$$

$$\geq A_{r+1} f(\bar{\boldsymbol{x}}^{(r+1)}) + \underbrace{\frac{A_{r+1}}{n} \sum_{i=1}^{n} \left\langle \nabla f_i(\bar{\boldsymbol{x}}^{(r+1)}) + \boldsymbol{h}_i^{(r)}, \boldsymbol{x}_i^{(r+\frac{1}{2})} - \bar{\boldsymbol{x}}^{(r+1)} \right\rangle}_{\mathcal{T}^{\star\star}}$$

$$+ \frac{\mu A_{r+1}}{2n} \sum_{i=1}^{n} \left\| \boldsymbol{x}_i^{(r+\frac{1}{2})} - \bar{\boldsymbol{x}}^{(r+1)} \right\|^2 + \frac{B_{r+1}}{2n} \sum_{i=1}^{n} \left\| \boldsymbol{v}_i^{(r+1)} - \boldsymbol{x}^\star \right\|^2$$

$$+ \frac{\lambda A_{r+1}}{32n} \sum_{i=1}^{n} \left\| \boldsymbol{x}_i^{(r+\frac{1}{2})} - \boldsymbol{y}_i^{(r)} \right\|^2 - \frac{33\lambda A_r}{16n} \sum_{i=1}^{n} \left\| \bar{\boldsymbol{x}}^{(r)} - \boldsymbol{x}_i^{(r)} \right\|^2 + \frac{\mu A_r}{2n} \sum_{i=1}^{n} \left\| \boldsymbol{x}_i^{(r+\frac{1}{2})} - \bar{\boldsymbol{x}}^{(r)} \right\|^2$$

$$+ \frac{\mu a_{r+1}}{2n} \sum_{i=1}^{n} \left\| \boldsymbol{x}_i^{(r+\frac{1}{2})} - \boldsymbol{v}_i^{(r+\frac{1}{2})} \right\|^2 + \frac{B_r}{12n} \sum_{i=1}^{n} \left\| \boldsymbol{v}_i^{(r)} - \boldsymbol{v}_i^{(r+\frac{1}{2})} \right\|^2$$

$$+ \frac{B_{r+1}}{2n} \left( \frac{1}{\rho^2} - 1 \right) \sum_{i=1}^{n} \left\| \boldsymbol{v}_i^{(r+1)} - \bar{\boldsymbol{v}}^{(r+1)} \right\|^2.$$

Using the following inequality, we obtain

$$\mathcal{T}^{\star\star} = \frac{A_{r+1}}{n} \sum_{i=1}^{n} \left\langle \nabla f_i(\bar{\boldsymbol{x}}^{(r+1)}) + \boldsymbol{h}_i^{(r)} - \nabla f(\bar{\boldsymbol{x}}^{(r+1)}), \boldsymbol{x}_i^{(r+\frac{1}{2})} - \bar{\boldsymbol{y}}^{(r)} \right\rangle$$

$$\overset{(11)}{\geq} -\frac{A_{r+1}}{2\delta n} \sum_{i=1}^{n} \left\| \nabla f_i(\bar{\boldsymbol{x}}^{(r+1)}) + \boldsymbol{h}_i^{(r)} - \nabla f(\bar{\boldsymbol{x}}^{(r+1)}) \right\|^2 - \frac{\delta A_{r+1}}{2n} \sum_{i=1}^{n} \left\| \boldsymbol{x}_i^{(r+\frac{1}{2})} - \bar{\boldsymbol{y}}^{(r)} \right\|^2$$

$$\overset{(12),(3)}{\geq} -\delta A_{r+1} \left\| \bar{\boldsymbol{x}}^{(r+1)} - \bar{\boldsymbol{y}}^{(r)} \right\|^2 - \frac{A_{r+1}}{\delta n} \sum_{i=1}^{n} \left\| \nabla f_i(\bar{\boldsymbol{y}}^{(r)}) + \boldsymbol{h}_i^{(r)} - \nabla f(\bar{\boldsymbol{y}}^{(r)}) \right\|^2 - \frac{\delta A_{r+1}}{2n} \sum_{i=1}^{n} \left\| \boldsymbol{x}_i^{(r+\frac{1}{2})} - \bar{\boldsymbol{y}}^{(r)} \right\|^2$$

$$\overset{(13)}{\geq} -\frac{2\delta A_{r+1}}{n} \sum_{i=1}^{n} \left\| \boldsymbol{x}_i^{(r+\frac{1}{2})} - \boldsymbol{y}_i^{(r)} \right\|^2 - \frac{A_{r+1}}{\delta} \mathcal{E}^{(r)} - \frac{\delta A_{r+1}}{n} \sum_{i=1}^{n} \left\| \boldsymbol{y}_i^{(r)} - \bar{\boldsymbol{y}}^{(r)} \right\|^2$$

$$\geq -\frac{2\delta A_{r+1}}{n} \sum_{i=1}^{n} \left\| \boldsymbol{x}_i^{(r+\frac{1}{2})} - \boldsymbol{y}_i^{(r)} \right\|^2 - \frac{A_{r+1}}{\delta} \mathcal{E}^{(r)} - \frac{\delta A_r}{n} \sum_{i=1}^{n} \left\| \boldsymbol{x}_i^{(r)} - \bar{\boldsymbol{x}}^{(r)} \right\|^2 - \frac{\delta a_{r+1}}{n} \sum_{i=1}^{n} \left\| \boldsymbol{v}_i^{(r)} - \bar{\boldsymbol{v}}^{(r)} \right\|^2,$$

where we use Jensen's inequality in the last inequality. Using the above inequality, we get

$$A_r f(\bar{\boldsymbol{x}}^{(r)}) + a_{r+1} f(\boldsymbol{x}^\star) + \frac{B_r}{2n} \sum_{i=1}^n \left\| \boldsymbol{v}_i^{(r)} - \boldsymbol{x}^\star \right\|^2 + \frac{A_{r+1}}{\delta} \mathcal{E}^{(r)}$$

$$\geq A_{r+1} f(\bar{\boldsymbol{x}}^{(r+1)}) + \frac{B_{r+1}}{2n} \sum_{i=1}^n \left\| \boldsymbol{v}_i^{(r+1)} - \boldsymbol{x}^\star \right\|^2$$

$$- \frac{\delta a_{r+1}}{n} \sum_{i=1}^n \left\| \boldsymbol{v}_i^{(r)} - \bar{\boldsymbol{v}}^{(r)} \right\|^2 + \frac{\mu A_{r+1}}{2n} \sum_{i=1}^n \left\| \boldsymbol{x}_i^{(r+\frac{1}{2})} - \bar{\boldsymbol{x}}^{(r+1)} \right\|^2$$

$$+ \frac{A_{r+1}}{n} \left( \frac{\lambda}{32} - 2\delta \right) \sum_{i=1}^n \left\| \boldsymbol{x}_i^{(r+\frac{1}{2})} - \boldsymbol{y}_i^{(r)} \right\|^2 - \left( \frac{33\lambda A_r}{16n} + \frac{\delta A_r}{n} \right) \sum_{i=1}^n \left\| \boldsymbol{x}_i^{(r)} - \bar{\boldsymbol{x}}^{(r)} \right\|^2 + \frac{\mu A_r}{2n} \sum_{i=1}^n \left\| \boldsymbol{x}_i^{(r+\frac{1}{2})} - \bar{\boldsymbol{x}}^{(r)} \right\|^2$$

$$+ \frac{\mu a_{r+1}}{2n} \sum_{i=1}^n \left\| \boldsymbol{x}_i^{(r+\frac{1}{2})} - \boldsymbol{v}_i^{(r+\frac{1}{2})} \right\|^2 + \frac{B_r}{12n} \sum_{i=1}^n \left\| \boldsymbol{v}_i^{(r+\frac{1}{2})} - \boldsymbol{v}_i^{(r)} \right\|^2$$

$$+ \frac{B_{r+1}}{2n} \left( \frac{1}{\rho^2} - 1 \right) \sum_{i=1}^n \left\| \boldsymbol{v}_i^{(r+1)} - \bar{\boldsymbol{v}}^{(r+1)} \right\|^2$$

$$\geq A_{r+1} f(\bar{\boldsymbol{x}}^{(r+1)}) + \frac{B_{r+1}}{2n} \sum_{i=1}^n \left\| \boldsymbol{v}_i^{(r+1)} - \boldsymbol{x}^\star \right\|^2$$

$$- \frac{\delta a_{r+1}}{n} \sum_{i=1}^n \left\| \boldsymbol{v}_i^{(r)} - \bar{\boldsymbol{v}}^{(r)} \right\|^2 + \frac{\mu A_{r+1}}{2\rho^2 n} \sum_{i=1}^n \left\| \boldsymbol{x}_i^{(r+1)} - \bar{\boldsymbol{x}}^{(r+1)} \right\|^2$$

$$+ \frac{A_{r+1}}{n} \left( \frac{\lambda}{32} - 2\delta \right) \sum_{i=1}^n \left\| \boldsymbol{x}_i^{(r+\frac{1}{2})} - \boldsymbol{y}_i^{(r)} \right\|^2 - \left( \frac{33\lambda A_r}{16n} + \frac{\delta A_r}{n} \right) \sum_{i=1}^n \left\| \boldsymbol{x}_i^{(r)} - \bar{\boldsymbol{x}}^{(r)} \right\|^2 + \frac{\mu A_r}{2n} \sum_{i=1}^n \left\| \boldsymbol{x}_i^{(r+\frac{1}{2})} - \bar{\boldsymbol{x}}^{(r)} \right\|^2$$

$$+ \frac{\mu a_{r+1}}{2n} \sum_{i=1}^n \left\| \boldsymbol{x}_i^{(r+\frac{1}{2})} - \boldsymbol{v}_i^{(r+\frac{1}{2})} \right\|^2 + \frac{B_r}{12n} \sum_{i=1}^n \left\| \boldsymbol{v}_i^{(r+\frac{1}{2})} - \boldsymbol{v}_i^{(r)} \right\|^2$$

$$+ \frac{B_{r+1}}{2n} \left( \frac{1}{\rho^2} - 1 \right) \sum_{i=1}^n \left\| \boldsymbol{v}_i^{(r+1)} - \bar{\boldsymbol{v}}^{(r+1)} \right\|^2$$

By summarizing the above inequality, we obtain

$$A_r f(\bar{\boldsymbol{x}}^{(r)}) + a_{r+1} f(\boldsymbol{x}^\star) + \frac{B_r}{2n} \sum_{i=1}^n \left\| \boldsymbol{v}_i^{(r)} - \boldsymbol{x}^\star \right\|^2 + \frac{A_{r+1}}{\delta} \mathcal{E}^{(r)}$$

$$\geq A_{r+1} f(\bar{\boldsymbol{x}}^{(r+1)}) + \frac{B_{r+1}}{2n} \sum_{i=1}^n \left\| \boldsymbol{v}_i^{(r+1)} - \boldsymbol{x}^\star \right\|^2$$

$$+ \frac{A_{r+1}}{n} \left( \frac{\lambda}{32} - 2\delta \right) \sum_{i=1}^n \left\| \boldsymbol{x}_i^{(r+\frac{1}{2})} - \boldsymbol{y}_i^{(r)} \right\|^2 + \frac{B_r}{12n} \sum_{i=1}^n \left\| \boldsymbol{v}_i^{(r+\frac{1}{2})} - \boldsymbol{v}_i^{(r)} \right\|^2 + \frac{B_{r+1}}{2n} \left( \frac{1}{\rho^2} - 1 \right) \sum_{i=1}^n \left\| \boldsymbol{v}_i^{(r+1)} - \bar{\boldsymbol{v}}^{(r+1)} \right\|^2$$

$$- \left( \frac{33\lambda A_r}{16n} + \frac{\delta A_r}{n} \right) \sum_{i=1}^n \left\| \boldsymbol{x}_i^{(r)} - \bar{\boldsymbol{x}}^{(r)} \right\|^2 - \frac{\delta a_{r+1}}{n} \sum_{i=1}^n \left\| \boldsymbol{v}_i^{(r)} - \bar{\boldsymbol{v}}^{(r)} \right\|^2 + \frac{\mu A_{r+1}}{2\rho^2 n} \sum_{i=1}^n \left\| \boldsymbol{x}_i^{(r+1)} - \bar{\boldsymbol{x}}^{(r+1)} \right\|^2$$

$$+ \frac{\mu a_{r+1}}{2n} \sum_{i=1}^n \left\| \boldsymbol{x}_i^{(r+\frac{1}{2})} - \boldsymbol{v}_i^{(r+\frac{1}{2})} \right\|^2.$$

This concludes the statement. $\qquad\square$

**Lemma 22.** *Suppose that Assumptions 2 and 5 hold. Then, when $M = 1$, it holds that*

$$
4\rho^2 \mathcal{E}^{(r)} \geq \mathcal{E}^{(r+1)} - 8\rho^2\delta^2 \frac{a_{r+2}}{A_{r+2}} \left\| \bar{\boldsymbol{v}}^{(r+1)} - \bar{\boldsymbol{x}}^{(r+1)} \right\|^2
$$

$$
- 8\rho^2\delta^2 \frac{1}{n} \sum_{i=1}^{n} \left\| \boldsymbol{x}_i^{(r+\frac{1}{2})} - \boldsymbol{y}_i^{(r)} \right\|^2 - 32L^2 \frac{A_{r+1}}{A_{r+2}} \Xi_{\boldsymbol{x}}^{(r+1)} - 32L^2 \frac{a_{r+2}}{A_{r+2}} \Xi_{\boldsymbol{v}}^{(r+1)}, \tag{34}
$$

*where* $\Xi_{\boldsymbol{x}}^{(r)} := \frac{1}{n} \sum_{i=1}^{n} \left\| \boldsymbol{x}_i^{(r)} - \bar{\boldsymbol{x}}^{(r)} \right\|^2$, $\Xi_{\boldsymbol{v}}^{(r)} := \frac{1}{n} \sum_{i=1}^{n} \left\| \boldsymbol{v}_i^{(r)} - \bar{\boldsymbol{v}}^{(r)} \right\|^2$, *and* $\mathcal{E}^{(r)} := \frac{1}{n} \sum_{i=1}^{n} \left\| \boldsymbol{h}_i^{(r)} - \nabla h_i(\bar{\boldsymbol{y}}^{(r)}) \right\|^2$.

*Proof.* From Lemma 8, we have

$$
\mathcal{E}^{(r+1)} \leq 4\rho^2 \mathcal{E}^{(r)} + 4\rho^2\delta^2 \left\| \bar{\boldsymbol{y}}^{(r+1)} - \bar{\boldsymbol{y}}^{(r)} \right\|^2 + 32L^2 \Xi_{\boldsymbol{y}}^{(r+1)}, \tag{35}
$$

Using $\bar{\boldsymbol{y}}^{(r+1)} = \frac{A_{r+1}\bar{\boldsymbol{x}}^{(r+1)} + a_{r+2}\bar{\boldsymbol{v}}^{(r+1)}}{A_{r+2}}$, we have

$$
\mathcal{E}^{(r+1)} \leq 4\rho^2 \mathcal{E}^{(r)} + 8\rho^2\delta^2 \left( \frac{a_{r+2}}{A_{r+2}} \right)^2 \left\| \bar{\boldsymbol{v}}^{(r+1)} - \bar{\boldsymbol{x}}^{(r+1)} \right\|^2
$$

$$
+ 8\rho^2\delta^2 \left\| \bar{\boldsymbol{x}}^{(r+1)} - \bar{\boldsymbol{y}}^{(r)} \right\|^2 + 32L^2 \frac{A_{r+1}}{A_{r+2}} \Xi_{\boldsymbol{x}}^{(r+1)} + 32L^2 \frac{a_{r+2}}{A_{r+2}} \Xi_{\boldsymbol{v}}^{(r+1)}
$$

$$
\leq 4\rho^2 \mathcal{E}^{(r)} + 8\rho^2\delta^2 \frac{a_{r+2}}{A_{r+2}} \left\| \bar{\boldsymbol{v}}^{(r+1)} - \bar{\boldsymbol{x}}^{(r+1)} \right\|^2
$$

$$
+ 8\rho^2\delta^2 \left\| \bar{\boldsymbol{x}}^{(r+1)} - \bar{\boldsymbol{y}}^{(r)} \right\|^2 + 32L^2 \frac{A_{r+1}}{A_{r+2}} \Xi_{\boldsymbol{x}}^{(r+1)} + 32L^2 \frac{a_{r+2}}{A_{r+2}} \Xi_{\boldsymbol{v}}^{(r+1)}
$$

$$
\leq 4\rho^2 \mathcal{E}^{(r)} + 8\rho^2\delta^2 \frac{a_{r+2}}{A_{r+2}} \left\| \bar{\boldsymbol{v}}^{(r+1)} - \bar{\boldsymbol{x}}^{(r+1)} \right\|^2
$$

$$
+ 8\rho^2\delta^2 \frac{1}{n} \sum_{i=1}^{n} \left\| \boldsymbol{x}_i^{(r+\frac{1}{2})} - \boldsymbol{y}_i^{(r)} \right\|^2 + 32L^2 \frac{A_{r+1}}{A_{r+2}} \Xi_{\boldsymbol{x}}^{(r+1)} + 32L^2 \frac{a_{r+2}}{A_{r+2}} \Xi_{\boldsymbol{v}}^{(r+1)}.
$$

This concludes the statement. $\qquad\square$

**Lemma 23.** *Suppose that Assumptions 1 and 4 hold. Then, when $M = 1$, it holds that*

$$
\left\| \bar{\boldsymbol{v}}^{(r+1)} - \bar{\boldsymbol{x}}^{(r+1)} \right\|^2 \tag{36}
$$

$$
\leq \frac{A_r}{A_{r+1}} \left\| \bar{\boldsymbol{v}}^{(r)} - \bar{\boldsymbol{x}}^{(r)} \right\|^2 + \frac{4A_{r+1}}{a_{r+1}} \frac{1}{n} \sum_{i=1}^{n} \left\| \boldsymbol{x}_i^{(r+\frac{1}{2})} - \boldsymbol{y}_i^{(r)} \right\|^2 + \frac{2A_{r+1}}{a_{r+1}} \frac{1}{n} \sum_{i=1}^{n} \left\| \boldsymbol{v}_i^{(r+\frac{1}{2})} - \boldsymbol{v}_i^{(r)} \right\|^2.
$$

*Proof.* For all $\gamma_1, \gamma_2 > 0$, it holds that

$$
\left\| \bar{\boldsymbol{v}}^{(r+1)} - \bar{\boldsymbol{x}}^{(r+1)} \right\|^2
$$

$$
\overset{(12)}{\leq} (1 + \gamma_1) \left\| \bar{\boldsymbol{v}}^{(r)} - \bar{\boldsymbol{x}}^{(r+1)} \right\|^2 + \left( 1 + \frac{1}{\gamma_1} \right) \left\| \bar{\boldsymbol{v}}^{(r+1)} - \bar{\boldsymbol{v}}^{(r)} \right\|^2
$$

$$
\overset{(12)}{\leq} (1 + \gamma_1)(1 + \gamma_2) \left\| \bar{\boldsymbol{v}}^{(r)} - \bar{\boldsymbol{y}}^{(r)} \right\|^2 + (1 + \gamma_1) \left( 1 + \frac{1}{\gamma_2} \right) \left\| \bar{\boldsymbol{x}}^{(r+1)} - \bar{\boldsymbol{y}}^{(r)} \right\|^2 + \left( 1 + \frac{1}{\gamma_1} \right) \left\| \bar{\boldsymbol{v}}^{(r+1)} - \bar{\boldsymbol{v}}^{(r)} \right\|^2
$$

$$
\leq (1 + \gamma_1)(1 + \gamma_2) \left( \frac{A_r}{A_{r+1}} \right)^2 \left\| \bar{\boldsymbol{v}}^{(r)} - \bar{\boldsymbol{x}}^{(r)} \right\|^2 + (1 + \gamma_1) \left( 1 + \frac{1}{\gamma_2} \right) \left\| \bar{\boldsymbol{x}}^{(r+1)} - \bar{\boldsymbol{y}}^{(r)} \right\|^2 + \left( 1 + \frac{1}{\gamma_1} \right) \left\| \bar{\boldsymbol{v}}^{(r+1)} - \bar{\boldsymbol{v}}^{(r)} \right\|^2.
$$

Substituting $\gamma_1 = \gamma_2 = \min\{1, \sqrt{\frac{A_{r+1}}{A_r}} - 1\}$, we obtain

$$(1 + \gamma_1)(1 + \gamma_2) \le \frac{A_{r+1}}{A_r},$$

$$\left(1 + \frac{1}{\gamma_1}\right) = \max\left\{2, \frac{\sqrt{A_{r+1}}\left(\sqrt{A_{r+1}} + \sqrt{A_r}\right)}{A_{r+1} - A_r}\right\} \le \max\left\{2, \frac{2A_{r+1}}{a_{r+1}}\right\} = \frac{2A_{r+1}}{a_{r+1}},$$

$$(1 + \gamma_1)\left(1 + \frac{1}{\gamma_2}\right) \le \max\left\{4, \frac{4A_{r+1}}{a_{r+1}}\right\} = \frac{4A_{r+1}}{a_{r+1}},$$

where we use $A_{r+1} \ge A_r$ and $A_{r+1} \ge a_{r+1}$. Using the above inequalities, we obtain the statement. $\square$

**Lemma 24.** *Suppose that Assumption 4 holds. Then, when $M = 1$, it holds that*

$$A_{r+1}\Xi_{\boldsymbol{x}}^{(r+1)} \le \rho A_r \Xi_{\boldsymbol{x}}^{(r)} + \frac{\rho^2 A_{r+1}}{1-\rho}\frac{1}{n}\sum_{i=1}^{n}\left\|\boldsymbol{x}_i^{(r+\frac{1}{2})} - \boldsymbol{y}_i^{(r)}\right\|^2 + \frac{\rho^2 a_{r+1}}{1-\rho}\Xi_{\boldsymbol{v}}^{(r)}, \tag{37}$$

*where $\Xi_{\boldsymbol{x}}^{(r)} := \frac{1}{n}\sum_{i=1}^{n}\left\|\boldsymbol{x}_i^{(r)} - \bar{\boldsymbol{x}}^{(r)}\right\|^2$ and $\Xi_{\boldsymbol{v}}^{(r)} := \frac{1}{n}\sum_{i=1}^{n}\left\|\boldsymbol{v}_i^{(r)} - \bar{\boldsymbol{v}}^{(r)}\right\|^2$.*

*Proof.* It holds that for all $\gamma > 0$,

$$\frac{1}{n}\sum_{i=1}^{n}\left\|\boldsymbol{x}_i^{(r+1)} - \bar{\boldsymbol{x}}^{(r+1)}\right\|^2$$

$$\overset{(5)}{\le} \frac{\rho^2}{n}\sum_{i=1}^{n}\left\|\boldsymbol{x}_i^{(r+\frac{1}{2})} - \bar{\boldsymbol{x}}^{(r+1)}\right\|^2$$

$$\overset{(14)}{=} \frac{\rho^2}{n}\sum_{i=1}^{n}\left\|\boldsymbol{x}_i^{(r+\frac{1}{2})} - \bar{\boldsymbol{y}}^{(r)}\right\|^2 - \rho^2\left\|\bar{\boldsymbol{x}}^{(r+1)} - \bar{\boldsymbol{y}}^{(r)}\right\|^2$$

$$\overset{(12)}{\le} \frac{(1+\gamma)\rho^2}{n}\sum_{i=1}^{n}\left\|\boldsymbol{x}_i^{(r+\frac{1}{2})} - \boldsymbol{y}_i^{(r)}\right\|^2 + \left(1 + \frac{1}{\gamma}\right)\frac{\rho^2}{n}\sum_{i=1}^{n}\left\|\boldsymbol{y}_i^{(r)} - \bar{\boldsymbol{y}}^{(r)}\right\|^2 - \rho^2\left\|\bar{\boldsymbol{x}}^{(r+1)} - \bar{\boldsymbol{y}}^{(r)}\right\|^2$$

$$\le \frac{(1+\gamma)\rho^2}{n}\sum_{i=1}^{n}\left\|\boldsymbol{x}_i^{(r+\frac{1}{2})} - \boldsymbol{y}_i^{(r)}\right\|^2 + \left(1 + \frac{1}{\gamma}\right)\frac{\rho^2 A_r}{nA_{r+1}}\sum_{i=1}^{n}\left\|\boldsymbol{x}_i^{(r)} - \bar{\boldsymbol{x}}^{(r)}\right\|^2 + \left(1 + \frac{1}{\gamma}\right)\frac{\rho^2 a_{r+1}}{nA_{r+1}}\sum_{i=1}^{n}\left\|\boldsymbol{v}_i^{(r)} - \bar{\boldsymbol{v}}^{(r)}\right\|^2$$

$$- \rho^2\left\|\bar{\boldsymbol{x}}^{(r+1)} - \bar{\boldsymbol{y}}^{(r)}\right\|^2,$$

where we use Jensen's inequality in the last inequality. Substituting $\gamma = \frac{\rho}{1-\rho}$, we obtain the statement. $\square$

### F.3. Convex Case

**Lemma 25.** *Suppose that Assumptions 1, 2, and 4 hold with $\mu = 0$, and the following inequality is satisfied:*

$$\sum_{i=1}^{n}\left\|\nabla F_{i,r}(\boldsymbol{x}_i^{(r+\frac{1}{2})})\right\|^2 \le \frac{\lambda^2}{352}\sum_{i=1}^{n}\left\|\boldsymbol{x}_i^{(r+\frac{1}{2})} - \boldsymbol{y}_i^{(r)}\right\|^2.$$

*Then, when $\lambda = 208\delta$, $M = 1$, and*

$$\rho \le \min\left\{\frac{9}{1600}, \frac{\delta}{6L}, \frac{5}{48(R+1)^3}\right\},$$

*it holds that*

$$A_r f(\bar{\boldsymbol{x}}^{(r)}) + a_{r+1} f(\boldsymbol{x}^\star) + \frac{1}{2n} \sum_{i=1}^n \left\| \boldsymbol{v}_i^{(r)} - \boldsymbol{x}^\star \right\|^2$$

$$+ \frac{2A_{r+1}}{\delta} \mathcal{E}^{(r)} + 990\delta A_r \Xi_{\boldsymbol{x}}^{(r)} + \delta a_{r+1} \Xi_{\boldsymbol{v}}^{(r)} + \frac{R\rho}{24(R+1)^2} \left\| \bar{\boldsymbol{v}}^{(r)} - \bar{\boldsymbol{x}}^{(r)} \right\|^2$$

$$\geq A_{r+1} f(\bar{\boldsymbol{x}}^{(r+1)}) + \frac{1}{2n} \sum_{i=1}^n \left\| \boldsymbol{v}_i^{(r+1)} - \boldsymbol{x}^\star \right\|^2$$

$$+ \frac{2A_{r+2}}{\delta} \mathcal{E}^{(r+1)} + 990\delta A_{r+1} \Xi_{\boldsymbol{x}}^{(r+1)} + \delta a_{r+2} \Xi_{\boldsymbol{v}}^{(r+1)} + \frac{R\rho}{24(R+1)^2} \left\| \bar{\boldsymbol{v}}^{(r+1)} - \bar{\boldsymbol{x}}^{(r+1)} \right\|^2,$$

*where* $\Xi_{\boldsymbol{x}}^{(r)} := \frac{1}{n} \sum_{i=1}^n \left\| \boldsymbol{x}_i^{(r)} - \bar{\boldsymbol{x}}^{(r)} \right\|^2$, $\Xi_{\boldsymbol{v}}^{(r)} := \frac{1}{n} \sum_{i=1}^n \left\| \boldsymbol{v}_i^{(r)} - \bar{\boldsymbol{v}}^{(r)} \right\|^2$, *and* $\mathcal{E}^{(r)} := \frac{1}{n} \sum_{i=1}^n \left\| \boldsymbol{h}_i^{(r)} - \nabla h_i(\bar{\boldsymbol{y}}^{(r)}) \right\|^2$.

*Proof.* Using $\rho \leq \frac{1}{4}$, we can simplify Lemmas 22 and 24 as follows:

$$\rho \mathcal{E}^{(r)} \geq \mathcal{E}^{(r+1)} - 8\rho^2 \delta^2 \frac{a_{r+2}}{A_{r+2}} \left\| \bar{\boldsymbol{v}}^{(r+1)} - \bar{\boldsymbol{x}}^{(r+1)} \right\|^2$$

$$- 8\rho^2 \delta^2 \frac{1}{n} \sum_{i=1}^n \left\| \boldsymbol{x}_i^{(r+\frac{1}{2})} - \boldsymbol{y}_i^{(r)} \right\|^2 - 32L^2 \frac{A_{r+1}}{A_{r+2}} \Xi_{\boldsymbol{x}}^{(r+1)} - 32L^2 \frac{a_{r+2}}{A_{r+2}} \Xi_{\boldsymbol{v}}^{(r+1)}, \tag{38}$$

$$\rho A_r \Xi_{\boldsymbol{x}}^{(r)} \geq A_{r+1} \Xi_{\boldsymbol{x}}^{(r+1)} - \frac{4}{3} \rho^2 A_{r+1} \frac{1}{n} \sum_{i=1}^n \left\| \boldsymbol{x}_i^{(r+\frac{1}{2})} - \boldsymbol{y}_i^{(r)} \right\|^2 - \frac{4}{3} \rho^2 a_{r+1} \Xi_{\boldsymbol{v}}^{(r)}. \tag{39}$$

Using Lemma 19, we can simplify Lemma 23 as follows:

$$\frac{R}{R+1} \left\| \bar{\boldsymbol{v}}^{(r)} - \bar{\boldsymbol{x}}^{(r)} \right\|^2$$

$$\geq \left\| \bar{\boldsymbol{v}}^{(r+1)} - \bar{\boldsymbol{x}}^{(r+1)} \right\|^2 - \frac{4A_{r+1}}{a_{r+1}} \frac{1}{n} \sum_{i=1}^n \left\| \boldsymbol{x}_i^{(r+\frac{1}{2})} - \boldsymbol{y}_i^{(r)} \right\|^2 - \frac{2A_{r+1}}{a_{r+1}} \frac{1}{n} \sum_{i=1}^n \left\| \boldsymbol{v}_i^{(r+\frac{1}{2})} - \boldsymbol{v}_i^{(r)} \right\|^2. \tag{40}$$

From Eq. (33) $+ \frac{2A_{r+2}}{\delta} \times$ Eq. (38) $+ \frac{\rho}{24(R+1)} \times$ Eq. (40) $+ \frac{560\delta}{\rho} \times$ Eq. (39) and $\lambda = 208\delta$, we obtain

$$A_r f(\bar{\boldsymbol{x}}^{(r)}) + a_{r+1} f(\boldsymbol{x}^\star) + \frac{1}{2n} \sum_{i=1}^n \left\| \boldsymbol{v}_i^{(r)} - \boldsymbol{x}^\star \right\|^2 + \left( 1 + \frac{2A_{r+2}}{A_{r+1}} \rho \right) \frac{A_{r+1}}{\delta} \mathcal{E}^{(r)}$$

$$+ 990\delta A_r \Xi_{\boldsymbol{x}}^{(r)} + \delta a_{r+1} \Xi_{\boldsymbol{v}}^{(r)} + \frac{R\rho}{24(R+1)^2} \left\| \bar{\boldsymbol{v}}^{(r)} - \bar{\boldsymbol{x}}^{(r)} \right\|^2$$

$$\geq A_{r+1} f(\bar{\boldsymbol{x}}^{(r+1)}) + \frac{1}{2n} \sum_{i=1}^n \left\| \boldsymbol{v}_i^{(r+1)} - \boldsymbol{x}^\star \right\|^2 + \frac{2A_{r+2}}{\delta} \mathcal{E}^{(r+1)}$$

$$+ \left( \frac{560\delta}{\rho} - \frac{32L^2}{\delta} \right) A_{r+1} \Xi_{\boldsymbol{x}}^{(r+1)} + \left[ \frac{1}{2a_{r+2}} \left( \frac{1}{\rho^2} - 1 \right) - \frac{32L^2}{\delta} - \frac{2240a_{r+1}}{3a_{r+2}} \delta\rho \right] a_{r+2} \Xi_{\boldsymbol{v}}^{(r+1)}$$

$$+ \left( \frac{\rho}{24(R+1)} - 16a_{r+2}\rho^2 \delta \right) \left\| \bar{\boldsymbol{v}}^{(r+1)} - \bar{\boldsymbol{x}}^{(r+1)} \right\|^2$$

$$+ \left[ \frac{9}{2}\delta - \frac{16A_{r+2}}{A_{r+1}} \rho^2 \delta - \frac{\rho}{6a_{r+1}(R+1)} - \frac{2240}{3}\delta\rho \right] \frac{A_{r+1}}{n} \sum_{i=1}^n \left\| \boldsymbol{x}_i^{(r+\frac{1}{2})} - \boldsymbol{y}_i^{(r)} \right\|^2$$

$$+ \left( \frac{1}{12} - \frac{A_{r+1}}{12a_{r+1}(R+1)} \rho \right) \frac{1}{n} \sum_{i=1}^n \left\| \boldsymbol{v}_i^{(r+\frac{1}{2})} - \boldsymbol{v}_i^{(r)} \right\|^2,$$

where we use that fact that $B_r = 1$ for all $r$ when $\mu = 0$. From Lemmas 18, 19, 19, and 20, we have

$$\frac{A_{r+1}}{a_{r+1}} \overset{(30)}{\leq} 1 + R, \quad \frac{A_{r+2}}{A_{r+1}} \overset{(30)}{\leq} 4, \quad \frac{a_{r+1}}{a_{r+2}} \overset{(32)}{\leq} 1, \quad \frac{1}{a_{r+2}} \overset{(32)}{\leq} \frac{1}{a_1} = 208\delta, \quad \frac{1}{a_{r+1}} \overset{(32)}{\leq} \frac{1}{a_1} = 208\delta, \quad a_{r+2} \overset{(30),(28)}{\leq} \frac{R+1}{40\delta}.$$

Using the above inequalities, we obtain

$$A_r f(\bar{\boldsymbol{x}}^{(r)}) + a_{r+1} f(\boldsymbol{x}^\star) + \frac{1}{2n} \sum_{i=1}^n \left\| \boldsymbol{v}_i^{(r)} - \boldsymbol{x}^\star \right\|^2 + (1 + 8\rho) \frac{A_{r+1}}{\delta} \mathcal{E}^{(r)}$$

$$+ 990\delta A_r \Xi_{\boldsymbol{x}}^{(r)} + \delta a_{r+1} \Xi_{\boldsymbol{v}}^{(r)} + \frac{R\rho}{24(R+1)^2} \left\| \bar{\boldsymbol{v}}^{(r)} - \bar{\boldsymbol{x}}^{(r)} \right\|^2$$

$$\geq A_{r+1} f(\bar{\boldsymbol{x}}^{(r+1)}) + \frac{1}{2n} \sum_{i=1}^n \left\| \boldsymbol{v}_i^{(r+1)} - \boldsymbol{x}^\star \right\|^2 + \frac{2A_{r+2}}{\delta} \mathcal{E}^{(r+1)}$$

$$+ \left( \frac{560\delta}{\rho} - \frac{32L^2}{\delta} \right) A_{r+1} \Xi_{\boldsymbol{x}}^{(r+1)} + \left[ 104\delta \left( \frac{1}{\rho^2} - 1 \right) - \frac{32L^2}{\delta} - 750\delta\rho \right] a_{r+2} \Xi_{\boldsymbol{v}}^{(r+1)}$$

$$+ \left( \frac{\rho}{24(R+1)} - \frac{2(R+1)}{5} \rho^2 \right) \left\| \bar{\boldsymbol{v}}^{(r+1)} - \bar{\boldsymbol{x}}^{(r+1)} \right\|^2$$

$$+ \left[ \frac{9}{2}\delta - 16\rho\delta - \frac{35}{(R+1)}\delta\rho - \frac{2240}{3}\delta\rho \right] \frac{A_{r+1}}{n} \sum_{i=1}^n \left\| \boldsymbol{x}_i^{(r+\frac{1}{2})} - \boldsymbol{y}_i^{(r)} \right\|^2$$

$$+ \left( \frac{1}{12} - \frac{1}{12}\rho \right) \frac{1}{n} \sum_{i=1}^n \left\| \boldsymbol{v}_i^{(r+\frac{1}{2})} - \boldsymbol{v}_i^{(r)} \right\|^2,$$

Then, using the assumption on $\rho$, we have

$$1 + 8\rho \leq 2,$$

$$\frac{560\delta}{\rho} - \frac{32L^2}{\delta} \geq 900\delta,$$

$$104\delta \left( \frac{1}{\rho^2} - 1 \right) - \frac{32L^2}{\delta} - 750\delta\rho \geq \delta,$$

$$\frac{\rho}{24(R+1)} - \frac{2(R+1)}{5}\rho^2 \geq \frac{R\rho}{24(R+1)^2},$$

$$\frac{9}{2}\delta - 16\delta\rho - \frac{35}{R+1}\delta\rho - \frac{2240}{3}\delta\rho \geq 0.$$

Thus, we can simplify the above inequality as follows:

$$A_r f(\bar{\boldsymbol{x}}^{(r)}) + a_{r+1} f(\boldsymbol{x}^\star) + \frac{1}{2n} \sum_{i=1}^n \left\| \boldsymbol{v}_i^{(r)} - \boldsymbol{x}^\star \right\|^2$$

$$+ \frac{2A_{r+1}}{\delta} \mathcal{E}^{(r)} + 990\delta A_r \Xi_{\boldsymbol{x}}^{(r)} + \delta a_{r+1} \Xi_{\boldsymbol{v}}^{(r)} + \frac{R\rho}{24(R+1)^2} \left\| \bar{\boldsymbol{v}}^{(r)} - \bar{\boldsymbol{x}}^{(r)} \right\|^2$$

$$\geq A_{r+1} f(\bar{\boldsymbol{x}}^{(r+1)}) + \frac{1}{2n} \sum_{i=1}^n \left\| \boldsymbol{v}_i^{(r+1)} - \boldsymbol{x}^\star \right\|^2$$

$$+ \frac{2A_{r+2}}{\delta} \mathcal{E}^{(r+1)} + 990\delta A_{r+1} \Xi_{\boldsymbol{x}}^{(r+1)} + \delta a_{r+2} \Xi_{\boldsymbol{v}}^{(r+1)} + \frac{R\rho}{24(R+1)^2} \left\| \bar{\boldsymbol{v}}^{(r+1)} - \bar{\boldsymbol{x}}^{(r+1)} \right\|^2.$$

Thus, we obtain the desired result. □

**Lemma 26.** *Suppose that Assumptions 1, 2, and 4 hold with $\mu = 0$, and the following inequality is satisfied:*

$$\sum_{i=1}^n \left\| \nabla F_{i,r}(\boldsymbol{x}_i^{(r+\frac{1}{2})}) \right\|^2 \leq \frac{\lambda^2}{352} \sum_{i=1}^n \left\| \boldsymbol{x}_i^{(r+\frac{1}{2})} - \boldsymbol{y}_i^{(r)} \right\|^2.$$

*Then, when $\lambda = 208\delta$, $M = 1$, and*

$$\rho \leq \min \left\{ \frac{9}{1600}, \frac{\delta}{6L}, \frac{5}{48(R+1)^3} \right\},$$

*it holds that $f(\bar{\boldsymbol{x}}^{(R)}) - f(\boldsymbol{x}^\star) \leq \epsilon$ after*

$$R \geq \sqrt{\frac{416\delta \left\| \bar{\boldsymbol{x}}^{(0)} - \boldsymbol{x}^\star \right\|^2}{\epsilon}}$$

*rounds.*

*Proof.* From Lemma 25, we have

$$A_{r+1} \left( f(\bar{\boldsymbol{x}}^{(r+1)}) - f(\boldsymbol{x}^\star) \right)$$

$$\leq A_r \left( f(\bar{\boldsymbol{x}}^{(r)}) - f(\boldsymbol{x}^\star) \right) + \frac{1}{2n} \sum_{i=1}^n \left\| \boldsymbol{v}_i^{(r)} - \boldsymbol{x}^\star \right\|^2 - \frac{1}{2n} \sum_{i=1}^n \left\| \boldsymbol{v}_i^{(r+1)} - \boldsymbol{x}^\star \right\|^2$$

$$+ \frac{2A_{r+1}}{\delta} \mathcal{E}^{(r)} - \frac{2A_{r+2}}{\delta} \mathcal{E}^{(r+1)} + 990\delta \left( A_r \Xi_{\boldsymbol{x}}^{(r)} - A_{r+1} \Xi_{\boldsymbol{x}}^{(r+1)} \right) + \delta \left( a_{r+1} \Xi_{\boldsymbol{v}}^{(r)} - a_{r+2} \Xi_{\boldsymbol{v}}^{(r+1)} \right)$$

$$+ \frac{R(1-\rho)}{24(R+1)^2} \left( \left\| \bar{\boldsymbol{v}}^{(r)} - \bar{\boldsymbol{x}}^{(r)} \right\|^2 - \left\| \bar{\boldsymbol{v}}^{(r+1)} - \bar{\boldsymbol{x}}^{(r+1)} \right\|^2 \right).$$

Telescoping the sum from $r = 0$ to $r = R - 1$ and using $\Xi_{\boldsymbol{x}}^{(0)} = \Xi_{\boldsymbol{v}}^{(0)} = \mathcal{E}^{(0)} = 0$ and $\boldsymbol{v}_i^{(0)} = \boldsymbol{x}_i^{(0)}$, we have

$$f(\bar{\boldsymbol{x}}^{(R)}) - f(\boldsymbol{x}^\star) \leq \frac{1}{2A_R} \left\| \bar{\boldsymbol{v}}^{(0)} - \boldsymbol{x}^\star \right\|^2.$$

Using Lemma 17, we obtain the desired result. $\qquad\square$

**Lemma 27.** *Suppose that Assumptions 1, 2, and 4 hold with $\mu = 0$, and the following inequality is satisfied:*

$$\sum_{i=1}^n \left\| \nabla F_{i,r}(\boldsymbol{x}_i^{(r+\frac{1}{2})}) \right\|^2 \leq \frac{\lambda^2}{352} \sum_{i=1}^n \left\| \boldsymbol{x}_i^{(r+\frac{1}{2})} - \boldsymbol{y}_i^{(r)} \right\|^2.$$

*Then, when $\lambda = 208\delta$, $\gamma = \frac{1-\sqrt{1-\rho^2}}{1+\sqrt{1+\rho^2}}$, and $M \geq \frac{3}{2\sqrt{1-\rho}} \log(\max\{ \frac{12L}{\delta}, \frac{20384\delta \|\bar{\boldsymbol{x}}^{(0)} - \boldsymbol{x}^\star\|^2}{\epsilon} \})$, it holds that $f(\bar{\boldsymbol{x}}^{(R)}) - f(\boldsymbol{x}^\star) \leq \epsilon$ after*

$$R = \mathcal{O} \left( \sqrt{\frac{\delta \left\| \bar{\boldsymbol{x}}^{(0)} - \boldsymbol{x}^\star \right\|^2}{\epsilon}} \right)$$

*rounds.*

*Proof.* Define $\tilde{\boldsymbol{W}}^{(m)}$ as follows:

$$\tilde{\boldsymbol{W}}^{(-1)} = \boldsymbol{I}$$
$$\tilde{\boldsymbol{W}}^{(0)} = \boldsymbol{I}$$
$$\tilde{\boldsymbol{W}}^{(m+1)} = (1+\gamma)\tilde{\boldsymbol{W}}^{(m)}\boldsymbol{W} - \gamma\tilde{\boldsymbol{W}}^{(m-1)}.$$

Then, the output of Alg. 5 is equivalent to

$$\sum_{j=1}^n \tilde{\boldsymbol{W}}_{ij}^{(M)} \boldsymbol{a}_j.$$

From Proposition 1 in Yuan et al. (2022), it holds that for any $\boldsymbol{a}_1, \boldsymbol{a}_2, \ldots, \boldsymbol{a}_n \in \mathbb{R}^d$

$$\sum_{i=1}^{n} \left\| \sum_{j=1}^{n} \tilde{W}_{ij}^{(M)} \boldsymbol{a}_j - \bar{\boldsymbol{a}} \right\| \leq \sqrt{2} \left( 1 - \sqrt{1-\rho} \right)^M \sum_{i=1}^{n} \| \boldsymbol{a}_i - \bar{\boldsymbol{a}} \|,$$

where $\bar{\boldsymbol{a}} := \frac{1}{n} \sum_{i=1}^{n} \boldsymbol{a}_i$. Thus, When

$$M \geq \frac{1}{\sqrt{1-\rho}} \log \left( \max \left\{ \frac{6\sqrt{2}L}{\delta}, \frac{1600\sqrt{2}R^3}{9} \right\} \right),$$

we can satisfy the condition on $\rho$ in Lemma 26. We used $R \geq 1$ to simplify the condition on $\rho$. From Lemma 26, we obtain the desired result. $\qquad \square$

### F.4. Strongly Convex Case

**Lemma 28.** *Suppose that Assumptions 1 and 4 hold and the following inequality is satisfied:*

$$\sum_{i=1}^{n} \left\| \nabla F_{i,r}(\boldsymbol{x}_i^{(r+\frac{1}{2})}) \right\|^2 \leq \frac{\lambda^2}{352} \sum_{i=1}^{n} \left\| \boldsymbol{x}_i^{(r+\frac{1}{2})} - \boldsymbol{y}_i^{(r)} \right\|^2.$$

*Then, when $\lambda = 96\delta$, $M = 1$ and*

$$\rho \leq \min \left[ \sqrt{\frac{\mu\delta}{1728L^2}}, \sqrt{\frac{48\delta^2\mu}{260L^2(192\delta+\mu)}}, \right.$$
$$\left. \left( 4 + 2 \left( 1 + \frac{\mu}{96\delta} \right) \left( 1 + \sqrt{\frac{384\delta}{\mu}} \right) \right)^{-1}, \frac{\mu}{4\delta} \left( \left( 1 + \frac{96\delta}{\mu} \right) \left( 1 + \frac{192\delta}{\mu} \right) \left( 1 + \sqrt{1 + \frac{384\delta}{\mu}} \right) \right)^{-1} \right],$$

*it holds that*

$$A_r f(\bar{\boldsymbol{x}}^{(r)}) + a_{r+1} f(\boldsymbol{x}^\star) + \frac{B_r}{2n} \sum_{i=1}^{n} \left\| \boldsymbol{v}_i^{(r)} - \boldsymbol{x}^\star \right\|^2 + \frac{2A_{r+1}}{\delta} \mathcal{E}^{(r)} + 200\delta A_r \Xi_{\boldsymbol{x}}^{(r)} + \delta a_{r+1} \Xi_{\boldsymbol{v}}^{(r)} \tag{41}$$
$$\geq A_{r+1} f(\bar{\boldsymbol{x}}^{(r+1)}) + \frac{B_{r+1}}{2n} \sum_{i=1}^{n} \left\| \boldsymbol{v}_i^{(r+1)} - \boldsymbol{x}^\star \right\|^2 + \frac{2A_{r+2}}{\delta} \mathcal{E}^{(r+1)} + 200\delta A_{r+1} \Xi_{\boldsymbol{x}}^{(r+1)} + \delta a_{r+2} \Xi_{\boldsymbol{v}}^{(r+1)},$$

*where $\Xi_{\boldsymbol{x}}^{(r)} := \frac{1}{n} \sum_{i=1}^{n} \left\| \boldsymbol{x}_i^{(r)} - \bar{\boldsymbol{x}}^{(r)} \right\|^2$, $\Xi_{\boldsymbol{v}}^{(r)} := \frac{1}{n} \sum_{i=1}^{n} \left\| \boldsymbol{v}_i^{(r)} - \bar{\boldsymbol{v}}^{(r)} \right\|^2$, and $\mathcal{E}^{(r)} := \frac{1}{n} \sum_{i=1}^{n} \left\| \boldsymbol{h}_i^{(r)} - \nabla h_i(\bar{\boldsymbol{y}}^{(r)}) \right\|^2$.*

*Proof.* From Lemma 8, $\rho \leq \frac{1}{4}$ and $\bar{\boldsymbol{y}}^{(r+1)} = \frac{A_{r+1} \bar{\boldsymbol{x}}^{(r+1)} + a_{r+2} \bar{\boldsymbol{v}}^{(r+1)}}{A_{r+2}}$, we have

$$\rho \mathcal{E}^{(r)} \geq \mathcal{E}^{(r+1)} - 2\rho\delta^2 \frac{a_{r+2}}{A_{r+2}} \left\| \bar{\boldsymbol{v}}^{(r+1)} - \bar{\boldsymbol{x}}^{(r+1)} \right\|^2$$
$$- 2\rho\delta^2 \frac{1}{n} \sum_{i=1}^{n} \left\| \boldsymbol{x}_i^{(r+\frac{1}{2})} - \boldsymbol{y}_i^{(r)} \right\|^2 - 32L^2 \frac{A_{r+1}}{A_{r+2}} \Xi_{\boldsymbol{x}}^{(r+1)} - 32L^2 \frac{a_{r+2}}{A_{r+2}} \Xi_{\boldsymbol{v}}^{(r+1)}, \tag{42}$$

From Eq. (33) $+\frac{2A_{r+2}}{\delta} \times$ Eq. (42) and $\lambda = 96\delta$, we have

$$A_r f(\bar{\boldsymbol{x}}^{(r)}) + a_{r+1} f(\boldsymbol{x}^\star) + \frac{B_r}{2n} \sum_{i=1}^n \left\| \boldsymbol{v}_i^{(r)} - \boldsymbol{x}^\star \right\|^2 + \frac{A_{r+1}}{\delta} \left( 1 + \frac{2A_{r+2}\rho}{A_{r+1}} \right) \mathcal{E}^{(r)} + 200\delta A_r \Xi_{\boldsymbol{x}}^{(r)} + \delta a_{r+1} \Xi_{\boldsymbol{v}}^{(r)}$$

$$\geq A_{r+1} f(\bar{\boldsymbol{x}}^{(r+1)}) + \frac{B_{r+1}}{2n} \sum_{i=1}^n \left\| \boldsymbol{v}_i^{(r+1)} - \boldsymbol{x}^\star \right\|^2 + \frac{2A_{r+2}}{\delta} \mathcal{E}^{(r+1)}$$

$$+ \delta A_{r+1} \left[ \frac{\mu}{2\delta\rho^2} - \frac{64L^2}{\delta^2} \right] \Xi_{\boldsymbol{x}}^{(r+1)} + \delta a_{r+2} \left[ \frac{B_{r+1}}{2\delta a_{r+2}} \left( \frac{1}{\rho^2} - 1 \right) - \frac{64L^2}{\delta^2} \right] \Xi_{\boldsymbol{v}}^{(r+1)} + \frac{B_r}{12n} \sum_{i=1}^n \left\| \boldsymbol{v}_i^{(r+\frac{1}{2})} - \boldsymbol{v}_i^{(r)} \right\|^2$$

$$+ \frac{\delta}{n} \left[ A_{r+1} - 4\rho A_{r+2} \right] \sum_{i=1}^n \left\| \boldsymbol{x}_i^{(r+\frac{1}{2})} - \boldsymbol{y}_i^{(r)} \right\|^2 + \frac{1}{n} \left[ \frac{\mu a_{r+1}}{2} - 4\rho\delta a_{r+2} \right] \sum_{i=1}^n \left\| \boldsymbol{x}_i^{(r+\frac{1}{2})} - \boldsymbol{v}_i^{(r+\frac{1}{2})} \right\|^2 .$$

Using Lemmas 19, and 20, $\delta \leq L$, and the condition of $\rho$, we have

$$1 + \frac{2A_{r+2}}{A_{r+1}}\rho = 1 + 2\left(1 + \frac{a_{r+2}}{A_{r+1}}\right)\rho \leq 2,$$

$$\frac{\mu}{2\delta\rho^2} - \frac{64L^2}{\delta^2} \geq 200,$$

$$\frac{B_{r+1}}{2\delta a_{r+2}}\left(\frac{1}{\rho^2} - 1\right) - \frac{64L^2}{\delta^2} = \frac{48a_{r+2}}{A_{r+2}}\left(\frac{1}{\rho^2} - 1\right) - \frac{64L^2}{\delta^2} \geq 1,$$

$$A_{r+1} - 4\rho A_{r+2} \geq 0,$$

$$\frac{\mu a_{r+1}}{2} - 4\rho\delta a_{r+2} \geq 0.$$

Using the above inequalities, we obtain the desired result. $\qquad\square$

**Lemma 29.** *Suppose that Assumptions 1, 2, and 4 hold and the following inequality is satisfied:*

$$\sum_{i=1}^n \left\| \nabla F_{i,r}(\boldsymbol{x}_i^{(r+\frac{1}{2})}) \right\|^2 \leq \frac{\lambda^2}{352} \sum_{i=1}^n \left\| \boldsymbol{x}_i^{(r+\frac{1}{2})} - \boldsymbol{y}_i^{(r)} \right\|^2 .$$

*Then, when $\lambda = 96\delta$, $M = 1$, and*

$$\rho \leq \min\left[ \sqrt{\frac{\mu\delta}{1728L^2}}, \sqrt{\frac{48\delta^2\mu}{260L^2(192\delta + \mu)}}, \right.$$

$$\left. \left(4 + 2\left(1 + \frac{\mu}{96\delta}\right)\left(1 + \sqrt{\frac{384\delta}{\mu}}\right)\right)^{-1}, \frac{\mu}{4\delta}\left(\left(1 + \frac{96\delta}{\mu}\right)\left(1 + \frac{192\delta}{\mu}\right)\left(1 + \sqrt{1 + \frac{384\delta}{\mu}}\right)\right)^{-1} \right],$$

*it holds that $f(\bar{\boldsymbol{x}}^{(R)}) - f(\boldsymbol{x}^\star) \leq \epsilon$ after*

$$R = \mathcal{O}\left( \sqrt{\frac{\delta + \mu}{\mu}} \log\left(1 + \sqrt{\frac{\min\{\mu,\delta\}\|\bar{\boldsymbol{x}}^{(0)} - \boldsymbol{x}^\star\|^2}{\epsilon}}\right) \right)$$

*rounds.*

*Proof.* We have

$$A_r\left(f(\bar{\boldsymbol{x}}^{(r)}) - f(\boldsymbol{x}^\star)\right) + \frac{B_r}{2n} \sum_{i=1}^n \left\| \boldsymbol{v}_i^{(r)} - \boldsymbol{x}^\star \right\|^2 + \frac{2A_{r+1}}{\delta} \mathcal{E}^{(r)} + 200\delta A_r \Xi_{\boldsymbol{x}}^{(r)} + \delta a_{r+1} \Xi_{\boldsymbol{v}}^{(r)}$$

$$\geq A_{r+1}\left(f(\bar{\boldsymbol{x}}^{(r+1)}) - f(\boldsymbol{x}^\star)\right) + \frac{B_{r+1}}{2n} \sum_{i=1}^n \left\| \boldsymbol{v}_i^{(r+1)} - \boldsymbol{x}^\star \right\|^2 + \frac{2A_{r+2}}{\delta} \mathcal{E}^{(r+1)} + 200\delta A_{r+1} \Xi_{\boldsymbol{x}}^{(r+1)} + \delta a_{r+2} \Xi_{\boldsymbol{v}}^{(r+1)}.$$

Telescoping the sum from $r = 0$ to $r = R - 1$ and using $\Xi_v^{(0)} = \Xi_x^{(0)} = \mathcal{E}^{(0)} = 0$, we have

$$A_R \left( f(\bar{\boldsymbol{x}}^{(R)}) - f(\boldsymbol{x}^\star) \right) \leq \frac{B_0}{2n} \sum_{i=1}^n \left\| \boldsymbol{v}_i^{(0)} - \boldsymbol{x}^\star \right\|^2 = \frac{1}{2} \left\| \bar{\boldsymbol{x}}^{(0)} - \boldsymbol{x}^\star \right\|^2.$$

From Lemma 17, we obtain

$$f(\bar{\boldsymbol{x}}^{(R)}) - f(\boldsymbol{x}^\star) \leq \begin{cases} \dfrac{2\mu}{\left[ \left( 1 + \sqrt{\frac{\mu}{384\delta}} \right)^R - 1 \right]^2} \left\| \bar{\boldsymbol{x}}^{(0)} - \boldsymbol{x}^\star \right\|^2 & \text{if } \mu \leq 384\delta, \\[2ex] \dfrac{96\delta}{4^{R-2}} \left\| \bar{\boldsymbol{x}}^{(0)} - \boldsymbol{x}^\star \right\|^2 & \text{if } \mu > 384\delta \end{cases},$$

for any $R \geq 1$. Then, we obtain the desired result. $\qquad\square$

**Lemma 30.** *Suppose that Assumptions 1, 2, and 4 hold and the following inequality is satisfied:*

$$\sum_{i=1}^n \left\| \nabla F_{i,r}(\boldsymbol{x}_i^{(r+\frac{1}{2})}) \right\|^2 \leq \frac{\lambda^2}{352} \sum_{i=1}^n \left\| \boldsymbol{x}_i^{(r+\frac{1}{2})} - \boldsymbol{y}_i^{(r)} \right\|^2.$$

*Then, when $\lambda = 96\delta$, $\gamma = \frac{1 - \sqrt{1 - \rho^2}}{1 + \sqrt{1 + \rho^2}}$, and $M \geq \frac{4}{\sqrt{1-\rho}} \log(\frac{8L^2(432\delta + \mu)}{\mu\delta^2})$, it holds that $f(\bar{\boldsymbol{x}}^{(R)}) - f(\boldsymbol{x}^\star) \leq \epsilon$ after*

$$R = \mathcal{O}\left( \sqrt{\frac{\delta + \mu}{\mu}} \log\left( 1 + \sqrt{\frac{\min\{\mu, \delta\} \|\bar{\boldsymbol{x}}^{(0)} - \boldsymbol{x}^\star\|^2}{\epsilon}} \right) \right)$$

*rounds.*

*Proof.* Define $\tilde{\boldsymbol{W}}^{(m)}$ as follows:

$$\tilde{\boldsymbol{W}}^{(-1)} = \boldsymbol{I}$$
$$\tilde{\boldsymbol{W}}^{(0)} = \boldsymbol{I}$$
$$\tilde{\boldsymbol{W}}^{(m+1)} = (1 + \gamma)\tilde{\boldsymbol{W}}^{(m)}\boldsymbol{W} - \gamma\tilde{\boldsymbol{W}}^{(m-1)}.$$

Then, the output of Alg. 5 is equivalent to

$$\sum_{j=1}^n \tilde{\boldsymbol{W}}_{ij}^{(M)} \boldsymbol{a}_j.$$

From Proposition 1 in Yuan et al. (2022), it holds that for any $\boldsymbol{a}_1, \boldsymbol{a}_2, \ldots, \boldsymbol{a}_n \in \mathbb{R}^d$

$$\sum_{i=1}^n \left\| \sum_{j=1}^n \tilde{\boldsymbol{W}}_{ij}^{(M)} \boldsymbol{a}_j - \bar{\boldsymbol{a}} \right\|^2 \leq 2 \left( 1 - \sqrt{1 - \rho} \right)^{2M} \sum_{i=1}^n \|\boldsymbol{a}_i - \bar{\boldsymbol{a}}\|^2,$$

where $\bar{\boldsymbol{a}} := \frac{1}{n} \sum_{i=1}^n \boldsymbol{a}_i$.

When $M \geq \frac{1}{\sqrt{1-\rho}} \log(\frac{\sqrt{2}}{c})$, we have $\sqrt{2}(1 - \sqrt{1 - \rho})^M \leq c$. Thus, when

$$M \geq \frac{4}{\sqrt{1-\rho}} \log\left( \frac{18L^2(192\delta + \mu)}{\mu\delta^2} \right),$$

we can satisfy the condition of $\rho$ in Lemma 29 where we use $L \geq \delta$ and $L \geq \mu$. Thus, from Lemma 29, we obtain the desired result.

$\qquad\square$

# G. Analysis of Computational Complexity

## G.1. Useful Lemmas

**Lemma 31.** *Let $F : \mathbb{R}^d \to \mathbb{R}$ be a $\mu$-strongly convex function and $\boldsymbol{x}^\star$ be a minimizer of $F$. If the following is satisfied for some $\alpha \leq \mu$ and $\boldsymbol{x}, \boldsymbol{y} \in \mathbb{R}^d$*

$$\|\nabla F(\boldsymbol{x})\| \leq \alpha \|\boldsymbol{y} - \boldsymbol{x}^\star\|,$$

*it holds that*

$$\|\nabla F(\boldsymbol{x})\| \leq \frac{\alpha \mu}{\mu - \alpha} \|\boldsymbol{x} - \boldsymbol{y}\|.$$

*Proof.* We have

$$\|\boldsymbol{x} - \boldsymbol{y}\| \geq \|\boldsymbol{y} - \boldsymbol{x}^\star\| - \|\boldsymbol{x} - \boldsymbol{x}^\star\|$$
$$\geq \|\boldsymbol{y} - \boldsymbol{x}^\star\| - \frac{1}{\mu} \|\nabla F(\boldsymbol{x})\|,$$

where we use the strongly-convexity of $F$ in the last inequality. Using $\|\nabla F(\boldsymbol{x})\| \leq \alpha \|\boldsymbol{y} - \boldsymbol{x}^\star\|$, we obtain the desired result. $\square$

**Lemma 32.** *Let $F : \mathbb{R}^d \to \mathbb{R}$ be a $\mu$-strongly convex and $L$-smooth function and $\boldsymbol{x}^\star$ be a minimizer of $F$. If we use Nesterov's Accelerated Gradient Descent with initial parameter $\boldsymbol{x}^{(0)} \in \mathbb{R}^d$, it holds that for any $\beta > 0$*

$$\left\|\nabla F(\boldsymbol{x}^{(T)})\right\| \leq \beta \left\|\boldsymbol{x}^{(0)} - \boldsymbol{x}^{(T)}\right\|$$

*after*

$$T = \mathcal{O}\left(\sqrt{\frac{L}{\mu}} \log\left(\frac{L(\mu + L)(\mu + \beta)^2}{\mu^2 \beta^2}\right)\right)$$

*iterations where $\boldsymbol{x}^{(T)}$ is the output of Nesterov's Accelerated Gradient Descent.*

*Proof.* From Theorem 3.18 in Bubeck (2015), it holds that

$$F(\boldsymbol{x}^{(T)}) - F(\boldsymbol{x}^\star) \leq \frac{\mu + L}{2} \left\|\boldsymbol{x}^{(0)} - \boldsymbol{x}^\star\right\|^2 \exp\left(-T\sqrt{\frac{\mu}{L}}\right).$$

Using the $L$-smoothness, we have

$$\left\|\nabla F(\boldsymbol{x}^{(T)})\right\|^2 \leq L(\mu + L) \left\|\boldsymbol{x}^{(0)} - \boldsymbol{x}^\star\right\|^2 \exp\left(-T\sqrt{\frac{\mu}{L}}\right).$$

Thus, when $T \geq \sqrt{\frac{L}{\mu}} \log(\frac{L(\mu+L)}{\alpha^2})$, it holds that

$$\left\|\nabla F(\boldsymbol{x}^{(T)})\right\| \leq \alpha \left\|\boldsymbol{x}^{(0)} - \boldsymbol{x}^\star\right\|.$$

Thus, using Lemma 31, we obtain the desired result. $\square$

**Lemma 33.** *Let $F : \mathbb{R}^d \to \mathbb{R}$ be a $\mu$-strongly convex and $L$-smooth function and $\boldsymbol{x}^\star$ be a minimizer of $F$. If we use the algorithm proposed in Remark 1 in Nesterov et al. (2018) with initial parameter $\boldsymbol{x}^{(0)} \in \mathbb{R}^d$, it holds that for any $\beta > 0$*

$$\left\|\nabla F(\boldsymbol{x}^{(T)})\right\| \leq \beta \left\|\boldsymbol{x}^{(0)} - \boldsymbol{x}^{(T)}\right\|$$

*after*

$$T = \mathcal{O}\left(\sqrt{\frac{L(\mu + \beta)}{\beta \mu}}\right)$$

*iterations where $\boldsymbol{x}^{(T)}$ is the output.*

*Proof.* From Remark 1 in Nesterov et al. (2018), it holds that

$$\|\nabla F(\boldsymbol{x}^{(T)})\| \leq \mathcal{O}\left(\frac{L\|\boldsymbol{x}^{(T)} - \boldsymbol{x}^\star\|}{T^2}\right).$$

Thus, when $T \geq \sqrt{\frac{L}{\alpha}}$, it holds that

$$\|\nabla F(\boldsymbol{x}^{(T)})\| \leq \mathcal{O}\left(\alpha\|\boldsymbol{x}^{(T)} - \boldsymbol{x}^\star\|\right).$$

Thus, using Lemma 31, we obtain the desired result. $\square$

### G.2. Proof of Theorem 2

**Lemma 34.** *Consider Alg. 1. Suppose that Assumptions 1, 2 and 4 hold, and $\boldsymbol{h}_i^{(0)} = \nabla f(\boldsymbol{x}_i^{(0)}) - \nabla f_i(\boldsymbol{x}_i^{(0)})$, $\boldsymbol{x}_i^{(0)} = \bar{\boldsymbol{x}}^{(0)}$, and we use the same $\lambda$ and $M$ as in Lemma 12. Then, if we use Nesterov's Accelerated Gradient Descent with initial parameter $\boldsymbol{x}_i^{(r)}$ to approximately solve the subproblem in line 4, each node requires at most*

$$\mathcal{O}\left(\sqrt{\frac{L}{\mu + \delta}} \log\left(\frac{L^2(r+2)^2}{\delta(\delta + \mu)}\right)\right)$$

*iterations to satisfy Eq. (8).*

*Proof.* $F_{i,r}$ is $\Omega(\mu + \delta)$-strongly convex and $\mathcal{O}(L + \delta)$-smooth. Thus, using Lemma 32, $\mu \leq L$, and $\delta \leq L$, we obtain the desired result. $\square$

### G.3. Proof of Theorem 4

**Lemma 35.** *Consider Alg. 3. Suppose that Assumptions 1, 2, and 4 hold, and $\boldsymbol{h}_i^{(0)} = \nabla f(\boldsymbol{v}_i^{(0)}) - \nabla f_i(\boldsymbol{v}_i^{(0)})$, $\boldsymbol{v}_i^{(0)} = \bar{\boldsymbol{v}}^{(0)}$, and we use the same $\lambda$ and $M$ as in Lemma 16. Then, if we use the algorithm proposed in Remark 1 in Nesterov et al. (2018) with initial parameter $\boldsymbol{v}_i^{(r)}$ to approximately solve the subproblem in line 4, each node requires at most*

$$\mathcal{O}\left(\sqrt{\frac{L}{\delta}}\right)$$

*iterations to satisfy Eq. (9).*

*Proof.* $F_{i,r}$ is $\Omega(\mu + \delta)$-strongly convex and $\mathcal{O}(L + \delta)$-smooth. Thus, using Lemma 33, $\mu \leq L$, and $\delta \leq L$, we obtain the desired result. $\square$

### G.4. Proof of Remark 2

**Lemma 36.** *Consider Alg. 3. Suppose that Assumptions 1, 2, and 4 hold, and $\boldsymbol{h}_i^{(0)} = \nabla f(\boldsymbol{v}_i^{(0)}) - \nabla f_i(\boldsymbol{v}_i^{(0)})$, $\boldsymbol{v}_i^{(0)} = \bar{\boldsymbol{v}}^{(0)}$, and we use the same $\lambda$ and $M$ as in Lemma 16. Then, if we use Nesterov's Accelerated Gradient Descent with initial parameter $\boldsymbol{v}_i^{(r)}$ to approximately solve the subproblem in line 4, each node requires at most*

$$\mathcal{O}\left(\sqrt{\frac{L}{\mu + \delta}} \log\left(\frac{L}{\delta}\right)\right)$$

*iterations to satisfy Eq. (9).*

*Proof.* $F_{i,r}$ is $\Omega(\mu + \delta)$-strongly convex and $\mathcal{O}(L + \delta)$-smooth. Thus, using Lemma 32, $\mu \leq L$, and $\delta \leq L$, we obtain the desired result. $\square$

### G.5. Proof of Theorem 6

**Lemma 37.** *Consider Alg. 4. Suppose that Assumptions 1, 2, and 4 hold, and $\boldsymbol{h}_i^{(0)} = \nabla f(\boldsymbol{v}_i^{(0)}) - \nabla f_i(\boldsymbol{v}_i^{(0)})$, $\boldsymbol{v}_i^{(0)} = \bar{\boldsymbol{v}}^{(0)}$, and we use the same $\gamma$, $M$, and $\lambda$ as Theorem 5. Then, if we use the algorithm proposed in Remark 1 in Nesterov et al. (2018) with initial parameter $\boldsymbol{y}_i^{(r)}$ to approximately solve the subproblem in line 12, each node requires at most*

$$\mathcal{O}\left(\sqrt{\frac{L}{\delta}}\right)$$

*iterations to satisfy Eq. (10).*

*Proof.* $F_{i,r}$ is $\Omega(\mu + \delta)$-strongly convex and $\mathcal{O}(L + \delta)$-smooth. Thus, using Lemma 33, $\mu \leq L$, and $\delta \leq L$, we obtain the desired result. $\qquad\square$

### G.6. Proof of Remark 3

**Lemma 38.** *Consider Alg. 4. Suppose that Assumptions 1, 2, and 4 hold, and $\boldsymbol{h}_i^{(0)} = \nabla f(\boldsymbol{v}_i^{(0)}) - \nabla f_i(\boldsymbol{v}_i^{(0)})$, $\boldsymbol{v}_i^{(0)} = \bar{\boldsymbol{v}}^{(0)}$, and we use the same $\gamma$, $M$, and $\lambda$ as Theorem 5. Then, if we use Nesterov's Accelerated Gradient Descent with initial parameter $\boldsymbol{y}_i^{(r)}$ to approximately solve the subproblem in line 12, each node requires at most*

$$\mathcal{O}\left(\sqrt{\frac{L}{\mu + \delta}} \log\left(\frac{L}{\delta}\right)\right)$$

*iterations to satisfy Eq. (10).*

*Proof.* $F_{i,r}$ is $\Omega(\mu + \delta)$-strongly convex and $\mathcal{O}(L + \delta)$-smooth. Thus, using Lemma 32, $\mu \leq L$, and $\delta \leq L$, we obtain the desired result. $\qquad\square$

# H. Experimental Setup

## H.1. Setup for Adaptive Number of Local Steps

In Fig. 1(a), we ran gradient descent until the following condition was satisfied.

- **Inexact-PDO:** $\|\nabla F_{i,r}(\boldsymbol{x}_i^{(r+\frac{1}{2})})\|^2 \leq \frac{\lambda(\lambda+\mu)}{(r+1)(r+2)}\|\boldsymbol{x}_i^{(r+\frac{1}{2})} - \boldsymbol{x}_i^{(r)}\|^2$.

- **SPDO:** $\|\nabla F_{i,r}(\boldsymbol{x}_i^{(r+1)})\| \leq \lambda\|\boldsymbol{x}_i^{(r+1)} - \boldsymbol{v}_i^{(r)}\|$.

- **Inexact Accelerated SONATA:** $\|\nabla F_{i,r}(\boldsymbol{x}_i^{(r+1)})\| \leq (1 - \sqrt{\frac{\mu}{\beta}})^r (\frac{16}{17})^t \|\boldsymbol{x}_i^{(r+1)} - \boldsymbol{x}_i^{(r)}\|$ where $t$ is the number of inner loop.

- **Accelerated-SPDO:** $\|\nabla F_{i,r}(\boldsymbol{x}_i^{(r+\frac{1}{2})})\| \leq \lambda\|\boldsymbol{x}_i^{(r+\frac{1}{2})} - \boldsymbol{y}_i^{(r)}\|$.

Tian et al. (2022) showed that Accelerated SONATA can achieve low communication complexity shown in Table 1 if $F_{i,r}(\boldsymbol{x}_i^{(r+1)}) - F_{i,r}(\boldsymbol{x}_{i,r}^\star)$ is sufficiently small, where $\boldsymbol{x}_{i,r}^\star$ is the minimizer of $F_{i,r}$. However, unlike the condition for our proposed methods, e.g., Eq. (9), this condition cannot be computed in practice. Therefore, we slightly modified this condition in our experiments to make it actually computable.

## H.2. Hyperparameter Setting

In Fig. 1, we tuned the hyperparameters by grid search. Specifically, we ran grid search over the hyperparameters listed in Table 3 and chose the hyperparameter that minimizes the norm of the last gradient.

*Table 3.* Experimental setups. Note that for Inexact Accelerated SONATA, $\delta$ is the coefficient of the additional L2 regularization in the subproblem, and it is different from Definition 1.

| | | |
|---|---|---|
| **Gradient Tracking** | $\eta$ | $\{0.05, 0.01, 0.005, 0.001\}$ |
| | $M$ | $\{10, 20, 30\}$ |
| **Inexact-PDO** | $\eta$ | $\{0.05, 0.01, 0.005, 0.001\}$ |
| | $\lambda$ | $\{1, 10\}$ |
| | $M$ | $\{10, 20, 30\}$ |
| **(Accelerated)-SPDO** | $\mu$ | We set the same value as the coefficient of L2 regularization. |
| | $\eta$ | $\{0.05, 0.01, 0.005, 0.001\}$ |
| | $\lambda$ | $\{1, 10\}$ |
| | $M$ | $\{10, 20, 30\}$ |
| **Inexact Accelerated SONATA** | $\eta$ | $\{0.05, 0.01, 0.005, 0.001\}$ |
| | $\beta$ | $\{1, 10\}$ |
| | $M$ | $\{10, 20, 30\}$ |
| | The number of inner loop $T$ | $\{1, 2, 3, 4, 5\}$ |
| | The coefficient of the regularizer $\delta$ | $\{0.1, 1, 10\}$ |

# I. Additional Experimental Results

## I.1. Relationship between $\alpha$ and $\delta$

In our experiments, we used the Dirichlet distribution with hyperparameter $\alpha$ to control $\delta$. In this section, we numerically verified that increasing $\alpha$ can decrease $\delta$. We randomly sampled 100 points from $\mathcal{N}(\mathbf{0}, \frac{1}{\sqrt{2d}}\mathbf{I})$ and reported the approximation of $\delta$. Table 4 indicates that $\delta$ decreases as $\alpha$ increases. Thus, examining the performance of methods with various $\alpha$ is a proper way to evaluate the effect of $\delta$.

*Table 4.* Relationship between $\alpha$ and $\delta$.

| $\alpha$ | 0.01 | 0.1 | 1.0 | 10.0 |
|---|---|---|---|---|
| Approximation of $\delta$ | $1.6 \times 10^{-2}$ | $9.2 \times 10^{-3}$ | $2.1 \times 10^{-3}$ | $5.0 \times 10^{-4}$ |

## I.2. Sensitivity to Hyperparameters

In this section, we evaluated the sensitivity of SPDO to hyperparameters. In Fig. 3, we fixed one hyperparameter, such as $M$, and tuned the other hyperparameters as in Fig. 1(a).

Figure 3(a) indicates that setting $M$ to 3 is optimal. Comparing the results with $M = 1$, increasing the number of gossip averaging decreased the gradient norm. This implies that performing gossip averaging multiple times is important. Furthermore, increasing the number of gossip averaging too much increases the gradient norm since the total number of communications is fixed. These observations are consistent with Theorem 3.

In Fig. 3(b), we evaluated how the choice of $\lambda$ affects the convergence behavior of SPDO. The results indicate that setting $\lambda$ to 1 is optimal. If we use a very large $\lambda$, SPDO requires more communication since the parameters are almost the same even after solving the subproblem. We can see consistent observations in Fig. 3(b).

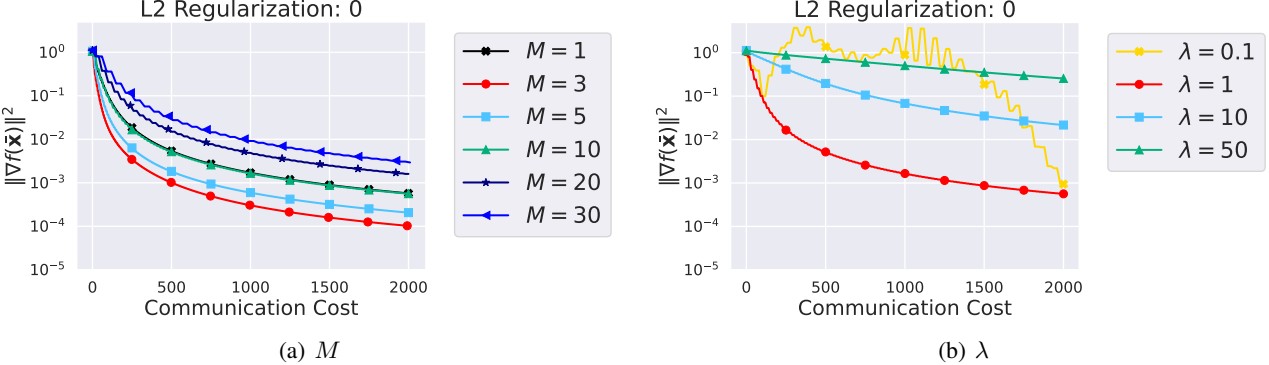

(a) $M$        (b) $\lambda$

*Figure 3.* Convergence of the gradient norm of SPDO with various $M$ and $\lambda$. The experimental setting was same as Fig. 1(a).

## I.3. Experiments with Different Seed Value

Figure 4 shows the results when using a different seed value for the experiments shown in Fig. 1.

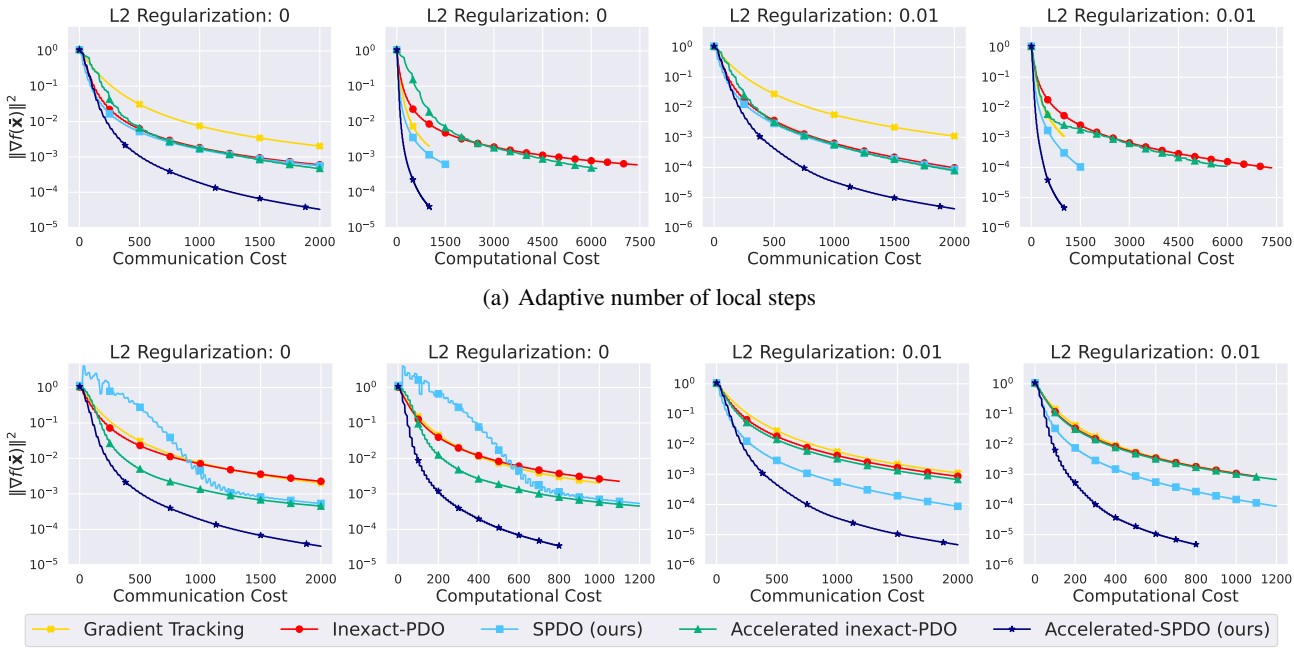

(a) Adaptive number of local steps

(b) Fixed number of local steps

*Figure 4.* Convergence of the gradient norm with $\alpha = 0.1$ when a different seed value was used for the experiments shown in Fig. 1. In (a), we ran gradient descent until the condition for an approximate subproblem solution was satisfied for all methods except for Gradient Tracking. For Gradient Tracking, we ran gradient descent for 10 times. In (b), we ran gradient descent for 10 times to approximately solve the subproblem. See Sec. H for a more detailed setting.

