# OpenReview forum: "Exploiting Similarity for Computation and Communication-Efficient Decentralized Optimization"
_ICML.cc/2025/Conference — ICML 2025 poster_

### Official Review · Reviewer_vkTV · 2025-02-25

**Overall Recommendation:** 3

**Summary:**

This paper introduces the Stabilized Proximal Decentralized Optimization (SPDO) method, which achieves state-of-the-art communication and computational complexities within the Proximal Decentralized Optimization (PDO) framework. The authors also propose an accelerated variant (Accelerated-SPDO) based on the Monteiro and Svaiter acceleration method. The paper is well-written, and the proposed algorithms appear promising. Below are detailed comments and suggestions to further improve the paper.


## update after rebuttal
After the rebuttal, the authors promised to add the proof sketch in the camera-ready version and fix the paper to clarify the contribution.

**Claims And Evidence:**

Yes

**Essential References Not Discussed:**

No

**Experimental Designs Or Analyses:**

Yes

**Methods And Evaluation Criteria:**

Yes

**Other Comments Or Suggestions:**

Suggestion:

- In the main body of the paper, the authors first present the non-accelerated SPDO method followed by the accelerated version. However, the contribution bullets list the accelerated method first. To improve clarity and consistency, it would be better to align the sequence of the contribution bullets with the flow of the paper.

**Other Strengths And Weaknesses:**

Strengths:
- The paper is well-written, and the proposed algorithms appear promising with rigorous theoretical validation.

Weaknesses:
- The paper would benefit from providing a proof sketch for key theoretical results, particularly for the convergence analysis. This would make the paper more accessible to readers.
- The Accelerated-SPDO algorithm seems to be a straightforward extension of SPDO by incorporating the Monteiro and Svaiter acceleration method. However, are there any unique challenges or modifications required to apply this acceleration in the decentralized optimization setting? If so, these challenges should be highlighted and discussed in detail.
- The experimental section is promising but could be significantly strengthened if including experiments on additional datasets and models to validate the generality of the proposed methods.

**Questions For Authors:**

- Can the proposed SPDO and Accelerated-SPDO algorithms be applicable to nonconvex optimization problems?
- The authors claim that SPDO achieves improved communication complexity compared to existing methods. Can the authors provide more intuition or insight into why SPDO achieves this improvement?

**Relation To Broader Scientific Literature:**

This paper addresses key challenges in the literature, particularly in terms of computation and communication efficiency.

**Theoretical Claims:**

No

---

> ### Author Rebuttal · Authors · 2025-04-01
>
> We thank the reviewer for the positive feedback.
>
> > The paper would benefit from providing a proof sketch for key theoretical results, particularly for the convergence analysis.
>
> Thank you for the suggestion.
> We promise to add the proof sketch in the camera-ready version.
>
> > The Accelerated-SPDO algorithm seems to be a straightforward extension of SPDO by incorporating the Monteiro and Svaiter acceleration method. However, are there any unique challenges or modifications required to apply this acceleration in the decentralized optimization setting?
>
> The primary challenge of proposing our methods is that a straightforward combination of Hybrid Projection Proximal-point Method, multiple gossip averaging, and gradient tracking does not work (See the update rule in lines 262-274 and discussion in lines 275-295).
> To overcome this, we proposed a carefully designed modification in Algorithm 3 (see the update rule highlighted in blue).
> The same modification is also necessary for Accelerated SPDO, which is one of the primary novelties of our paper.
>
> > [...] the contribution bullets list the accelerated method first. To improve clarity and consistency, it would be better to align the sequence of the contribution bullets with the flow of the paper.
>
> Thank you for the suggestion.
> We will fix the paper to clarify the contribution.
>
> > Can the proposed SPDO and Accelerated-SPDO algorithms be applicable to nonconvex optimization problems?
>
> PDO, SPDO, and Accelerated-SPDO are designed specifically for convex optimization problems, and their applicability to non-convex optimization is not guaranteed.
> However, developing algorithms in the convex case is an important first step for studying algorithms in the more challenging non-convex case.
> We believe that our work provides valuable insights that could contribute to the development of future algorithms for non-convex optimization problems.
>
> > [...] Can the authors provide more intuition or insight into why SPDO achieves this improvement?
>
> The main drawback of existing methods, including Decentralized SGD and Gradient Tracking, is that they cannot utilize the multiple local steps.
> In their local update, each node performs gradient descent, but fully minimizing the objective function is not desirable.
> For Decentralized SGD, [1] showed that we need to decrease the stepsize when we increase the number of local updates to not fully minimize the objective function.
> Compared with them, PDO, SPDO, and Accelerated PDO can solve the subproblem in their local update, which leads to lower communication complexities.
>
> ## Reference
>
> [1] Koloskova, A., Loizou, N., Boreiri, S., Jaggi, M., and Stich, S. A unified theory of decentralized SGD with changing topology and local updates. In ICML, 2020.

---

> > ### Comment · Reviewer_vkTV · 2025-04-03
> >
> > Thanks for your careful response. I have no further question and decide to keep my rating.

---

> > > ### Author Response · Authors · 2025-04-08
> > >
> > > Thank you for your valuable feedback.

---

### Official Review · Reviewer_pn8d · 2025-03-12

**Overall Recommendation:** 4

**Summary:**

This paper studies decentralized optimization, and it proposes several decentralized methods and analyses their convergence. Specifically, they show that their methods achieve the state-of-the-art communication and computation complexity.

**Claims And Evidence:**

The claims are fair.

**Essential References Not Discussed:**

They mainly discuss gradient-tracking type methods when comparing the computation and communication complexity. I suggest them to also include primal-dual type methods for comparison. To list a few:

[A] S. A. Alghunaim, E. Ryu, K. Yuan, and A. H. Sayed, “Decentralized proximal gradient algorithms with linear convergence rates,” IEEE Trans. Autom. Control, vol. 66, no. 6, pp. 2787–2794, Jun. 2021.

[B] J. Xu, Y. Tian, Y. Sun, and G. Scutari, “Distributed algorithms for com- posite optimization: Unified framework and convergence analysis,” IEEE Trans. Signal Process., vol. 69, pp. 3555–3570, Jun. 2021.

[C] A. Makhdoumi and A. Ozdaglar, “Convergence rate of distributed ADMM over networks,” IEEE Trans. Autom. Control, vol. 62, no. 10, pp. 5082–5095, Oct. 2017.

[D] X. Wu and J. Lu, "A Unifying Approximate Method of Multipliers for Distributed Composite Optimization," in IEEE Transactions on Automatic Control, vol. 68, no. 4, pp. 2154-2169, 2023

The communication complexity of these primal-dual methods are often competitive. For example, the communication complexity of [C] is $O(\frac{\sqrt{\kappa}}{1-\rho})$ where $\kappa$ is the condition number. This communication complexity is competitive compare to that of the non-accelerated methods discussed in Table 1.

**Experimental Designs Or Analyses:**

Didn't find any issue with the experimental design.

**Methods And Evaluation Criteria:**

1. The methods are closely related to two existing works (Scutari & Sun, Li 2020) as discussed in the paper. However, the connection is not clear enough:

- In Page 3, they said that "PDO contains SONATA (Sun et al., 2022) as a special instance when the proximal subproblem is solved exactly. The proofs are deferred to Sec. C and F.2.". I tried to find the explanation of why SONATA is a special case, but didn't find it in Appendix C and F.2.

- In Page 5, it says that "The framework of PDO was initially introduced by Li et al. (2020) and Sun et al. (2022)". However, it's difficult for me to understand their connection when I checked the two references.

2. It is difficult for me to understand the difference between PDO and Stablized PDO. It seems that the update of v in SPDO is simply a gradient descent step with respect to the objective function in the update of x. If so, then $v_i^{(r+1)}$ is also an approximate solution to the minimization problem associated with the update of $x_i$. If this is true, then the novelty and contribution of SPDO will be questionable.

3. The authors should provide more details on the algorithm development. I mean, it is important for the readers to understand the philosophy of the algorithm development and in the current version, it is not easy to understand these methods.

4. Question: In Algorithm 1, the last term in the update of h_i should be $\nabla f_i(x_i^{(r+1)})$ or $\nabla f_i(x_i^{(r)})$? I ask this question mainly because that in gradient-tracking type methods, the updates usually involve old gradients.

5. Typo: In Line 4 of Algorithm 2, it should be $a_j^{(m)}$ rather than $a_j^{(r)}$.

**Other Comments Or Suggestions:**

No

**Other Strengths And Weaknesses:**

Discussed above.

**Questions For Authors:**

No

**Relation To Broader Scientific Literature:**

Unclear.

**Theoretical Claims:**

I didn't check the proof. Regarding the theorem itself, the assumption seems to be very strong, and except for affine functions, it is difficult for me to find any other functions that satisfy Assumption 3.

---

> ### Author Rebuttal · Authors · 2025-04-01
>
> We thank the reviewer for examining our paper.
>
> > The methods are closely related to two existing works (Scutari \& Sun, Li 2020) as discussed in the paper. However, the connection is not clear enough: [...]
>
> Our paper and existing papers [1,2] use a slightly different notation, which might confuse the reviewer.
> We clarify it in the following.
>
> Let define $y_i^{(r)} :=  h_i^{(r)} + \nabla f_i (x_i^{(r)})$. If we replace $h_i$ with $y_i$ in PDO, we obtain almost the same algorithm as Algorithm 1 shown in [1] as SONATA.
> The difference is that PDO solves the subproblem approximately, while SONATA needs to solve it exactly.
> Note that the original SONATA [2] has an additional hyperparameter $\alpha$ (See Algorithm 1 in [2]), but as shown in [1], $\alpha$ is not essential, and we can set $\alpha=1$.
> Thus, more precisely speaking, PDO contains the original SONATA with $\alpha=1$ as a special instance.
>
> If the reviewer still has concerns, we would appreciate being informed.
> We will be happy to resolve them.
>
> > It is difficult for me to understand the difference between PDO and Stablized PDO. [...] If this is true, then the novelty and contribution of SPDO will be questionable.
>
> We respectfully disagree with this reviewer's comment.
> Thanks to the update rule of $v$, Stabilized PDO can solve the subproblem more coarsely than PDO (see Eqs. (8) and (9)) and is computationally less expensive.
>
> Specifically, the update rule of $v$ is not a simple gradient descent step.
> When $\mu=0$, the update rule of $v$ is $v_i^{(r)} - \frac{1}{\lambda} (\nabla f_i (x_i^{(r+1)}) + h_i^{(r)})$. The gradient is computed at $x$ instead of $v$, and it is not a simple gradient descent.
> It plays a crucial role in reducing computational complexity.
>
> > In Algorithm 1, the last term in the update of $h_i$ should be $\nabla f_i(x_i^{(r+1)})$ or $\nabla f_i(x_i^{(r)})$? [...]
>
> Thank you for carefully checking our algorithm.
> This is not a typo, and Algorithm 1 is correct.
> We wonder if the reviewer might be confused since we use slightly different notations from [1] and [2].
> See our response to your first comment.
> If the reviewer still has concerns, we would be glad to resolve them.
>
> > Typo: In Line 4 of Algorithm 2, it should be $a_j^{(m)}$ rather than $a_j^{(r)}$.
>
> Thank you for pointing this out.
> We will fix it in the revised manuscript.
>
> > Regarding the theorem itself, the assumption seems to be very strong, and except for affine functions, it is difficult for me to find any other functions that satisfy Assumption 3.
>
> We would respectfully disagree with this reviewer's comment.
> **All of our theorems, Theorems 1-6, use only Assumptions 1, 2, and 4 and do not use Assumption 3.**
>
> As we explained in lines 120-126, the existing methods, SONATA and Accelerated SONATA, used Assumption 3, while our proposed methods and theorems successfully omit Assumption 3. Instead of using Assumption 3, we use Definition 1. As we mentioned in Remark 1, Definition 1 is not an assumption since **all $L$-smooth functions satisfy Eq. (3) with $\delta \leq 2 L$.**
> We can check this by the following inequalities:
> \begin{align*}
> \frac{1}{n} \sum_{i=1}^n \| \nabla h_i(x) - \nabla h_i (y)\|^2
> \leq 2 \| \nabla f (x) - \nabla f(y)\|^2 + \frac{2}{n} \sum_{i=1}^n \| \nabla f_i(x) - \nabla f_i (y)\|^2 \leq 4 L^2 \| x - y\|^2
> \end{align*}
>
> > [...] I suggest them to also include primal-dual type methods for comparison.
>
> Thank you for your suggestion.
> The papers the reviewer mentioned [A,B,C,D] do not utilize the similarity $\delta$. Thus, our proposed methods, PDO, Stabilized PDO, and Accelerated-PDO, can achieve better communication complexity by utilizing the similarity of local functions, especially when $\delta \ll L$.
> We will cite the papers the reviewer mentioned and add the discussion in the revised manuscript.
>
>
> We believe these additions clarify the relationship between existing and our proposed methods, strengthening our paper.
> Therefore, we kindly ask the reviewer to reassess their score. If further concerns remain, we are happy to address them.
>
>
> ## Reference
> [1] Tian, Y., Scutari, G., Cao, T., and Gasnikov, A. Acceleration in distributed optimization under similarity. In AISTATS, 2022.
>
> [2] Sun, Y., Scutari, G., and Daneshmand, A. Distributed optimization based on gradient tracking revisited: Enhancing convergence rate via surrogation. In SIAM Journal on Optimization, 2022.

---

### Official Review · Reviewer_6acg · 2025-03-15

**Overall Recommendation:** 3

**Summary:**

The paper studies decentralized optimization where multiple nodes, each holding a local function f_i, aim to minimize the average f(x) = \tfrac{1}{n}\sum_i f_i(x). Traditional decentralized methods are constrained by communication overhead and data heterogeneity. The authors propose a Proximal Decentralized Optimization (PDO) framework that leverages (1) a proximal-point formulation (improving upon naive gradient-based updates) and (2) a refined measure of local function similarity (replacing reliance on the worst-case delta_{\max} with an average \delta).
To handle large models realistically, they introduce two variants:
1. Stabilized-PDO (SPDO)– A “stabilized” method that relaxes the requirement for exact subproblem solutions, ensuring that even inexact local solves yield competitive global convergence rates.
2. Accelerated-SPDO – Extends the above approach with an acceleration scheme (inspired by Monteiro–Svaiter) and faster gossip averaging, targeting improved communication complexities in both convex and strongly convex settings.

**Claims And Evidence:**

1. The authors do not measure or estimate 𝛿 (the second-order similarity measure) directly on these data splits, so the link between “Dirichlet 𝛼” and actual functional similarity is inferred but not verified.

2. The paper does not compare the cost of multiple gossip rounds vs. direct use of a more advanced averaging approach (like exponentiated gradient-based gossip). Hence, the evaluation partly relies on a simplified measure of “communication rounds” that may not reflect real overhead in heterogeneous network conditions.

3. The authors do not show error bars or repeated runs to reveal the variance. If the approach can converge quickly in a median sense but sometimes fails or stalls, that variability matters.

4. Real distributed environments might have node-level heterogeneity in CPU/GPU power. The paper’s simple model (uniform local iteration cost) might mask how well the proposed method handles uneven computation resources.

5. They do not evaluate generalization or test accuracy on complex tasks, even though logistic regression on MNIST is at least somewhat classification-driven. Since they focus on the training objective, it remains unclear whether the improved convergence speed translates to superior test-time performance or any difference in model quality for real tasks.

**Essential References Not Discussed:**

NA.

**Experimental Designs Or Analyses:**

1.  Nowhere do the authors directly measure or approximate \delta (the second-order dissimilarity measure) in the actual experiments. Instead, they rely on \alpha from the Dirichlet distribution as a surrogate. Mathematically, it would be more rigorous to approximate \delta in practice—for instance by sampling the norms \|\nabla f_i(x)-\nabla f_j(x)\| for random x values—and plotting them, so the audience can see how \delta scales with \alpha. This would confirm that changes in \alpha truly reflect the second-order similarity the theorems rely on.

2. When regularization is small or absent \mu \approx 0, the theoretical bounds predict the “convex case” complexities. However, the experiments do not provide a quantitative link between \mu and the actual speed. In particular, one expects the theoretical number of rounds for a certain \varepsilon-accuracy to be on the order of \mathcal{O}(\tfrac{\delta}{(1-\rho)\varepsilon}) if \mu = 0. Yet the main figures only show *empirical* curves without clarifying whether they align with or deviate from \mathcal{O}(\frac{1}{\varepsilon} or \mathcal{O}(\log(\tfrac{1}{\varepsilon})). It would be helpful to fit the observed convergence data to a model or produce a slope in log–log space, so the experimental results can be interpreted alongside the proposed theorems.

3. It would strengthen the experimental section if the authors demonstrated at least one experiment systematically varying M—for instance, letting M take on \{1, 5, 10, 20\}—so that one can see how error from incomplete averaging impacts performance. For example, if \rho is large (due to a sparse topology), the authors claim we need more gossip steps to maintain \rho^M \le \tfrac{\delta}{6L}. But no chart in the paper explicitly shows how changing M in practice affects communication overhead, final test accuracy, or speed of convergence. This is a missed opportunity to confirm the theory.

4.  In the main results, multiple hyperparameters—\(\lambda\), \(M\), local-step learning rate \(\eta\), etc.—are presumably tuned. However, the paper shows only a single set of final curves comparing the proposed method with baselines.  It would be more persuasive to include plots (or at least a table) that show how the final performance depends on each hyperparameter. For example, if \(\lambda\) is set too large, do local subproblems become trivial or degenerate, and does that hamper performance? If \(\eta\) in the local subproblem solver is too large, do we see instability? The authors do mention some tuning, but they do not systematically present the experimental process. This makes reproducibility more difficult, and it also leaves open the question of how sensitive the proposed method is to hyperparameters compared to classical gradient tracking or SONATA.

5. The paper’s plots do not show confidence intervals or variance across runs. This is standard practice in many experimental ML contexts to gauge robustness.  If the proposed method’s advantage is that it converges faster in both “communication rounds” and “computation steps,” but at the cost of some potential variance from the approximate subproblem solutions, readers should see that tradeoff empirically. For instance, one might observe that fewer gossip steps or fewer local subproblem iterations lead to higher variance in the final solution. The paper could show standard deviation bands or quartiles across multiple random seeds.

**Methods And Evaluation Criteria:**

1. The paper focuses on scenarios where nodes each hold unique local data and where communication is expensive (ring, mesh, or general sparse networks). The proposed proximal-point-based methods (Inexact-PDO, SPDO, Accelerated-SPDO) are appropriate because they are designed to exploit partial overlap or “similarity” among local data while mitigating the high cost of frequent parameter exchanges.
   - **Proximal Decentralized Framework**: The premise that each node solves a proximal subproblem (to reduce the discrepancy introduced by local data differences) matches well with the objective of improving the method’s tolerance to data heterogeneity. By explicitly modeling subproblem accuracy, the approach is well-suited for large-scale problems where exact solutions would be prohibitive.

2. **Evaluation Criteria in Experiments**
   - **Communication Cost vs. Computation Cost**: The paper measures success primarily through (i) *communication rounds* needed to achieve a target accuracy and (ii) total *local gradient steps* or “computation.” These criteria are standard in the literature for decentralized and federated settings, where the ratio of communication time to local computation time can be crucial.
   - **Choice of Datasets**: MNIST is typical for small-scale proof-of-concept experiments, and the authors artificially control heterogeneity (via the Dirichlet parameter \(\alpha\)), which *does* illustrate how the algorithms handle varying levels of data similarity. Although it is limited in scope, it still effectively shows the benefit of reduced communication for more homogeneous local data.
   - **Metric of Accuracy**: The authors primarily track objective function decrease or gradient-norm decrease as a function of both communication and computational steps, consistent with standard optimization benchmarks.

3. **Potential Gaps or Areas for Improvement**
   - **More Diverse Benchmarks**: While MNIST is a reasonable start, real-world distributed data could be more irregular than a simple Dirichlet partition. Adding more varied datasets (e.g., from large-scale text or image tasks, or from industry-scale streaming data) would broaden the demonstration of the method’s performance.
   - **Full Validation of \(\delta\)**: The authors claim that as \(\alpha\) increases, local objectives become more similar (\(\delta\) decreases). However, they do not empirically measure or approximate \(\delta\). A direct empirical validation of \(\delta\)’s role would strengthen the link between method and claimed advantages.

Overall, the proposed methods and evaluation criteria (communication rounds, local gradient steps, final training loss or gradient norm) are coherent for the paper’s decentralized-learning context. While the experimental design could be expanded with more datasets and more direct measurement of data similarity, the chosen metrics and problem setups do largely align with how such methods are tested in the broader decentralized optimization literature.

**Other Comments Or Suggestions:**

NA

**Other Strengths And Weaknesses:**

1. While the paper introduces a well-crafted approach, it is also somewhat incremental: it adapts prior tools (e.g. Monteiro–Svaiter acceleration, Gossip averaging, SONATA) rather than introducing a wholly new algorithmic paradigm. The novelty lies chiefly in how these ideas are fused, rather than in an entirely new foundational concept.
2. Although the appendices are extensive, the main body can feel dense at times. Key distinctions—like why the new “stabilized” scheme so dramatically eases local subproblem accuracy demands—could be emphasized more. Some readers may need more immediate intuition or examples within the text (rather than buried in appendices).
3. The methods involve multiple hyperparameters (λ, 𝑀, local step sizes, acceleration constants). The paper gives guidelines, but does not comprehensively detail how sensitive the final performance is to these parameters.

**Questions For Authors:**

NA

**Relation To Broader Scientific Literature:**

1. The paper fundamentally relies on the classic Proximal-Point Method, originally studied in the single-node, centralized setting. Past work has shown that one can address inexact subproblem solutions via “hybrid” or “projection-based” proximal iterations (Solodov, Mikhail V., and Benar Fux Svaiter. "A new projection method for variational inequality problems." SIAM Journal on Control and Optimization 37.3 (1999): 765-776. Monteiro, Renato DC, and Benar F. Svaiter. "Iteration-complexity of block-decomposition algorithms and the alternating direction method of multipliers." SIAM Journal on Optimization 23.1 (2013): 475-507.). By adapting these ideas to the decentralized environment, this paper continues a line of research on how to relax exact subproblem requirements while still maintaining strong theoretical guarantees.

2. In the decentralized setting, accelerating methods like SONATA or gradient-based approaches often requires delicate handling of local steps or momentum terms. (Tian, Ye, et al. "Acceleration in distributed optimization under similarity." International Conference on Artificial Intelligence and Statistics. PMLR, 2022) for second-order similarity (with \delta_{\max}) but mandated exact or high-precision subproblem solves at each round.

**Theoretical Claims:**

1. The paper introduces \delta via
       \frac{1}{n} sum_{i=1}^n \|\nabla h_i(x) - \nabla h_i(y)\|^2
       \delta^2 \,\|x - y\|^2,
       where h_i(x) = f(x) - f_i(x).

     This condition implies a “second-order” type similarity across nodes. However, in practical large-scale systems, it often suffices to assume the simpler Lipschitz-smooth condition on each f_i. Your proposed approach places an emphasis on the difference h_i as if each node’s local function is nearly identical to f.
The text does not thoroughly discuss the real-world consequences if \delta is not as small as assumed. In particular, while \delta\leq L always holds when each f_i is L-smooth, the presentation glosses over the implications of \delta being close to L. When \delta\approx L, do the improvements over classical decentralized methods persist or vanish? A deeper quantitative exploration would strengthen the argument.

2. In the “Inexact-PDO” framework, one solves, at each node i:
       x_{i}^{(r + 1/2)}
       \approx
       \arg\min_{x} (
         f_i(x)+\;\langle h_i^{(r)},\,x\rangle + \tfrac{\lambda}{2}\|x - x_{i}^{(r)}\|^2
       ).

     This subproblem solution is then subjected to a gradient-norm accuracy requirement, for instance

       \sum_{i=1}^n \|\nabla F_{i,r}(x_{i}^{(r + 1/2)})\|^2
       \frac{\delta\,(4\,\delta+\mu)}{4(r+1)(r+2)}
      \sum_{i=1}^n
       \|x_{i}^{(r + 1/2)} - x_i^{(r)}\|^2.
     or something of that flavor depending on the theorem.
The paper asserts (e.g., Theorem 1 and subsequent corollaries) that as long as these local subproblems are solved “just enough,” one obtains the same rate as an exact solution. However, it leaves open the question of how many iterations of, say, Nesterov’s or plain gradient descent one actually needs on typical problem instances before the sum of squared gradient norms condition is met. In a typical large neural-net scenario, bounding the subproblem error explicitly would require non-trivial additional assumptions. The paper’s guidelines could be clearer, for example by including bounds on the per-round local iteration cost or enumerating precisely how local complexity scales with \delta \mu, and the network size n.

3.  The authors emphasize that the new method “only” needs \mathcal{\tilde O}(\frac{\delta}{\mu(1-\rho)}\log(\frac{1}{\varepsilon}) communications in the strongly convex case, improving from \delta_{\max} to \delta.
At the same time, they do not solve each local subproblem exactly, so the iteration cost is typically \mathcal{\tilde O}(\sqrt{\tfrac{L}{\delta+\mu}}) or something close (depending on the theorem) for each subproblem.  Although the paper is correct that \delta < \delta_{\max} may confer a significant advantage if local data are indeed “similar,” it might also be that local subproblems require many gradient steps if \delta is not truly small relative to L. The statements about “lower communication cost” can be somewhat overshadowed if the local computations blow up.

4.  While the authors adapt the classic Hybrid Projection Proximal-Point idea (Solodov & Svaiter style) to the decentralized setting, the transitions from \mathbf{x}-updates to \mathbf{v}-updates are presented somewhat abruptly. The role of the stabilizing variable \mathbf{v}, especially in how it prevents the subproblems from requiring ever-increasing precision, is mathematically elegant but might be clearer if placed in a stand-alone lemma that (i) proves stability and (ii) ensures the same or better rate as the simpler approach. Currently, the text interweaves the definitions with the main theorems, and it is easy for the reader to lose track of the key steps that guarantee the claimed \mathcal{\tilde O}(\log(\frac{1}{\varepsilon})) complexity.

---

> ### Author Rebuttal · Authors · 2025-04-01
>
> We thank the reviewer for your constructive comments.
> We have addressed the concerns about missing ablation studies and included the results in our rebuttal below. We agree that these additional experiments significantly strengthen our paper. We kindly ask the reviewer to reconsider the evaluation and, if possible, adjust the score to reflect these changes.
>
> > The authors do not measure or estimate $\delta$ (the second-order similarity measure) directly on these data splits [...]
>
> Thank you for your suggestion.
> We randomly sampled $100$ points from $\mathcal{N}(0, I / \sqrt{2d})$ and reported the approximation of $\delta$.
> The following table indicates that $\delta$ decreases as $\alpha$ increases.
> Thus, examining the performance of methods with various $\alpha$ is a proper way to evaluate the effect of $\delta$.
> We promise to add this table in the camera-ready version.
>
> |  $\alpha$ | $0.01$ | $0.1$ | $1.0$ | $10.0$  |
> |---|---|---|---|---|
> | Approximation of $\delta$ | $1.6 \times 10^{-2}$ | $9.2 \times 10^{-3}$ | $2.1 \times 10^{-3}$ | $5.0 \times 10^{-4}$ |
>
>  > The authors do not show error bars or repeated runs to reveal the variance.
>
> We promise to run our experiments several times and report their variance in the camera-ready version.
>
> > Real distributed environments might have node-level heterogeneity in CPU/GPU power.
>
> We agree with the importance of considering the settings where each node has different computational resources. However, developing algorithms in this simple setting is an important first step for studying algorithms in the more challenging setting, as the reviewer mentioned. We believe that our work provides helpful insights that could contribute to the development of future algorithms for more realistic settings.
>
> > It would strengthen the experimental section if the authors demonstrated at least one experiment systematically varying M—for instance, letting $M$ take on $\{1, 5, 10, 20\}$ [...]
>
> Thank you for the suggestion.
> In the following table, we fixed the number of multiple gossip averaging $M$ of SPDO and listed the gradient norm reached after $2000$ communication.
>
> | $M$  | $1$ | $2$ | $3$ | $5$ | $10$ | $20$ |
> |---|---|---|---|---|---|---|
> | $\| \nabla f (\bar{x}) \|^2$ (SPDO) | $9.13 \times 10^{-5}$ | $8.32 \times 10^{-5}$ | $2.24 \times 10^{-5}$ | $1.01 \times 10^{-5}$ | $8.55 \times 10^{-5}$ | $5.13 \times 10^{-4}$ |
>
> The hyperparameters, except for $M$, were tuned as in Figure 1(a) with $0.01$ L2 regularization.
> The table indicates that setting $M$ to $5$ is optimal.
> Comparing the results with $M=1, 2, 3, 5$, increasing the number of gossip averaging decreased the gradient norm.
> This implies that performing gossip averaging multiple times is important.
> Furthermore, increasing the number of gossip averaging too much increases the gradient norm since the total number of communication is fixed (See lines 422-425).
> These observations are consistent with Theorem 3.
>
> We promise to add this table to the revised manuscript.
>
> > It would be more persuasive to include plots (or at least a table) that show how the final performance depends on each hyperparameter. [...]
>
> According to the reviewer's suggestion, we examined the sensitivity of $\lambda$.
> The following table shows the gradient norm reached after running algorithms for $2000$ communication.
> |  $\lambda$ | $0.1$ | $1$ | $10$ | $50$ |
> |----|---|---|---|---|
> | $\| \nabla f (\bar{x})\|^2$ (SPDO) | $1.13 \times 10^{-1}$ | $8.55 \times 10^{-5}$  | $1.69 \times 10^{-2}$  | $2.44 \times 10^{-1}$  |
>
> If we use a very large $\lambda$, the number of iterations of a local solver can decrease (see Lemma 31), while it requires more communication since the parameters are almost the same even after solving the subproblem, which ultimately requires a large number of communications.
> We can see consistent observations in the above table.
>
> We will numerically analyze the sensitivity of other hyperparameters and promise to report the results in the revised manuscript.
>
> > When $\delta\approx L$, do the improvements over classical decentralized methods persist or vanish?
>
> In the worst case, such as $\delta \approx L$, Stabilized PDO and Gradient Tracking require the same communication complexities.
> However, in many practical scenarios, $\delta$ is smaller than $L$ [2,3].
> For instance, many existing papers, e.g., [1], considered that Dirichlet distribution with $\alpha=0.1$ was heterogeneous. Even in this setting, Figure 1 indicates that our proposed methods can achieve lower communication complexities than Gradient Tracking.
> We will clarify it in the revised manuscript.
>
> ## Reference
> [1] Lin et. al., Quasi-global momentum: Accelerating decentralized deep learning on heterogeneous data. In ICML 2021
>
> [2] Chayti et. al., Optimization with Access to Auxiliary Information, In TMLR 2024
>
> [3] Kovalev et. al., Optimal gradient sliding and its application to optimal distributed optimization under similarity.
> In NeurIPS 2022.

---

> > ### Comment · Reviewer_6acg · 2025-04-08
> >
> > Thanks for the authors' effort addressing the concerns. The concerns raised up were almost clarified and addressed. I would update my score accordingly.

---

> > > ### Author Response · Authors · 2025-04-08
> > >
> > > Thank you for raising your score. Once again, we sincerely appreciate the reviewer’s insightful comments.

---

### Official Review · Reviewer_aFm6 · 2025-03-17

**Overall Recommendation:** 3

**Summary:**

This paper provides a decentralized optimization method for convex optimization under the second-order similarity. The main contribution is improving the term $\delta_{\max}$ or $L$ to $\delta$ in the complexity to $\delta$.

## update after rebuttal
The authors have addressed my questions, and I decided to keep my overall rating.

**Claims And Evidence:**

Yes.

**Essential References Not Discussed:**

No.

**Experimental Designs Or Analyses:**

The comparison on the computational cost should be involved.

**Methods And Evaluation Criteria:**

Yes.

**Other Comments Or Suggestions:**

See questions.

**Other Strengths And Weaknesses:**

See questions.

**Questions For Authors:**

This paper is well-written and well-motivated. I have briefly sketched the proof, which sounds correct. There are some comments:
1. The main idea seems be to directly combine the inexact gradient sliding (Kovalev et al., 2022) with Multiple Gossip and gradient tracking. Can you highlight the main technical novelty in the algorithm and analysis?
2. The domains of $\bf x$ and $\bf v$ in lines 4-5 of Algorithm 3 and lines 12-13 in Algorithm 4 should be presented.
3. The experimental results only includes the comparison on the communication cost. I think the comparison on the computational cost is also required.
4. Although the main theorems improve the previous ones, the further discussion for the potentially better results is desired:

      a) Can we avoid the term $1/\sqrt{1-\rho}$ in the computational cost?

      b) Can we introduce the partial participate framework to reduce the complexity of local first-order oracle?

      c) Can we provide the lower bound the verify the optimality of proposed algorithms?

    The following references may be helpful to the discussion:

[1] Haishan Ye, Luo Luo, Ziang Zhou, and Tong Zhang. Multi-consensus decentralized accelerated gradient descent. Journal of machine learning research 24(306):1-50, 2023.

[2] Qihao Zhou, Haishan Ye, and Luo Luo. Near-Optimal Distributed Minimax Optimization under the Second-Order Similarity. In Advances in Neural Information Processing Systems, 2024.

[3] Aleksandr Beznosikov, Gesualdo Scutari, Alexander Rogozin, and Alexander Gasnikov. Distributed saddle-point problems under data similarity. In Advances in Neural Information Processing Systems, 2021.

**Relation To Broader Scientific Literature:**

See questions.

**Theoretical Claims:**

I have briefly read the proofs and it sounds correct.

---

> ### Author Rebuttal · Authors · 2025-04-01
>
> We thank the reviewer for the positive feedback and careful review.
>
> > The main idea seems be to directly combine the inexact gradient sliding (Kovalev et al., 2022) with Multiple Gossip and gradient tracking. Can you highlight the main technical novelty in the algorithm and analysis?
>
> We would like to emphasize that **a straightforward combination of gradient sliding, multiple gossip averaging, and gradient tracking does not work**. To overcome this, we proposed a carefully designed modification in Algorithm 3 (see the update rule highlighted in blue).
>
> Specifically, straightforwardly combining them yields the update rules shown in lines 263-274. However, as we described in lines 275-289, this simple combination does not work.
> Our paper is the first to analyze this challenge and develop a principled modification that ensures low communication and computational complexities.
>
> > The domains of $x$ and $v$ in lines 4-5 of Algorithm 3 and lines 12-13 in Algorithm 4 should be presented.
>
> The domains of $x$ and $v$ are $\mathbb{R}^d$. We will clarify them in the camera-ready version.
>
>
> > The experimental results only includes the comparison on the communication cost. I think the comparison on the computational cost is also required.
>
> The comparison of the computational costs is shown in the second and fourth figures in Figure 1.
> The first and third figures from the left show the communication complexities.
> We will clarify it in the revised manuscript.
>
> > Can we avoid the term $1/\sqrt{1-\rho}$ in the computational cost?
>
> We deeply appreciate the reviewer for checking it carefully.
> There are typos in Table 1; the computational costs for methods other than Gradient Tracking do not depend on $\rho$ since $\tfrac{1}{\sqrt{1 - \rho}}$ in communication complexity comes from the multiple gossip averaging, which does not affect the total computational complexity.
> The statements of our theorems in the paper are correct, and this correction does not affect the discussion in the entire paper.
> We apologize for your confusion.
> We promise to replace Table 1 with the following version in the revised manuscript.
>
> | Method                        | # Computation |
> |--------------------------------|---------------------------------------------|
> | Gradient Tracking             | $\tilde{\mathcal{O}} (\frac{L}{\mu (1 - \rho)^2} \log (\frac{1}{\epsilon}))$ |
> | Exact-PDO (SONATA)            | n.a. |
> | Inexact-PDO                   | $\tilde{\mathcal{O}} (\frac{\sqrt{\delta L}}{\mu} \log (\frac{1}{\epsilon}) \log \log (\frac{1}{\epsilon}))$ |
> | Stabilized-PDO                | $\tilde{\mathcal{O}} (\frac{\sqrt{\delta L}}{\mu} \log (\frac{1}{\epsilon}))$ |
> | Accelerated SONATA            | n.a. |
> | Inexact Accelerated SONATA    | $\tilde{\mathcal{O}} (\frac{\sqrt{\delta L}}{\mu} \log (\frac{1}{\epsilon})^2)$ |
> | Accelerated Stabilized-PDO    | $\tilde{\mathcal{O}} (\sqrt{\frac{L}{\mu}} \log (\frac{1}{\epsilon}))$ |
>
> > Can we provide the lower bound the verify the optimality of proposed algorithms?
>
> Thank you for the comments. We will add the following discussion in the camera-ready version.
>
> **Communication Complexity:**
> [1] showed that lower bound requires at least
> \begin{align*}
>     \Omega \left( \sqrt{\frac{\delta}{\mu (1 - \rho)}} \log \left( \frac{\mu \| x^\star \|^2}{\epsilon} \right) \right)
> \end{align*}
> communication to achieve $f (x) - f (x^\star) \leq \epsilon$.
> Our Accelerated SPDO can achieve the following communication complexity:
> \begin{align*}
>     \tilde{\mathcal{O}} \left( \sqrt{\frac{1 + \frac{\delta}{\mu}}{1 - \rho}} \log \left( 1 + \sqrt{\frac{\min \\{ \mu, \delta\\} \|x^{(0)} - x^\star \|^2}{\epsilon}} \right) \right)
> \end{align*}
> Thus, when $\delta \geq \mu$, Accelerated SPDO is optimal up to the logarithmic factor.
>
> **Computational Complexity:**
> For non-distributed cases, any first-order algorithms requires at least
> \begin{align*}
>     \Omega \left( \sqrt{\frac{L}{\mu}} \log \left(\frac{\mu \| x^{(0)} - x^\star \|^2}{\epsilon} \right) \right)
> \end{align*}
> gradient-oracles to satisfy $f(x) - f^\star \leq \epsilon$ (See Theorem 2.1.13 in [2]).
> Accelerated SPDO can achieve the following computational complexity:
> \begin{align*}
>     \tilde{\mathcal{O}} \left( \sqrt{\frac{L}{\mu}} \log \left( 1 + \sqrt{\frac{\min \\{ \mu, \delta\\} \|x^{(0)} - x^\star \|^2}{\epsilon}} \right) \right)
> \end{align*}
> Thus, the computational complexity of Accelerated SPDO is optimal up to logarithmic factors.
>
> ## Reference
>
> [1] Tian, Y., Scutari, G., Cao, T., and Gasnikov, A. Acceleration in distributed optimization under similarity. In AISTATS, 2022.
>
> [2] Nesterov, Y. Lectures on convex optimization. In Springer, 2018.

---

> > ### Comment · Reviewer_aFm6 · 2025-04-01
> >
> > Thanks for your careful response. I have no further question and decide to keep my rating.

---

> > > ### Author Response · Authors · 2025-04-08
> > >
> > > Thank you for your valuable feedback.

---

### Decision · Program_Chairs · 2025-05-01

**Decision:**

Accept (poster)

**Comment:**

All reviewers agree that this paper makes a solid contribution to the field of decentralized optimization and recommend acceptance. The paper addresses a key limitation of existing DPO methods, which often rely on highly accurate solutions to proximal subproblems and thus incur high computational costs. The proposed SPDO method effectively overcomes this issue, achieving state-of-the-art communication and computation complexities. Additionally, the authors provide a refined theoretical analysis by relaxing subproblem accuracy requirements and leveraging average functional similarity. The results are well-motivated, clearly presented, and supported by solid theoretical and empirical evidence. I recommend acceptance.